# MAVS signaling is required for preventing persistent chikungunya heart infection and chronic vascular tissue inflammation

Maria G. Noval[1,7] ✉, Sophie N. Spector[1], Eric Bartnicki [1], Franco Izzo[2,3], Navneet Narula[4], Stephen T. Yeung[1], Payal Damani-Yokota[1], M. Zahidunnabi Dewan[4], Valeria Mezzano[4], Bruno A. Rodriguez-Rodriguez[1], Cynthia Loomis[4,5], Kamal M. Khanna [1,6] & Kenneth A. Stapleford[1,7] ✉

Chikungunya virus (CHIKV) infection has been associated with severe cardiac manifestations, yet, how CHIKV infection leads to heart disease remains unknown. Here, we leveraged both mouse models and human primary cardiac cells to define the mechanisms of CHIKV heart infection. Using an immuno-competent mouse model of CHIKV infection as well as human primary cardiac cells, we demonstrate that CHIKV directly infects and actively replicates in cardiac fibroblasts. In immunocompetent mice, CHIKV is cleared from cardiac tissue without significant damage through the induction of a local type I interferon response from both infected and non-infected cardiac cells. Using mice deficient in major innate immunity signaling components, we found that signaling through the mitochondrial antiviral-signaling protein (MAVS) is required for viral clearance from the heart. In the absence of MAVS signaling, persistent infection leads to focal myocarditis and vasculitis of the large vessels attached to the base of the heart. Large vessel vasculitis was observed for up to 60 days post infection, suggesting CHIKV can lead to vascular inflammation and potential long-lasting cardiovascular complications. This study provides a model of CHIKV cardiac infection and mechanistic insight into CHIKV-induced heart disease, underscoring the importance of monitoring cardiac function in patients with CHIKV infections.

Arthropod-borne viruses (arboviruses) such as Zika virus, dengue virus, and chikungunya virus (CHIKV) are associated with the development of cardiomyopathies in patients[1–6]. CHIKV-associated cardiac manifestations have been reported in more than 15 countries worldwide[5,7]. Cardiac complications occur within the first week after symptom onset[5], and include symptoms ranging from arrhythmias, atrial fibrillation, abnormal echocardiograms and electrocardiograms[8],

reduction of the ejection fraction[6], myocarditis[8–10], to heart failure, and death[9,11–13]. Autopsies of individuals that succumbed to CHIKV infection reveal the presence of CHIKV antigen in the cardiac tissue[12,14] and immune cell infiltration indicative of viral myocarditis, as well as signs of cardiac edema, endocarditis, necrosis, and cardiac congestion[9]. Indeed, more than 20% of the CHIKV-related mortality cases have been associated with cardiac complications[9,12,15]. Although most cardiac

[1]Department of Microbiology, New York University Grossman School of Medicine, New York, NY, USA. [2]New York Genome Center, New York, NY, USA. [3]Division of Hematology and Medical Oncology, Department of Medicine and Meyer Cancer Center, Weill Cornell Medicine, New York, NY, USA. [4]Department of Pathology, New York University Grossman School of Medicine, New York, NY, USA. [5]Division of Advanced Research Technologies, New York University Grossman School of Medicine, New York, NY, USA. [6]Perlmutter Cancer Center, New York University Grossman School of Medicine, New York, NY, USA. [7]These authors jointly supervised this work: Maria G. Noval, Kenneth A. Stapleford. ✉e-mail: maria.noval@nyulangone.org; kenneth.stapleford@nyulangone.org

complications have been observed in older adults and individuals with comorbidities[11,16], cardiac manifestations associated with CHIKV infections have been reported in young individuals, including children and infants without known comorbidities[6,8,9].

While previous studies reported a link between CHIKV and infection of cardiac tissue in animal models[17–21], several questions remain unanswered: (i) is cardiac tissue a direct target of CHIKV infection and a site of active replication in immunocompetent hosts?; (ii) what are the mechanisms of CHIKV-cardiac tissue interaction?; (iii) how does the heart respond to CHIKV infection?; and (iv) how does CHIKV infection lead to heart disease?.

Here, using an immunocompetent mouse model of CHIKV infection as well as human primary cardiac cells, we demonstrate that CHIKV directly infects and actively replicates within cardiac tissue, with cardiac fibroblasts being the main cellular target of CHIKV infection. We show that CHIKV infects the myocardium, valves, atrium, and vessels, and that viral RNA persists longer in tissue containing atrium and vessels in comparison to the myocardium. We show that CHIKV-heart infection is rapidly cleared without signs of tissue inflammation or cardiac tissue damage in immunocompetent mice coinciding with a local type I interferon (IFN-I) response. Indeed, cardiac tissue infection induces a local IFN-I response in both infected and non-infected cardiac cells, and the loss of IFN-I signaling results in increased CHIKV infectious particles, virus spreading, and apoptosis in cardiac tissue. We demonstrate that the IFN-I response is essential to control CHIKV infection in human primary cardiac fibroblasts. In addition, we found that signaling through the mitochondrial antiviral-signaling protein (MAVS), a central hub for signal transduction initiated by cytosolic pattern recognition receptors (PRRs) RIG-I or MDA5, is required for efficient CHIKV clearance, with viral particles being detected up to 10 days post infection (dpi) in $Mavs^{-/-}$ mice. Importantly, we found that the persistence of CHIKV infection in cardiac tissue of $Mavs^{-/-}$ mice leads to cardiac tissue damage characterized by CD3[+] and CD11b[+] infiltrates in the myocardium and atrium as well as in the large vessels attached to the base of the heart, such as the aorta (Ao) and the pulmonary artery (PA). Interestingly, viral myocarditis was detected at 10 and 15 dpi, while large vessel vasculitis involving the Ao and PA persisted up to 60 dpi, suggesting that infection of cardiac tissue by CHIKV can lead to vascular inflammation and potential long-lasting cardiovascular complications.

Overall, our results show that a robust IFN-I response and MAVS signaling are required to control CHIKV infection in the heart and prevent focal myocarditis as well as Ao and PA vasculitis. Altogether, this study provides mechanistic insight into how CHIKV can lead to cardiac damage, underscoring the importance of monitoring cardiac function in patients with CHIKV infections and paving the way towards the identification of risk factors associated to the development of CHIKV-induced cardiac complications.

## Results

### CHIKV actively replicates in cardiac tissue and targets cardiac fibroblasts

While the presence of CHIKV viral products have been detected in cardiac tissue of infected animals[17–21], whether CHIKV is infecting and replicating within cardiac cells remains elusive. Cardiac tissue is highly irrigated; therefore, it is essential to verify that the virus detected in heart homogenates originates from infected cardiac cells rather than circulating virus. To address this, we tested the effect of perfusion on the highly susceptible interferon α/β receptor-deficient mice ($Ifnar1^{-/-}$), a mouse model in which CHIKV infectious particles were detected in non-perfused hearts[20]. Indeed, while perfused tissue showed detectable levels of CHIKV RNA, these were significantly decreased compared to non-perfused mice (Supplementary Fig. 1a). Thus, we incorporated a perfusion step to all our experiments. To define if cardiac tissue is a direct target of CHIKV infection in immunocompetent mice, we infected mice either subcutaneously through footpad injection, intravenously via tail vein injection, or via natural transmission by mosquito bite. We quantified CHIKV infectious virus by median tissue culture infectious dose ($TCID_{50}$) in BHK-21 cells and CHIKV RNA by RT-qPCR. We were able to detect CHIKV genomes and infectious particles from perfused hearts in infected mice regardless of the inoculation route (Fig. 1a and Supplementary Fig. 1b), supporting cardiac tissue as a bona fide target of CHIKV infection. To evaluate if the CHIKV viral particles detected in the heart were actively replicating, we measured the ratio between CHIKV nsp4 and E1 transcripts by RT-qPCR as a surrogate of genomic and sub-genomic RNA, respectively, at 2, 5 and 10 dpi. As a non-replicative control, we infected mice with heat-inactivated (HI) CHIKV. While no active replication was detected for the HI control, CHIKV active replication in the heart was detected for both subcutaneous and intravenous inoculation routes (Fig. 1b). For intravenously infected mice active replication was detected at 2 dpi (R [Ratio E1/nsp4] = 3.81, p = 0.047) and 5 dpi (R = 8.11, p = 0.008), while for subcutaneously infected mice active replication was detected at 5 dpi (R = 3.88, p = 0.008). The difference in kinetics between inoculation routes is consistent with the requirement of subcutaneous inoculation to reach viremia in order to disseminate and infect cardiac tissue, while the intravenous route bypasses this stage of the infection. These results demonstrate that cardiac tissue is a direct target of CHIKV infection in immunocompetent mice.

To determine the localization of CHIKV infection within cardiac tissue, we used a reporter CHIKV that expresses the ZsGreen fluorescent protein (CHIKV-ZsGreen) exclusively under active viral replication[17]. Mice were infected with CHIKV-ZsGreen and cardiac tissue was harvested at 2 dpi. We observed multiple ZsGreen positive foci corresponding with CHIKV-infected cells for mice infected both intravenously (Fig. 1c) as well as subcutaneously (Supplementary Fig. 1c). CHIKV infects different structures of the cardiac tissue, with presence of infected cells within the myocardium, aortic valves, atrium, and vessels (Fig. 1c). With the objective of measuring the differences in the level of infection across cardiac tissue, we physically separated the heart into: (i) top section (containing atria and enriched in vessels attached to the base of the heart, such as ascending thoracic aorta) and (ii) bottom section (enriched in the myocardium); and evaluated the kinetics of CHIKV infection. To address direct infection of cardiac tissue and to bypass the variability associated with differences in dissemination, we used the intravenous infection route unless stated otherwise. We found that while CHIKV can replicate efficiently in both top and bottom sections of the heart at 2 dpi (R = 1.47; Fig.1d left panel), CHIKV RNA genomes persist for longer times in the top section containing the vessels and atria (5 dpi: R = 5.07, and 10 dpi: R = 6.35, Fig. 1d, right panel). Indeed, we found that the top section of the heart contains higher amounts of viral genomes relative to the bottom section (Fig. 1d). These results support that different structures of the cardiac tissue are not equally susceptible to CHIKV infection, and that the section containing large vessels sustain higher CHIKV viral load.

To identify the specific cell types infected by CHIKV in cardiac tissue, hearts were harvested at 2 dpi, fixed, and processed to obtain serial cryosections followed by staining with markers for specific cardiac cell types: CD31 (endothelial cells), CD45 (leukocytes), αSMA (smooth muscle cells, pericytes, and myofibroblasts), cTnT (cardiac myocytes), and vimentin. Of note, vimentin is expressed at high levels in cardiac fibroblasts[22] but can also be expressed in smooth muscle cells, endothelial cells, and macrophages[23–25]. Therefore, we identified cardiac fibroblasts cells as vimentin[+]/αSMA[−]/CD31[−]/CD45[−] populations, a combination that we orthogonally validated via single-cell RNA expression from the *Tabula Muris*[26] (Supplementary Fig. 1d). By using fluorescence microscopy and colocalization analysis with the different cell type markers, we found that the 78.5% of CHIKV-ZsGreen signal colocalized with vimentin[+]/αSMA[−] cells (Fig. 1e, f and Supplementary

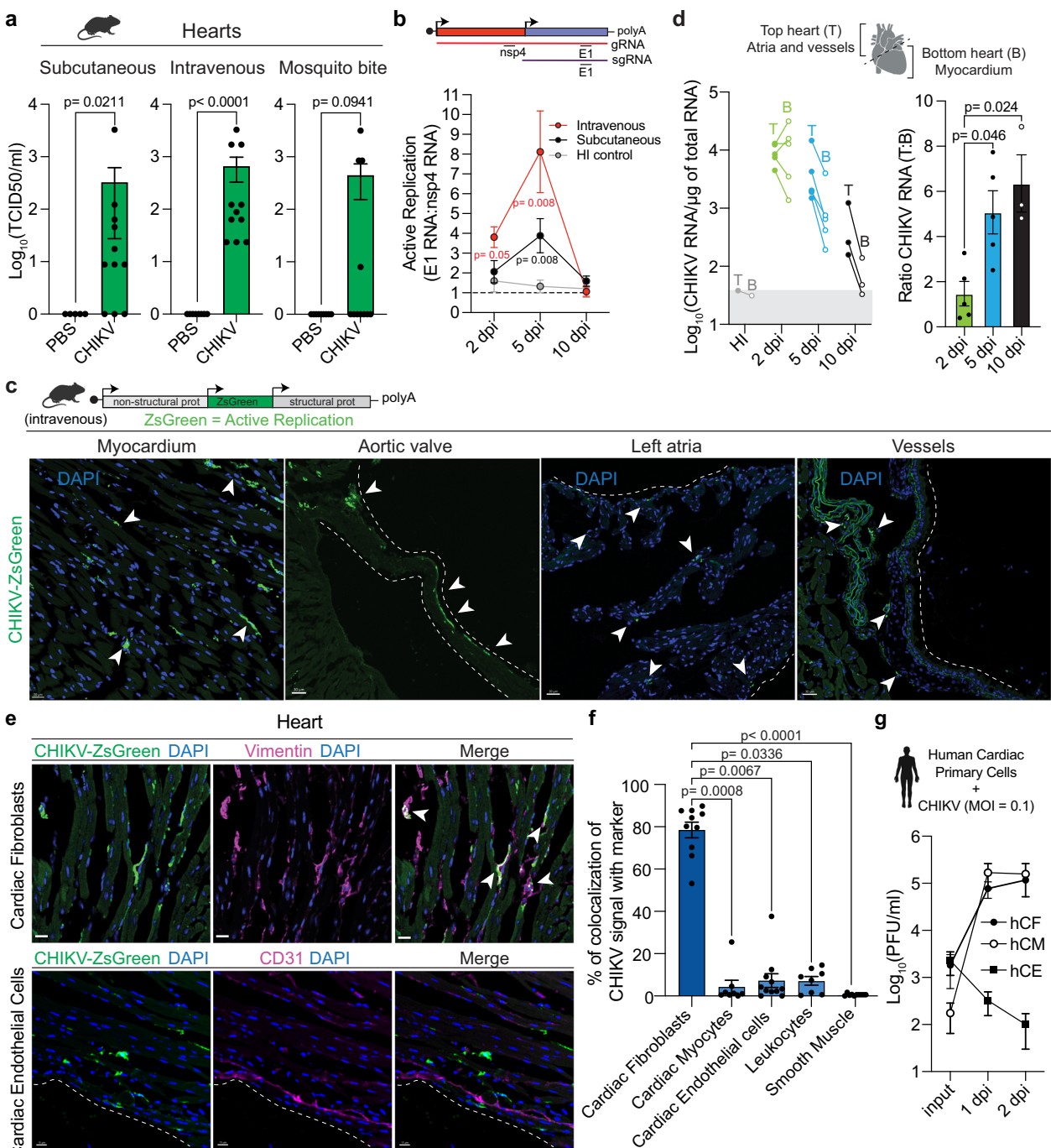

**Fig. 1 | CHIKV actively replicates in cardiac tissue and targets cardiac fibroblasts. a** Six-week-old C57BL/6 mice were inoculated with 1E5 PFU of CHIKV or mock-infected as specified in each panel. Subcutaneous: CHIKV-infected (n = 11) and mock-infected (n = 5). Intravenous: CHIKV-infected (n = 11) and mock-infected (n = 8). Mosquito bite CHIKV-infected (n = 11) and mock-infected (n = 9). **b** Ratio of E1 viral RNA to nsp4 viral RNA. Dotted line indicates the E1 RNA:nsp4 RNA ratio (R) of 1. Subcutaneous: 2 dpi (n = 4), 5 dpi (n = 8) and 10 dpi (n = 4). Intravenous: 2 dpi (n = 6), 5 dpi (n = 6) and 10 dpi (n = 4). HI-control: 2 dpi (n = 3), 5 dpi (n = 6) and 10 dpi (n = 4). **c** Representative images from 2 independent experiments showing infected myocardium (scale= 20 μm) and, aortic valve, left atria, and vessels (scale= 30 μm). White arrows: CHIKV-infected cells. **d** Mice were infected with 1E5 PFU of CHIKV or HI virus intravenously. Left panel: CHIKV RNA levels for top and bottom hearts. Right panel: ratio of viral RNA levels between top and bottom heart sections.

For 2 dpi n = 5, for 5 dpi n = 5, for 10 dpi n = 3. Gray box represents background RNA levels quantified for HI control at 5 dpi (n = 1). **e** Representative fluorescence microscopy images from 3 independent experiments of ventricular sections of CHIKV-infected hearts stained with different cardiac cell type markers. Scale= 15 μm. **f** Colocalization analysis between CHIKV-infected cells and different cell type markers. Data are represented as the percentage of the total green signal (infected cells) overlapping with the indicated markers. Colocalization analysis was done in 3-4 independent fields for n = 3 mice. **g** CHIKV multi-cycle growth curves in human primary cardiac cells. n = 2 independent experiments in technical duplicates. Data are represented as mean ± standard error of the mean (SEM, **a, b, d, f, g**). p values were calculated using two-tailed Mann-Whitney test (**a, b**), or Kruskal–Wallis and Dunn's multiple comparison test (**d, f**). Created with BioRender.com. Source data are provided as a source data file.

Fig. 1e, f). This result, together with the reduced colocalization between CHIKV-ZsGreen signal and markers for endothelial cells (CD31[+]; 7.63%), leukocytes (CD45[+]; 7.16%), smooth muscle/pericytes/myofibroblasts (αSMA[+]; 0.38%), and cardiac myocytes (cTnT[+]; 4.39%; Fig. 1f; Supplementary Fig. 1f) demonstrate that the cardiac fibroblasts are the primary target of CHIKV infection of cardiac tissue in vivo.

Finally, we sought to determine whether human cardiac cells show similar susceptibilities to CHIKV infection than those observed in mice. For that purpose, we evaluated the capacity of CHIKV to infect human primary cardiac cells in vitro. We infected human primary cardiac fibroblasts (hCF), human cardiac myocytes (hCM) and human cardiac microvasculature endothelial cells (hCE) at an MOI of 0.1 and measured the production of infectious particles. We found that CHIKV can efficiently infect hCF (Fig. 1g, black circles), but no hCE (Fig. 1g, squares), supporting our in vivo observations in mice. Interestingly, hCM in culture produced CHIKV infectious particles at similar levels than the observed for hCFs (Fig. 1g, open circles). However, careful interpretation of this result must be taken, since hCMs in culture lose specific features that can potentially impact susceptibility to CHIKV infection. For example, while cardiac myocytes in tissue are characterized for being multinucleated and quiescent in vivo, the in vitro culture revert these phenotypes, resulting in mono-nucleated and proliferating cells[27].

Altogether, our results demonstrate that CHIKV actively replicates in cardiac tissue with cardiac fibroblasts being the primary cellular target of CHIKV infection in immunocompetent mice.

## Cardiac tissue clears CHIKV infection without inducing cardiac damage in immunocompetent mice

To evaluate the consequences of CHIKV replication in cardiac tissue, we measured the kinetics of CHIKV infection in the heart, as well as cardiac damage, apoptosis and inflammation. Significant levels of CHIKV RNA were detected in the heart of intravenously infected mice as early as 12 h post infection (hpi) with a peak at 2 dpi, and the infection was cleared out to background detection levels by 9 dpi (Fig. 2a, left panel). Infectious particles follow a similar trend, with a peak of infectious particle production between 1 and 2 dpi, and no detectable levels by 9 dpi (Fig. 2a, right panel). Interestingly, albeit with higher CHIKV RNA levels, the viral replication kinetics follow a similar temporal pattern when measured in calf muscle, a primary organ of CHIKV replication (Supplementary Fig. 2a). Therefore, these results demonstrate that CHIKV infection is efficiently cleared of cardiac tissue in immunocompetent mice.

Elevated cardiac troponins have been observed in individuals with CHIKV-associated cardiac manifestations[6,7,13,16]. To define whether CHIKV can lead to cardiac damage in mice, we evaluated the serum levels of cardiac troponin-T (cTnT) at 2 and 5 dpi by ELISA. We observed no significant differences in levels of cardiac troponin-T in serum between infected and PBS or HI controls (Fig. 2b), suggesting that CHIKV replication does not induce damage to cardiac myocytes. Indeed, no sign of cardiac tissue apoptosis was observed at 3 or 5 dpi (Fig. 2c, positive staining control in Supplementary Fig. 2b). Thus, no cardiac damage is observed in WT mice upon CHIKV infection and clearance.

Although no cardiac damage was observed upon CHIKV in WT mice, we found that CHIKV infection stimulates the protein production of proinflammatory cytokine and chemokine in cardiac tissue (CCL2, CCL5, CCL3, TNFα, IL6, IFNγ, IL2, KC, IL1α, IL12p70, and IL12p40) at 2 dpi. However, IL12p40 is the only cytokine that remains upregulated at 5 dpi (Fig. 2d; Supplementary Fig. 2c, d), indicating that CHIKV infection is inducing a transient inflammatory state in cardiac tissue. Histopathological analysis of hematoxylin-eosin (H&E) staining of four-chamber view sections of infected and mock-infected at 3, 5, 10, and 15 dpi (Supplementary Data 1) revealed no visible mononuclear cell infiltrates in ventricles, septum, or atria of CHIKV-infected hearts.

Altogether, these results demonstrate that cardiac tissue can efficiently clear CHIKV infection without significant cardiac damage in immunocompetent mice.

## Local IFN-I response in cardiac tissue is essential to control CHIKV infection and prevent tissue damage

To understand how the heart rapidly clears CHIKV infection without cardiac tissue damage in WT mice, we measured the transcriptional response of CHIKV-infected heart homogenates by bulk RNA-seq. We used CHIKV-infected mice at 2 dpi (where we detect both CHIKV RNA and CHIKV infectious particles) or at 5 dpi (where we detect little to no CHIKV infectious particles, but CHIKV RNA levels are detected; Fig. 2a). To identify changes derived only from active CHIKV replication, we used the HI virus as a control. Principal component analysis (PCA) showed that biological replicates grouped by treatment (Supplementary Fig. 3a). A total of 377 upregulated genes and 84 downregulated genes were detected at 2 dpi, and 259 upregulated genes as well as 52 downregulated genes at 5 dpi (Fig. 2e, Supplementary Fig. 3b, c; Supplementary Data 2 and 3). The top-10 upregulated genes on both days include genes associated with protection of cardiac tissue damage and mitochondria metabolism (*Apod*, *Atf5*, *Mthfd2*)[28,29], IFN-I response (*Ifi27*, *Ddx60*, *Serping1*), cytokine signaling (*H2-K1*), and complement cascade (*C4a*, *C4b*, *C1ra*) (Fig. 2e). In line with these observations, GSEA pathway enrichment analysis for Reactome and Hallmark datasets show upregulation (FDR < 0.15 and normalized enrichment score [NES] > 0) of several pathways associated with innate immunity, adaptive immunity, IFN signaling (Fig. 2f, Supplementary Fig. 3c, and Supplementary Data 2 and 3, and Supplementary Tables 2 and 3). Downregulated pathways (FDR < 0.15 and NES < 0) in infected heart homogenates at 5 dpi include metabolic pathways such as metabolism of amino acids, respiratory electron transport, and oxidative phosphorylation, among others (Supplementary Fig. 3c).

The induction of IFN-I signaling pathways in non-hematopoietic cells is essential to control CHIKV infection in mammals[30,31]. We found that the IFN-I pathway is significantly induced in CHIKV-infected hearts at both 2 and 5 dpi (Fig. 3a, and Supplementary Fig. 3e). Particularly, we observed upregulation of interferon stimulated genes (ISGs; e.g. *Isg15*, *Ifitm3*, *Mx1*, *Mx2*, *Ifit1*, *Ifit2*, *Ifit3*, *Ifi27*, *Ddx58*, and *Ddx60*, among others) as well as genes involved in the IFN-I signaling pathway (e.g. *Irf7*, *Stat1*, *Stat2*; Supplementary Data 2 and 3). While the levels of serum IFNα and IFNβ were elevated at 24 hpi, no changes in local levels of IFNα and IFNβ were observed in infected hearts at 12 or 24 hpi (Supplementary Fig. 3f). This suggests that circulating IFN-I is a contributor to the local IFN-I response elicited by cardiac tissue upon CHIKV infection.

To gain insight into whether the IFN-I response observed by RNA-seq was locally mediated by infected cardiac fibroblasts, we measured the local protein production of ISG15 and IFITM3, both ISGs known to restrict CHIKV infection[32,33], and found that both were upregulated at 2 dpi in CHIKV-infected hearts (Fig. 3a, and Supplementary Fig. 3d). While IFITM3 is expressed at basal levels in the absence of interferon induction, its expression is highly induced by either IFN-I or interferon-γ[34]. We found that that 95% of CHIKV-infected cells are positive for the IFITM3 marker. However, in the absence of the IFN-I receptor (*Ifnar1*[-/-] mice), only 24.6% of CHIKV-infected cells colocalize with the IFITM3 signal (Supplementary Fig. 4a), demonstrating that the observed response is partially dependent on IFN-I. Local production of ISG15 and IFITM3 proteins were detected in cardiac tissue upon CHIKV infection in both infected and non-infected vimentin[+] cells (Fig. 3b and Supplementary Fig. 4a), suggesting that cardiac fibroblast among other vimentin[+] cells are the major IFN-I responders in infected cardiac tissue.

Next, we sought to determine the importance of IFN-I signaling in the infection of human cardiac fibroblasts (hCFs) by CHIKV. We evaluated this by: (i) measuring the capacity of hCFs to generate a robust IFN-I response after poly(I:C) stimulation and CHIKV infection; (ii)

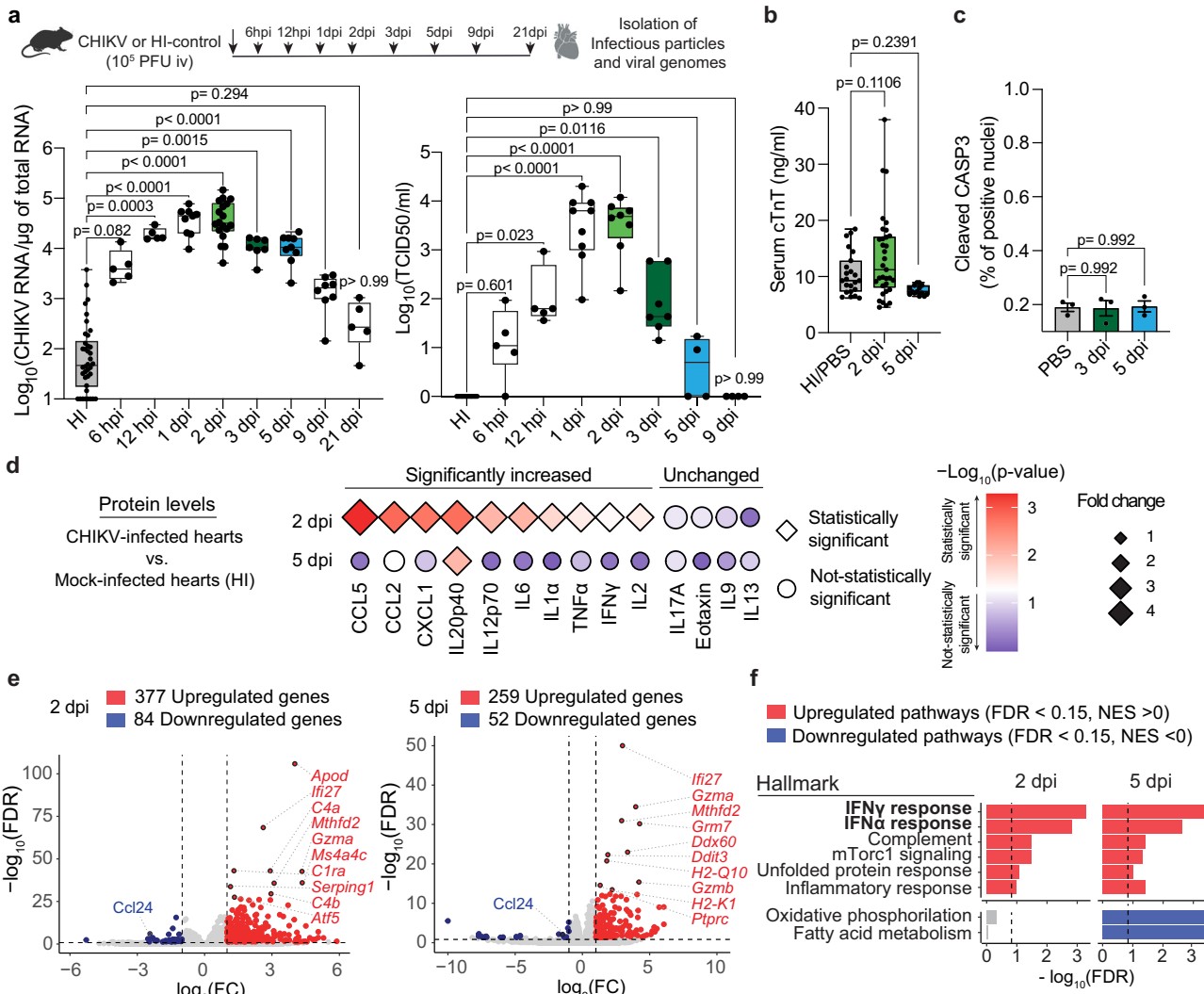

**Fig. 2 | Cardiac tissue clears CHIKV infection without inducing cardiac damage in immunocompetent mice. a** Upper panel: Schematic representation of the experimental design. Bottom panel: CHIKV viral genomes or CHIKV infectious particles from heart tissue homogenates determined by RT-qPCR or $TCID_{50}$, respectively. CHIKV viral genomes: HI-control (n = 39), 6 hpi (n = 5), 12 hpi (n = 5), 1 dpi (n = 9), 2 dpi (n = 18), 3 dpi (n = 7), 5 dpi (n = 13), 9 dpi (n = 8) and 21 dpi (n = 8). Infectious particles, HI-control (n = 18), 6 hpi (n = 5), 12 hpi (n = 5), 1 dpi (n = 9), 2 dpi (n = 8), 3 dpi (n = 7), 5 dpi (n = 4), and 9 dpi (n = 4). **b** Serum levels of cardiac troponin-T measured by ELISA: mock-infected (n = 23), 2 dpi (n = 33), and 5 dpi (n = 14). **c** Quantifications of cleaved CASP3 nuclei versus total nuclei of stained whole ventricular sections (see "Methods"). n = 3/group. **d** Plot showing the relative protein levels of proinflammatory cytokines and chemokines between CHIKV-versus mock-infected heart homogenates at 2 and 5 dpi. Relative expression is represented in log2 fold change (diamond size). Statistical significance is expressed as a color scale (-$Log_{10}$ adjusted p-value). 2 dpi: HI-control and CHIKV-infected hearts (n = 6/group). 5 dpi: HI-control (n = 4) and CHIKV-infected hearts (n = 5). **e** Volcano plots from differential expression analysis at 2 dpi (left panel) and 5 dpi (right panel). Red and blue indicate upregulated (FDR < 0.15 and $log_2$(FC) > 1) and downregulated genes (FDR < 0.15 and $log_2$(FC) < −1), respectively. Top-10 differentially expressed genes are indicated. n = 4 mice/group. **f** GSEA pathway enrichment analysis for Hallmark datasets showing top upregulated and downregulated pathways at 2 dpi and 5 dpi. n = 4 mice/group. Boxplots show median and quartile ranges, whiskers represent the range (**a, b**). Data is represented as mean ± SEM (**c**). p values were calculated using Kruskal−Wallis and Dunn's multiple comparison test (**a**), and one-way ANOVA with Dunnett's multiple comparison (**b, c**). Created with BioRender.com. Source data are provided as a source data file.

measuring the susceptibility of these cells to CHIKV infection in the absence of IFN-I signaling using the JAK1/JAK2 inhibitor ruxolitinib (Rux); and (iii) inducing a refractory state upon increasing doses of IFN-I pre-treatments. Indeed, we found that hCFs efficiently respond to poly(I:C) stimulation, resulting in ~88% of the monolayer expressing ISG15 (Fig. 3c, left panel). In agreement with our in vivo data, we found that infected hCFs responded to CHIKV infection by producing significant levels of ISG15 (Fig. 3c, left panel). In addition, Rux treatment significantly increased the percentage of CHIKV-infected cells (Fig. 3c, right panel), and correlated with a reduction of ISG15 production (Fig. 3c, left panel). Furthermore, we found that exogenous addition of either IFNα or IFNβ prevented CHIKV infection in a dose response manner, with IFNβ showing increased blocking capacity (Fig. 3d).

These results support that IFN-I signaling is critical in controlling CHIKV infection in human cardiac fibroblasts.

Finally, we evaluated the direct contribution of the IFN-I response in controlling CHIKV-heart infection and preventing tissue damage by using *Ifnar1*$^{-/-}$ mice. As expected, we observed that in the absence of IFN-I signaling, CHIKV spreading and infectious particle production were increased in cardiac tissue of *Ifnar1*$^{-/-}$ mice compared to immunocompetent WT mice (Fig. 3e, and Supplementary Fig. 4b). This increase in infectious particle production was also observed in the calf muscle, a primary site of CHIKV infection, but not observed in a non-CHIKV-target organ such as the pancreas (Supplementary Fig. 4b). While cardiac fibroblasts (Vimentin$^{+}$/aSMA$^{-}$; 42%) still represent the primary target of CHIKV infection in cardiac tissue in *Ifnar1*$^{-/-}$ mice, we

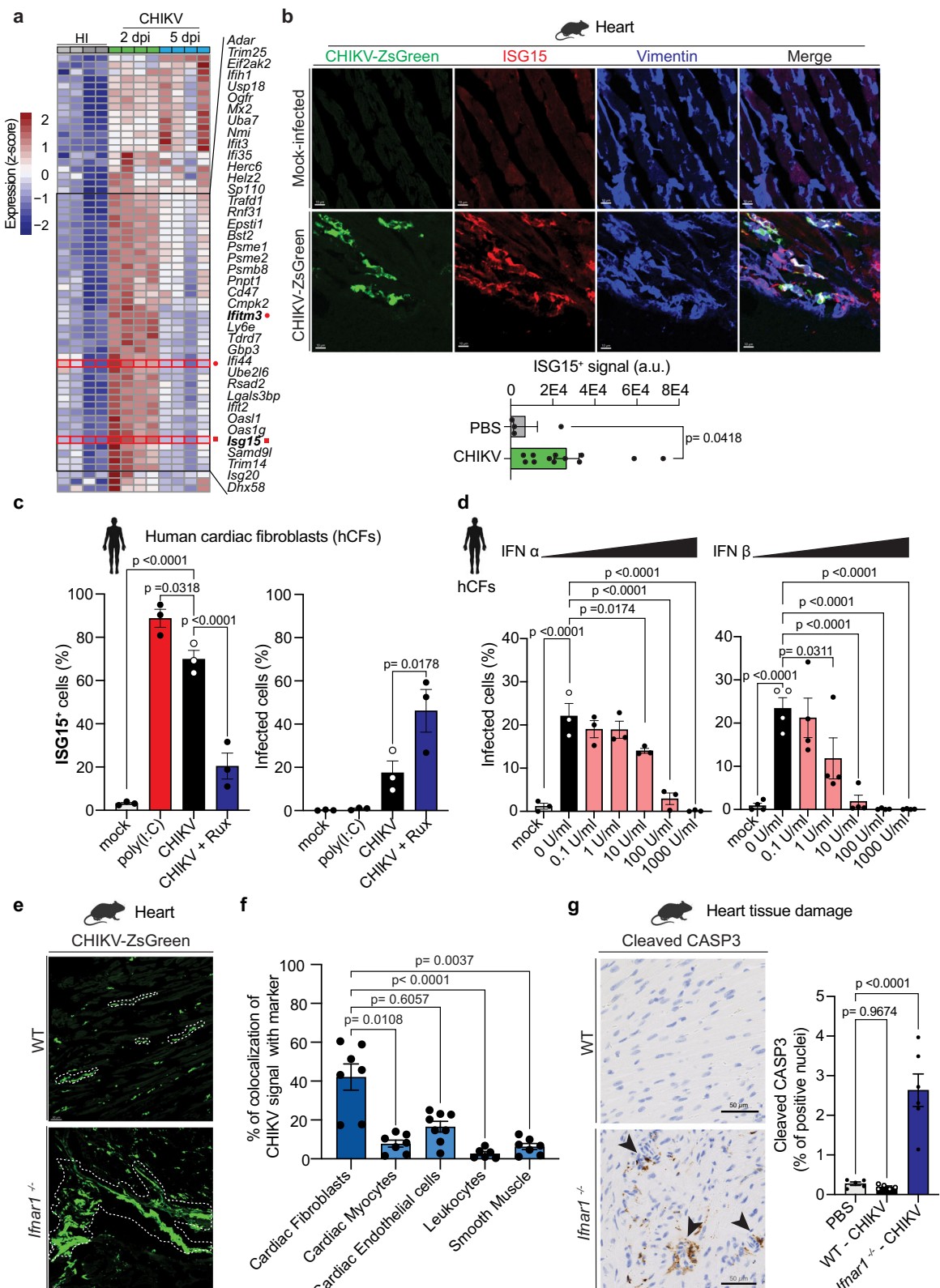

observed an increased proportion of endothelial infected cells (CD31[+]; 16.5%, Fig. 3f, and Supplementary Fig. 3c) relative to WT (Fig. 1f). Intriguingly, in *Ifnar1*[-/-] mice only 7.8% of the infected cells colocalized with cTnT marker (Fig. 3f, and Supplementary Fig. 3c), suggesting that even in the absence of IFN-I response, cardiac myocytes do not represent a target of CHIKV infection in mice. To determine whether the observed increased infection in *Ifnar1*[-/-] mice results in cardiac

damage even in the absence of cardiac myocyte infection, we assessed cardiac tissue apoptosis by cleaved CASP3 staining at 1 dpi. We observed that while infection of WT mice showed similar levels of cleaved CASP3 signal in comparison to mock-infected control, *Ifnar1*[-/-] mice showed a significantly increased level of cleaved CASP3 within the myocardium (Fig. 3g), demonstrating that unrestricted CHIKV infection can lead to cardiac tissue damage. Altogether, these results

**Fig. 3 | Local IFN-I response in cardiac tissue control CHIKV infection and prevent tissue damage. a** Heat map showing the differentially expressed genes (FDR < 0.15) from the Hallmark IFNα response pathway for CHIKV-infected and HI-control hearts. **b** Top panel: representative fluorescence microscopy images from two independent repetitions showing ventricular sections of CHIKV-infected hearts and PBS control stained with anti-ISG15 and anti-vimentin antibodies. Scale= 10 μm. Bottom panel: Quantification of ISG15⁺ signal/field. Analysis was done on two independent fields for PBS control (n = 2), and CHIKV-infected (n = 6). **c** Dependence on IFN-I signaling in human primary cardiac fibroblasts during CHIKV infection. Cells were pretreated as indicated in the figure and infected with CHIKV-ZsGreen at an MOI of 0.25. Data represent three independent repetitions in technical duplicates. **d** IFN-I pretreatment experiments in human primary cardiac cell types. Cells were pretreated with different concentrations of recombinant IFN-α (right panel) and IFN-β (left panel) for 24 h and then infected with CHIKV-ZsGreen at an MOI of 0.25 for 24 h. For IFN-α: three independent repetitions in technical

duplicates. For IFN-β: four independent repetitions in technical duplicates. **e** Representative fluorescence microscopy images from three independent repetitions showing CHIKV-ZsGreen WT or *Ifnar1⁻/⁻* infected hearts at 1 dpi. Scale= 30 μm. **f** Percentage of the total of infected cells (green signal) overlapping with the indicated makers (See Supplementary Fig. 1d, e). Colocalization analysis was done in three to five independent sections for n = 2 mice. **g** Left panel: Apoptosis staining was measured by IHC against cleaved CASP3. Scale= 50 μm. Arrows highlight apoptotic foci. Right panel: Quantifications of cleaved CASP3 nuclei versus total nuclei were performed in whole ventricular sections (LV, RV, and septum) using Visiopharm®. Each dot represents one individual mouse. PBS control (n = 5), CHIKV-infected WT (n = 5), and CHIKV-infected *Ifnar1⁻/⁻* mice (n = 6). Data are represented as mean ± SEM (**b–d, f, g**). P values were calculated using two-tailed Mann–Whitney (**b**), or one-way ANOVA with Dunnett's multiple comparison test (**c, d, g**), or Kruskal–Wallis with Dunn's multiple comparison test (**f**). Created with BioRender.com. Source data are provided as a source data file.

demonstrate that a local IFN-I response is essential to control CHIKV infection in the heart and prevent cardiac tissue damage.

## MAVS signaling is required for viral clearance in CHIKV-infected hearts

IFN-I signaling is triggered upon recognition of CHIKV RNA via either cytosolic (RIG-I/MDA5) or endosomal receptors (TLR3/7)[35]. To gain insight into the mechanisms of how cardiac tissue clears CHIKV infection, we took advantage of mice deficient in major innate immune signaling components including MYD88, TRIF, or MAVS. In addition, to address for the contribution of the NLRP3 inflammasome in CHIKV clearance[36,37] we used CASPASE 1/11 deficient mice. We infected each mouse intravenously with CHIKV and evaluated viral burden in cardiac tissue at 2, 5, and 7 dpi. Interestingly, while we found no significant differences in the infection kinetics between *Myd88⁻/⁻*, *Trif Lps2/Lps2* or *Casp1/11⁻/⁻* mice compared to WT mice, infectious particles were detectable up to 7 dpi for *Mavs⁻/⁻* mice (Fig. 4a). Indeed, we found that the heterozygote deletion of MAVS results in a phenotype similar to WT, demonstrating that the complete loss of MAVS is required for persistence CHIKV infection (Supplementary Fig. 5a). These results suggested that MAVS signaling is contributing to the clearance of CHIKV infection from infected hearts.

Next, we expanded our analysis in the heart up to 15 dpi and we found that infectious particles can be detected up to 10 dpi (Fig. 4b), while infectious particles in serum were cleared by 5 dpi for both *Mavs⁻/⁻* and WT mice (Fig. 4c). In line with this, CHIKV-ZsGreen infected cells were detected up to 10 dpi in both the myocardial region as well as in vessels in *Mavs⁻/⁻* mice (Supplementary Fig. 5b, c). To evaluate whether this deficiency in viral clearance from cardiac tissue is observed in other tissues we evaluated the levels of CHIKV RNA in calf muscle and heart at 2, 10, and 15 dpi for both WT and *Mavs⁻/⁻* mice. We found that the kinetics of CHIKV clearance from the calf muscle is different from cardiac tissue. For cardiac tissue, we observed a progressive clearance in the RNA levels of WT mice, while for *Mavs⁻/⁻* mice we observed that the levels of viral RNA are maintained up to 10 dpi, with decreased levels observed by 15 dpi (Fig. 4d). However, in calf muscle, both *Mavs⁻/⁻* and WT mice showed a progressive reduction of viral RNA over time, and at 10 dpi no significant differences in CHIKV RNA were observed (Fig. 4e). Interestingly, when we evaluated the CHIKV RNA levels in the top section or bottom section of the heart at 10 dpi, we observed higher viral RNA levels in top sections relative to the bottom sections (Fig. 4f). These results are in agreement with our observation in WT animals (Fig. 1d), and supports that heart sections enriched in atria and/or vessels are more prone to sustained infection over time. Collectively, these results demonstrated that sensing through MAVS is required for clearance of CHIKV infection from cardiac tissue.

Of note, we found that *Mavs⁻/⁻* infected hearts induced similar levels of *Isg15* mRNA compared to WT (Fig. 4g). Prompted by this

result, we measured the local (cardiac tissue homogenate) and systemic (serum) production of IFNα or IFNβ at 24 hpi, which corresponds to the peak of induction of *Isg15* in cardiac tissue (Fig. 4g). We found that while the local IFN-I production is low in cardiac tissue (Fig. 4h), both WT and *Mavs⁻/⁻* mice produce significant systemic levels of both IFNα and IFNβ (Fig. 4i). Surprisingly, the systemic levels of both IFNα and IFNβ were higher in *Mavs⁻/⁻* mice compared to WT (Fig. 4i), which correlates with the significantly higher viral loads observed in *Mavs⁻/⁻* (Fig. 4b and 4d, e). These data suggest that the delay in CHIKV clearance cannot be entirely explained by a reduced IFN-I induction associated with *Mavs⁻/⁻*.

## *Mavs⁻/⁻* mice develop focal myocarditis and large vessel vasculitis upon CHIKV infection

Next, we sought to determine whether the persistence of CHIKV-heart infection in *Mavs⁻/⁻* mice can lead to cardiac tissue damage. We infected *Mavs⁻/⁻*, *Mavs⁺/⁻*, and WT mice intravenously with CHIKV and harvested the heart for histopathological analysis at 10 dpi (Fig.5a, upper panel). As expected, we didn't observe any signs of inflammation in the myocardial tissue for neither *Mavs⁺/⁻* nor WT mice (Fig.5a, b and Supplementary Data 1). However, 53% of CHIKV-infected *Mavs⁻/⁻* mice (7 out of 13 mice) developed focal myocarditis at 10 dpi (Fig.5a-b and Supplementary Data 1). Strikingly, we found that all of the CHIKV-infected *Mavs⁻/⁻* mice (13 out of 13) had transmural inflammation of the vessel wall in the large vessels attached to the base of the heart (e.g., aorta, pulmonary artery) corresponding to large vessel vasculitis (Fig. 5a, c, and Supplementary Data 1). Of note, a fraction of WT and the *Mavs⁺/⁻* mice showed minimal signs of inflammation in the vessels corresponding to endothelialitis, characterized by mild inflammation of the intima layer of the vessel (Fig. 5a, bottom panel). Altogether, these results demonstrate that the persistence of CHIKV infection in cardiac tissue can lead to inflammatory infiltrates leading to focal myocarditis and large vessel vasculitis.

## Chronic inflammation in myocardium and vasculature in CHIKV-infected *Mavs⁻/⁻* mice persist for several months

Given the focal myocarditis and the large vessel vasculitis observed at 10 dpi in *Mavs⁻/⁻* mice, we asked whether this infiltration can result in chronic inflammation potentially leading to long-term tissue damage. For that purpose, *Mavs⁻/⁻* mice were infected intravenously or subcutaneously with CHIKV, and cardiac tissue was collected for histopathological analysis at 2 weeks, 1 month, or 2 months post infection. Tissue was stained with H&E, and mononuclear cell infiltrates were evaluated for CD3 and CD11b markers by IHC. We found that a fraction of *Mavs⁻/⁻* mice developed focal and multifocal myocarditis accentuated around the vessels in mice infected by either intravenous or subcutaneous inoculation routes (Fig. 6a, b and Supplementary Data 1). These inflammatory foci were characterized by CD11b⁺, and CD3⁺ cell infiltrates and persisted for up to 60 dpi or up to 31 dpi in

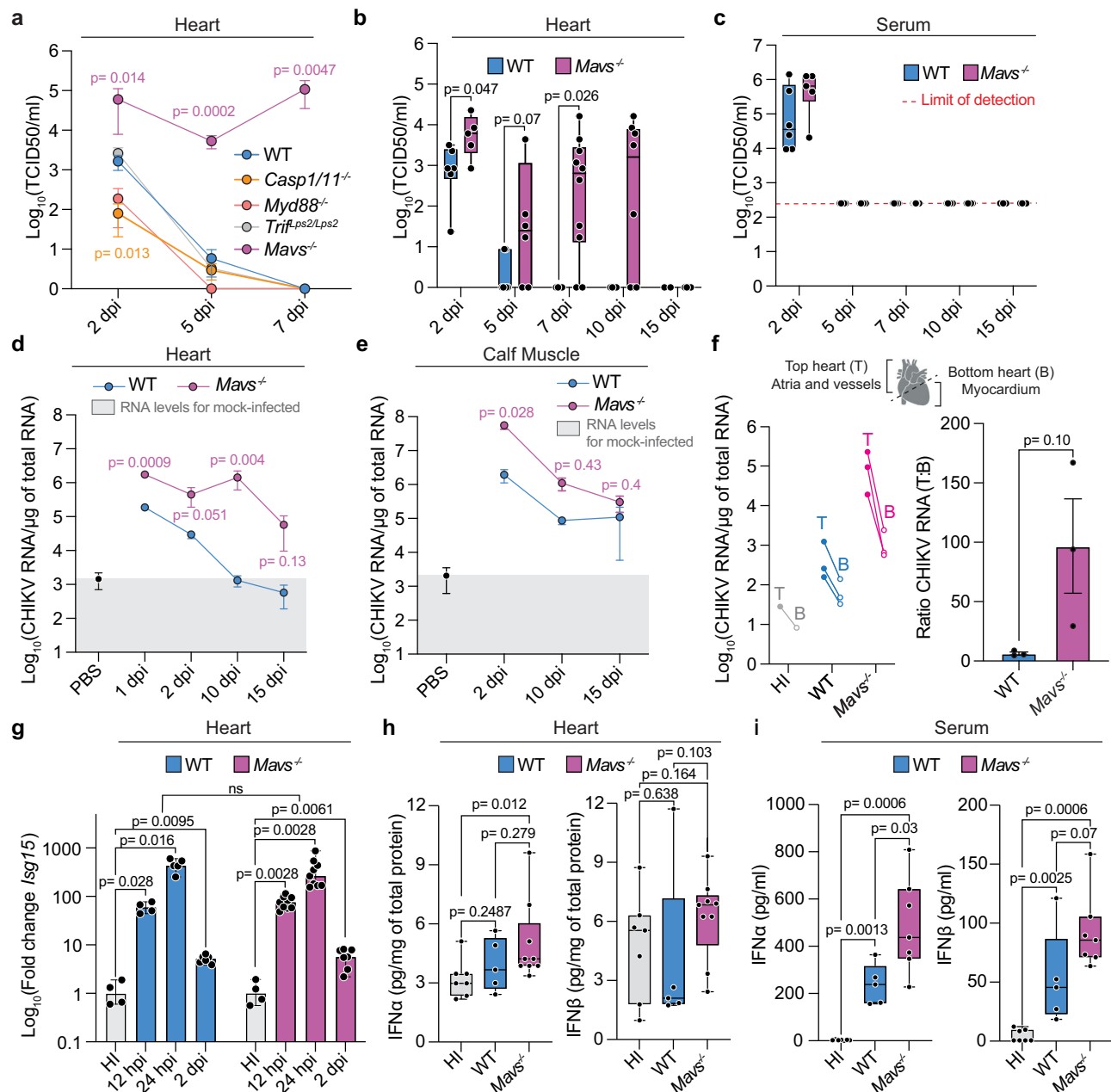

**Fig. 4 | MAVS signaling is required for clearance in CHIKV-infected hearts.**
**a** Differential susceptibilities to CHIKV-heart infection between *MyD88⁻/⁻*, *Trif^Lps2/Lps2*, *Casp1/11⁻/⁻*, *Mavs⁻/⁻*, and WT mice. WT: 2 dpi (n = 10), 5 dpi (n = 16), and 7 dpi (n = 6). *Casp1/11⁻/⁻*: 2 dpi (n = 5), 5 dpi (n = 8), and 7 dpi (n = 3). *MyD88⁻/⁻*: 2 dpi (n = 4), 5 dpi (n = 6), and 7 dpi (n = 3). *Trif^Lps2/Lps2*: 2 dpi (n = 5), 5 dpi (n = 5), and 7 dpi (n = 4). *Mavs⁻/⁻*: 2 dpi (n = 4), 5 dpi (n = 4), and 7 dpi (n = 4). **b**, **c** CHIKV-heart infection kinetics in *Mavs⁻/⁻* and WT mice. **b** WT: 2 dpi (n = 6), 5 dpi (n = 7), 7 dpi (n = 4), 10 dpi (n = 4), and 15 dpi (n = 2). *Mavs⁻/⁻*: 2 dpi (n = 5), 5 dpi (n = 6), 7 dpi (n = 10), 10 dpi (n = 8), and 15 dpi (n = 4). **c** WT: 2 dpi (n = 6), 5 dpi (n = 7), 7 dpi (n = 4), 10 dpi (n = 4), and 15 dpi (n = 2). *Mavs⁻/⁻*: 2 dpi (n = 5), 5 dpi (n = 6), 7 dpi (n = 8), 10 dpi (n = 5), and 15 dpi (n = 4). CHIKV RNA genomes from heart (**d**) and from calf muscle homogenates (**e**). **d** HI-control (n = 6). WT: 1 dpi (n = 5), 2 dpi (n = 6), 10 dpi (n = 4), and 15 dpi (n = 2). *Mavs⁻/⁻*: 1 dpi (n = 9), 2 dpi (n = 5), 10 dpi (n = 8), and 15 dpi (n = 4). **e** HI-control (n = 3). WT: 2 dpi (n = 4), 10 dpi (n = 5), and 15 dpi (n = 3). *Mavs⁻/⁻*: 2 dpi (n = 4), 10 dpi (n = 8), and 15 dpi (n = 4). **f** Left panel: CHIKV RNA levels in top and bottom heart sections at 10 dpi. HI-control (n = 1), WT (n = 3), and *Mavs⁻/⁻* (n = 3). Right panel: ratio of viral RNA levels between top and bottom heart sections. N = 3/group. WT dataset at 10 dpi is shared with Fig. 1c. **g** *Isg15* expression levels in *Mavs⁻/⁻* and WT CHIKV-infected hearts. WT: HI-control (n = 4), 12 hpi (n = 4), 24 hpi (n = 5), and 2 dpi (n = 6). *Mavs⁻/⁻*: HI-control (n = 4), 12 hpi (n = 9), 24 hpi (n = 9), and 2 dpi (n = 7). Protein levels of IFN-α and INF-β in heart (**h**) and serum (**i**) at 24 hpi. **h**. HI-control (n = 7), WT (n = 5), and *Mavs⁻/⁻* (n = 9). **i**. HI-control (n = 7), WT (n = 5), and *Mavs⁻/⁻* (n = 7). Data shown as mean ± SEM (**a**, **d-g**). Boxplots show median and quartile ranges, whiskers represent the range (**b**, **c**, **h**, **i**). P values were calculated using two-tailed Mann-Whitney test (**a-i**). Created with BioRender.com. Source data are provided as a source data file.

mice infected intravenously or subcutaneously, respectively (Fig. 6b, and Supplementary Fig. 6a, b, e). Interestingly, we did not detect changes in the serum levels of cardiac Troponin T (Fig. 6c), suggesting that despite myocarditis there is no significant cardiac myocyte damage in this model. Strikingly, large vessel vasculitis characterized by transmural CD3⁺ and CD11b⁺ infiltrates was detected up to 60 dpi in

infected *Mavs⁻/⁻* mice independently of the inoculation route (Fig. 6d; Supplementary Fig. 6c, d, and Supplementary Data 1).

Of note, cardiac tissue of an infected *Mavs⁻/⁻* mice that succumbed to CHIKV infection at 15 dpi (Supplementary Table 1), showed substantial tissue damage featured by patchy myocyte dystrophic calcification affecting single myocytes, and focal myocarditis composed by

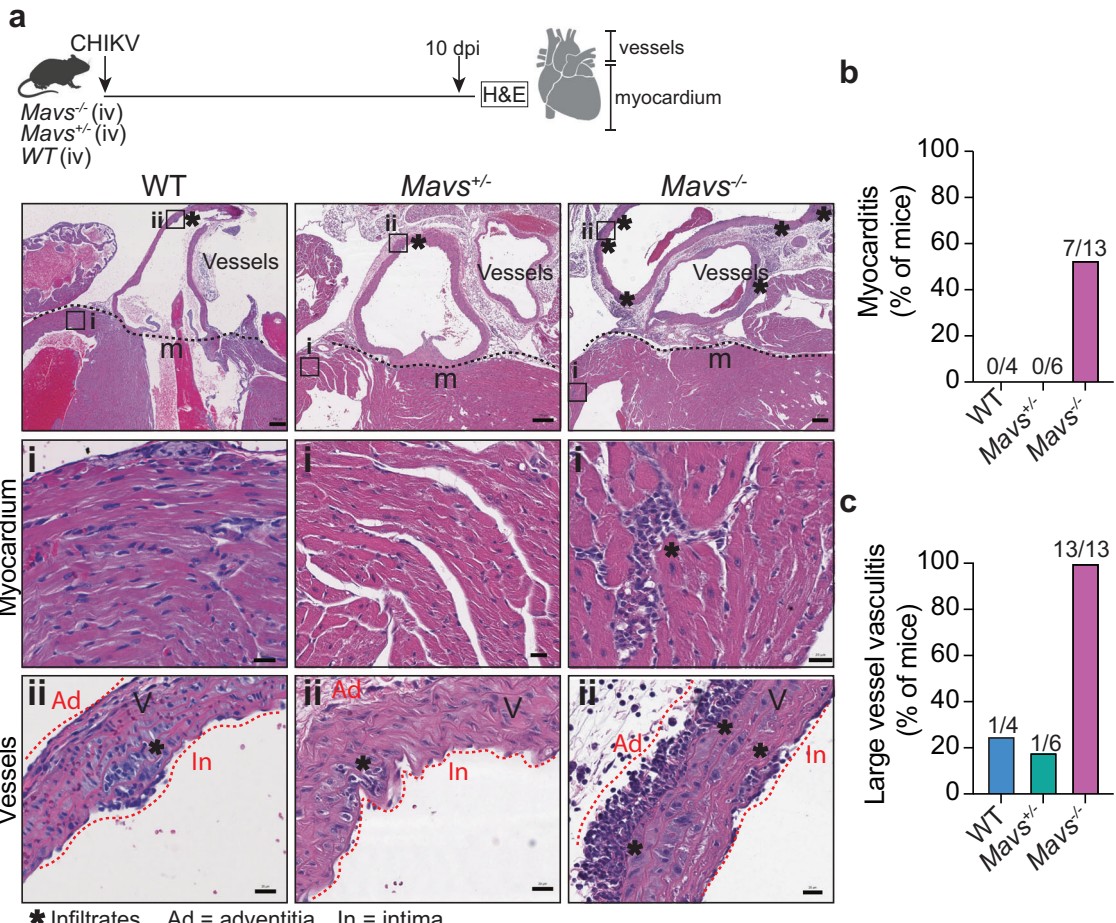

**Fig. 5 | *Mavs⁻/⁻* mice develop myocarditis and large vessel vasculitis upon CHIKV infection. a** Upper panel: Schematic representation of the experimental design. Bottom panel: Representative H&E from at least four independent mice showing CHIKV-infected WT, *Mavs⁺/⁻* and *Mavs⁻/⁻* hearts at 10 dpi. Scale = 200 µm. The black square indicates the section selected (inset i and ii). Inset i: myocardial region. Inset ii: large vessels attached to the base of the heart. Scale = 20 µm. **b** Quantification of the number of mice with positive signs of myocarditis. Data represent the percentage of mice with any sign of myocarditis over the total amount of mice analyzed. WT (n = 4), *Mavs⁺/⁻* (n = 6), and *Mavs⁻/⁻* (n = 13). **c** Quantification of the number of mice with large vessel vasculitis. Data represent the percentage of mice with any sign of vasculitis over the total amount of mice analyzed. WT (n = 4), *Mavs⁺/⁻* (n = 6), and *Mavs⁻/⁻* (n = 13). Created with BioRender.com. Source data are provided as a source data file (Related to Supplementary Data 1).

inflammatory infiltrates of CD11b⁺ and CD3⁺ cells, with no signs of fibrosis (Supplementary Fig. 7). Altogether, these results demonstrate that persistence of CHIKV infection in cardiac tissue leads to chronic inflammation characterized by focal or multifocal myocarditis and large vessel vasculitis.

## Discussion

In this study, we defined mechanisms implicated in CHIKV-cardiac tissue infection and identified host factors involved in the development of cardiac tissue damage. Here, we demonstrated: (i) that cardiac fibroblasts are a direct target of CHIKV infection in an immunocompetent host; (ii) that cardiac tissue elicits a local IFN-I response required for CHIKV clearance and preventing tissue damage; (iii) that in the absence of MAVS signaling CHIKV persists in infected hearts, and (iv) that failure to clear CHIKV in *Mavs⁻/⁻* mice leads to focal myocarditis and chronic large vessel vasculitis detectable up to 60 dpi.

Previous studies, including our own, reported the presence of infectious particles or viral RNA in non-perfused cardiac tissue of CHIKV-infected mice[19,20]. However, due to the lack of perfusion and high levels of CHIKV in circulation, whether cardiac tissue is a direct target of CHIKV infection remained elusive. Here, by including a perfusion step, and by using orthogonal experimental approaches, we demonstrated that CHIKV actively replicates in the cardiac tissue of immunocompetent mice regardless of the infection route. Moreover, we showed that CHIKV infects cardiac fibroblasts and that its infection follows a discrete and focalized pattern within ventricles, atria, valves, and the perivascular region. We found that primary human cardiac fibroblasts are susceptible and permissive to CHIKV. Indeed, CHIKV antigen in cardiac fibroblasts was previously identified in human necropsies of individuals who died from CHIKV[12]. Therefore, our results in murine models are consistent with the tropism observed in human necropsies.

We demonstrated that in WT mice, CHIKV is cleared from cardiac tissue within the first week of infection without causing tissue damage. Our RNA-seq data revealed that infected cardiac tissue shows upregulation of innate immunity, adaptive immunity, and IFN-I signaling pathways, similar to what has been described for footpad and lymph node CHIKV-infected tissue[38]. In addition, we found that genes previously associated with protection of the cardiac tissue and antioxidant metabolism (*Apod, Aft5, Fgf21, and Mthfd2*) were upregulated at 2 dpi[39,40]. *Apod* is a cardioprotective gene whose expression is induced in mouse hearts during myocardial infarction and has been proposed to protect the heart against ischemic damage[28]. In addition, *Apod* has been reported to be the most significantly

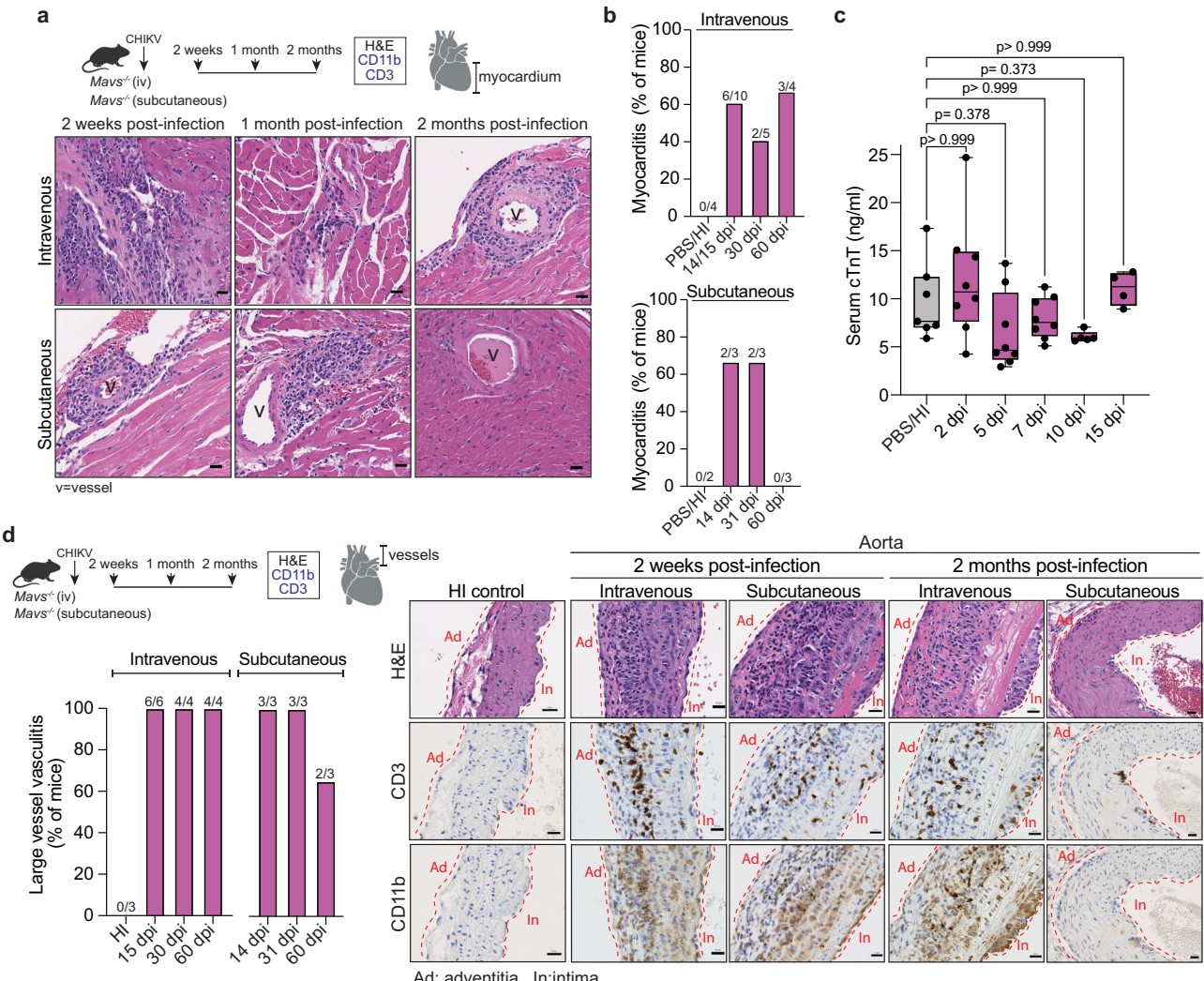

**Fig. 6 | Chronic inflammation in myocardium and vasculature in CHIKV-infected *Mavs⁻ᐟ⁻* mice. a** Upper panel: Schematic representation of the experimental design. Bottom panel: Representative H&E sections from at least three independent mice showing the myocardium of subcutaneous or intravenously inoculated CHIKV-infected *Mavs⁻ᐟ⁻* mice at 2 weeks, 1 month, or 2 months post infection. Vessels within the myocardial region are indicated. Scale = 20 μm. **b** Quantification of the number of mice with positive signs of myocarditis for both intravenous and subcutaneous inoculation routes. Data represent the percentage of mice with any sign of myocarditis over the total amount of mice analyzed. Intravenous: PBS/HI-control (n = 4), 14/15 dpi (n = 10), 30 dpi (n = 5), and 60 dpi (n = 4). Subcutaneous: PBS/HI-control (n = 2), 14 dpi (n = 3), 31 dpi (n = 3), and 60 dpi (n = 3). **c**. Serum levels of cardiac troponin-T measured by ELISA PBS/HI-control (n = 7), 2 dpi (n = 8), 5 dpi (n = 8), 7 dpi (n = 8), 10 dpi (n = 5), and 15 dpi (n = 4). **d**.

Schematic representation of the experimental design. Left panel: quantification of the number of mice with large vessel vasculitis. Data represent the percentage of mice with any sign of vasculitis over the total amount of mice analyzed. HI-control (n = 3). Intravenous: 15 dpi (n = 6), 30 dpi (n = 4), and 60 dpi (n = 4). Subcutaneous: 14 dpi (n = 3), 31 dpi (n = 3), and 60 dpi (n = 3). Right panel: representative H&E and CD3/CD11b IHC sections from at least three independent mice showing vessels attached to the base of the heart of subcutaneous or intravenously inoculated CHIKV-infected *Mavs⁻ᐟ⁻* mice at 2 weeks and 2 months post infection. Scale= 20 μm. Boxplots show median and quartile ranges, whiskers represent the range (**c**). P values were calculated using Kruskal–Wallis with Dunn's multiple comparisons test (**c**). Created with BioRender.com. Source data are provided as a source data file. Related to Supplementary Data 1 and Supplementary Fig 6.

upregulated gene in SARS-CoV-2-infected cardiac tissue in fatal COVID-19 cases[41] and has been found upregulated in a cluster of cardiac fibroblasts during Coxsackievirus B3 infection[42]. Further studies are required to determine the specific role of *Apod* in cardioprotection during CHIKV infection.

Intriguingly, using the KEGG datasets we observed upregulation of the viral myocarditis pathway and downregulation of the cardiac tissue contractility pathway at 5 dpi (Supplementary Fig. 3c, g–i; Supplementary Table 4). However, we did not observe signs of cardiac tissue damage by histopathology (Supplementary Data 1), suggesting that CHIKV infection is triggering transcriptional programs involved in cardiac dysfunction that are not sufficient to result in overt phenotypes of cardiac damage.

We found that CHIKV-infected hearts display a significant yet transient production of proinflammatory chemokines and cytokines, and induction of IFITM3 and ISG15 proteins in both CHIKV-infected and not-infected vimentin⁺ cardiac cells. Interestingly, it has been shown that upon TLRs or RIG-I-like receptor stimulation, human cardiac fibroblasts can produce proinflammatory cytokines and induce an antiviral immune response[43]. Our study confirms the critical role of IFN-I signaling in controlling CHIKV infection in human cardiac fibroblasts. In the context of an immunocompetent host, the infected cardiac tissue mounts a robust and local IFN-I response resolving CHIKV infection within days, with little to no cardiac damage. This observation provides an explanation of why CHIKV-cardiac manifestation is not a pervasive outcome in CHIKV-infected individuals.

In line with this, a report using 12-day-old KO mice for a protein implicated in the production of IFN-β during CHIKV infection (GADD34) revealed a significant increase in viral particles in the heart and development of myocarditis compared to WT[21]. Similar results were described for IFITM3 KO mice during influenza virus infection[44]. Therefore, the risk of cardiac complications in CHIKV-infected patients could be associated with genetic and non-genetic factors (such as co-infections, diet, among others) leading to a reduced IFN-I response. Single nucleotide polymorphisms (SNPs) in TLR3 found in patients with enteroviral myocarditis are associated with a decreased TLR3 mediated signaling and shown an impaired IFN-I response after Cox-sackievirus B3 infection[45]. Thus, SNPs in IFN-I and genes involved in MAVS signaling could underlie an increased susceptibility to cardiac complications in CHIKV-infected patients. SNPs in the coding region of the human *MAVS* gene can result in impaired function[46,47]. In addition, recent studies suggested a link between age and decreased sensing through MAVS[48], which could be a contributing factor to the observed correlation between age and CHIKV severity[5,15]. Further work should focus on determining whether variability in IFN-I and MAVS signaling among individuals can explain the differential susceptibilities to CHIKV-induced cardiac complications.

Multiple PRRs are involved in the in vivo control of CHIKV infection[30]. Here, we demonstrate that CHIKV clearance from infected cardiac tissue is MAVS-dependent, but independent of MYD88, TRIF, or the NLRP3 inflammasome. However, we found reduced number of infectious particles isolated from infected hearts of *Casp1/11*[-/-] and *Myd88*[-/-] mice compared to WT, suggesting that the NLRP3 inflammasome as well as signaling through TLR7 may play a role in the early steps of CHIKV-heart infection. We demonstrate that in the absence of MAVS, CHIKV persist in the cardiac tissue. Intriguingly, systemic levels of both IFNα and IFNβ were significantly higher in *Mavs*[-/-] mice compared to WT, and directly correlated with the viral loads observed in *Mavs*[-/-]. Multiple PRRs are involved in the IFN-I production and protection associated with alphavirus infection[31,49], suggesting that other PRRs, such as TLR3 or TLR7 may contribute to the rescue in systemic IFN-I production in this model[50]. Consistent with this, we found that infected hearts in *Mavs*[-/-] mice display similar levels of local *Isg15* compared to WT, indicating that cardiac tissue responds to IFN-I in the absence of MAVS signaling. These intriguing findings suggest that CHIKV clearance from cardiac tissue in *Mavs*[-/-] mice does not entirely depend on deficient local IFN-I production. Persistent CHIKV infection in cardiac tissue could be associated with the effects of other mediators downstream of MAVS, such as the NFKβ pathway[51]. Additionally, it is possible that the contribution of MAVS signaling in the clearance of CHIKV infection from cardiac tissue is associated with late steps of viral infection, potentially occurring after the systemic IFN-I response reaches negligible levels. In support of this, we observe a positive slope in the detection of CHIKV genomes between 2 to 10 dpi (Fig. 4d), suggesting that active viral replication in *Mavs*[-/-] displays different kinetics compared to WT. Future studies will focus on determining the mechanisms involved in the interaction between MAVS signaling and CHIKV viral persistence in cardiovascular tissue. It is important to note that another possible explanation for our results may be associated, in part, with the bulk nature of our study. Bulk analysis does not provide the resolution to define local effects associated to *Mavs*[-/-] specific responses. An example of the latter was observed for Rift Valley fever virus-infected brains, in which no differences in the global induction of the IFN-I signaling between *Mavs*[-/-] and WT were observed by bulk RNA-seq[52], but became apparent when analyzed at single-cell resolution[52]. Single-cell technologies have been recently applied to the study of cardiac tissue and viral myocarditis[42]. Thus, studies on CHIKV-infected cardiac tissue at single-cell level may help elucidate the role of the local cellular effects in CHIKV clearance.

*Mavs*[-/-] mice display different susceptibilities to related arthritogenic alphaviruses[30,53]. While *Mavs*[-/-] mice infected with Ross River virus developed severe disease and succumbed to infection after 8 dpi[53], CHIKV-infected mice survived when inoculated subcutaneously[30] (Supplementary Table 1). Comparative studies addressing the role of PRR in controlling the infection of related arthritogenic alphavirus would be fundamental for expanding our knowledge in virus-specific mechanisms of pathogenesis.

We demonstrated that CHIKV infection led to focal and multifocal myocarditis in *Mavs*[-/-] mice. Yet, serum levels of cTnT remained unaltered, supporting the absence of cardiac myocyte damage. We found that CHIKV does not infect cardiac myocytes in mice, consistent with the lack of evidence of cardiac myocyte infection in human post-mortem cardiac tissue samples[12]. However, elevated cardiac troponins have been observed in individuals with CHIKV-associated cardiac manifestations[6,7,13,16], raising the possibility that under certain conditions CHIKV can induce cardiac myocyte damage. Another alternative is that human and murine cardiac myocytes have different susceptibilities to CHIKV infection. Further studies using human heart tissue explants or human cardiomyocyte organoids are required to evaluate the relative susceptibilities of human cardiac myocytes to CHIKV infection.

Finally, we found that CHIKV infection led to chronic vascular inflammation in large vessels attached to the base of the heart. This chronic inflammation in the vasculature is intriguing and highlights the vasculature itself as a potential niche for CHIKV infection. Indeed, we detected a differential behavior in CHIKV susceptibilities between the top and the bottom sections of the heart, and CHIKV active replication was detected in cells within vessels up to 10 dpi in *Mavs*[-/-]. These results demonstrate that persistent infection in cardiac tissue leads to cardiac inflammation and potentially cardiac tissue damage, and highlights a potential risk to develop long-term vascular damage associated to CHIKV infection.

Overall, this study demonstrates the direct causality between CHIKV active replication in the heart and the development of cardiovascular manifestations. Future studies focalized on atypical manifestations of arboviral diseases and their pathogenesis are fundamental for understanding the full spectrum of the consequences of these infections. This knowledge is paramount for monitoring of atypical manifestations in endemic areas as well as in future epidemics. In turn, this could be leveraged for early diagnostics, prevention, and therapeutic interventions especially in endemic areas where the circulation of these viruses represents a public health burden[54,55]. Altogether, here we provide mechanistic insight into how CHIKV can lead to cardiac damage, underscoring the importance of monitoring cardiac function in patients with CHIKV infections and laying the foundation for the development of new approaches to prevent viral-induced cardiac complications.

## Methods

### Biosafety
Work with CHIKV was performed in a biosafety level 3 (BSL-3) laboratory at NYU Grossman School of Medicine.

### Cells
Cells were maintained at 37 °C in 5% CO$_2$. Baby hamster kidney (BHK-21, ATCC CCL-10) were grown in Dulbecco's modified Eagle's medium (DMEM; Corning) supplemented with 10% fetal bovine serum (FBS; Atlanta Biologicals) and 1% nonessential amino acids (NEAA, Gibco). Vero cells (ATCC CCL-81) were grown in DMEM with 10% newborn calf serum (NBCS; Gibco). Human primary cardiac cells were obtained from PromoCell, used up to passage 10, and grown following the company recommendations. Human Cardiac Fibroblast (C-12375), human Cardiac Microvascular Endothelial Cells (C-12285), and human Cardiac Myocytes (C-12810) were grown in Fibroblast Growth Medium 3 (PromoCell, C-23025), Endothelial Cell Growth Media MV (Promo-Cell, C-22020), and Myocyte Growth Media (PromoCell, C-22070),

supplemented with 1% Penicillin-Streptomycin (Cellgro, 30-002-CI), respectively. All the cell types were confirmed free of mycoplasma using Lookout Mycoplasma PCR detection kit (Sigma-Aldrich).

## Viruses

CHIKV (CHIKV strain 06-049; https://www.ncbi.nlm.nih.gov/nuccore/AM258994) and CHIKV reporter (CHIKV-ZsGreen; CHIKV strain 06-049) infectious clones were generated as previously described[20,56]. *Virus rescue from infectious clones.* Briefly, 10 µg of each of the infection clone plasmids were linearized overnight with NotI (Thermo-Scientific) at 37 °C, phenol:chloroform extracted, ethanol precipitated, and resuspended in nuclease-free water at 1 µg/µl. In vitro transcribed viral RNAs were produced using the SP6 mMESSAGE mMACHINE kit (Ambion) following the manufacturer's instructions. After RNA synthesis, samples were DNAse treated, and RNAs were purified by phenol:chloroform extraction, ethanol precipitation, and resuspended in nuclease-free water at a concentration of 1 µg/µl. RNA integrity was confirmed by gel electrophoresis (1% agarose). RNA was aliquoted and stored at −80 °C until electroporation. Infectious virus was produced by electroporating BHK cells with in vitro transcribed viral RNAs. For that purpose, cells were trypsinized, washed twice with ice-cold DPBS (Corning), and resuspended at $1 \times 10^7$ cells/ml in PBS. 390 µl of cells were mixed with 10 µg of in vitro transcribed RNA and added to a 2 mm electroporation cuvette (BioRad). BHK-21 cells were electroporated with 1 pulse at 1.2 kV, 25 µF, with infinite resistance. Cells were allowed to recover for 10 min at room temperature (RT), transferred into 6 ml of warm DMEM supplemented with 10% FBS and 1% NEAA, and placed in a T25 flask at 37 °C for 72 h. Virus was harvested, clarified at 1200 × g for 5 min, and 3 ml of the clarified supernatants were used to infect T175 flask to generate working stocks (Passage 1). Viral titers were determined by plaque assay in Vero cells. To generate heat-inactivated virus (HI), virus stocks with titers of $10^6$ PFU/ml were incubated in a water bath at 56 °C for 4 h. Viral titers and loss of infectivity of HI virus were confirmed by plaque assay on Vero cells (see below).

## Plaque assay

Ten fold dilutions of each virus in DMEM were added to a monolayer of Vero cells for 1 h at 37 °C. Following incubation, cells were overlaid with 0.8% agarose in DMEM containing 2% NCBS and incubated at 37 °C for 72 h. The cells were fixed with 4% formalin, the agarose plug removed, and plaques visualized by crystal violet staining. Titers were determined by counting plaques on the highest countable dilution.

## In vitro IFN-I response experiments

Human primary cardiac cells were seeded into 96-well flat transparent black plates (Corning, 07200588) at a density of 18,000 cells/well. After 24 h cells were treated as follows: (i) In vitro poly (I:C) stimulation, (ii) ruxolitinib treatment, or (iii) mock treatment. For poly(I:C) stimulation, human cardiac fibroblasts in 96-well plate were transfected with 62 ng of high molecular weight poly(I:C) (InvivoGen) using TransIT-mRNA Transfection Kit (Mirus Bio), or treated with TransIT transfection alone (carrier control). After 2 h cells were washed and used for downstream experiments. For ruxolitinib condition, cells were pretreated with 5 µM ruxolitinib (Invitrogen, INCB018424) or mock-treated for 3 h, cells were washed and used for downstream experiments. Cells pretreated with poly(I:C), ruxolitinib or mock-treated were then infected with CHIKV-ZsGreen at an MOI of 0.25 or mock-infected with DMEM for 1 h at 37 °C. After 1 h, the inoculum was removed, monolayers were washed once with cold PBS, and 100 µl of Fibroblast Growth Medium 3 (PromoCell, C-23025) was added to the cells. Fibroblast Growth Medium 3 with 5 µM ruxolitinib was added to the cells for the ruxolitinib condition. Cells were incubated at 37 °C with 5% CO2 for 24 h. Cells were fixed with 10% formalin for 1 h, and stained for microscopy (see below).

## In vitro recombinant IFN-I pretreatment experiments

Human primary cardiac cells were seeded into 96-well plates flat transparent black plates (Corning, 07200588) at a density of 18,000 cells/well. After 24 h cells were stimulated with the indicated concentrations of IFNα A/D (Millipore Sigma, #I4401) or IFNβ (Millipore Sigma, #IF014) for 24 h. Cells pretreated with IFN or mock-treated were infected with CHIKV-ZsGreen at an MOI of 0.25 or mock-infected with DMEM for 1 h at 37 °C. After 1 h, the inoculum was removed, monolayers were washed once with cold PBS, and 100 µl of Fibroblast Growth Medium 3 (PromoCell, C-23025) was added to the cells. Cells were incubated at 37 °C with 5% CO$_2$ for 24 h. Cells were fixed with 10% formalin for 1 h, and stained for microscopy (see below).

## Staining and high-content microscopy analysis

After fixation, cells were washed two times with PBS, permeabilized with 0.1% Triton-X/PBS (Fisher, 9002-93-1) for 10 min, and washed again with PBS. For recombinant IFN-I pretreatment experiments, cells were stained with DAPI (Thermo) at a dilution of 1:1000 for 10 min at room temperature and washed three times with PBS. For in vitro response experiments, cells were blocked overnight in 1% BSA/PBS at 4 °C. Cells were stained with anti-ISG15 antibody (ProteinTech, 15981-1-AP) at a dilution of 1:100 in 0.1% BSA/PBS for 1 h at 37 °C, and washed three times with PBS. Cells were stained with secondary AF555-anti-Rabbit antibody (Invitrogen) and DAPI (Thermo) at 1:1000 dilution for 1 h at 37 °C and washed three times with PBS. Plates were imaged using the CellInsight CX7 High-Content Screening Platform (Thermofisher) with the 4× objective in 9 fields covering completely each well from the 96-well plate. The high-content software was used for microscopy and imaging analysis.

## Mosquitoes

*Aedes (Ae.) albopictus* eggs from Tallman Island Wastewater Treatment Plant in Queens, New York City, were collected by the New York Department of Health[57]. Mosquitoes were reared and maintain in Memmert humidified chambers at 28 °C with 70% humidity and 12 h diurnal light cycle. All the transmission studies were done with mosquitoes in generation F7. *Mosquito infection. Ae. albopictus* mosquitoes from Tallman (F7) were exposed to an artificial blood meal containing 1E6 PFU/ml CHIKV. Briefly, viruses were mixed 1:2 with PBS-washed sheep blood (Fisher Scientific) supplemented with 5 mM ATP. Mosquitoes exposed to non-infectious blood meals were used as non-infected controls. Female mosquitoes were allowed to feed on 37 °C blood meals through an artificial membrane for 60–90 min. Engorged females were identified, sorted and incubated at 28 °C with 10% sucrose ad libitum for 7 days. Prior to transmission, mosquitoes were food deprived for 12 h.

## Mice

Mice were maintained at 21–23 °C with 30–70% humidity and a 12 h dark/light cycle in a pathogen-free facility at NYU Grossman School of Medicine. Housing room air exchange rates are set at 10–15 air changes per hour. Mice are provided with food and water ad libitum. All mice are housed in autoclaved individually ventilated caging (Tecniplast, West Chester, PA) at 50–70 cage-volume air changes hourly. Experimental animals were housed in groups of up to 5 mice per cage. Mice were bred and grown in house and maintained in the same facility under specific pathogen-free conditions included the following strains: C57BL/6J wild type mice (000664, Jackson Laboratories). C57BL/6(027) wild type mice (Charles River Laboratories), *Ifnar1*[−/−] (028288, Jackson Laboratories); *Myd88*[−/−] (009088, Jackson Laboratories); *Casp1/11*[−/−] (016621, Jackson Laboratories); *Mavs*[−/−] (008634, Jackson Laboratories), *Trif*[Lps2/Lps2] (005037, Jackson Laboratories). All mice were bred as heterozygotes to generate WT and KO littermate controls, and subsequent WT or KO colonies coming from littermate controls were used for experiments.

**Mouse infections.** Animal experiments were performed in the Animal Biosafety Level 3 (ABSL3) facility of NYU Grossmann School of Medicine, in accordance with all NYU Grossman School of Medicine Institutional Animal Care and Use Committee guidelines (IACUC) and in accordance with guidelines from the National Institutes of Health, the Animal Welfare Act, and the US Federal Law. The study received ethical approval by the NYU Grossman School of Medicine IACUC under IACUC protocol # IA16-01783. Experimental animals of both sexes were used in all experiments, with the exception of bulk RNA-seq experiments in where we used male mice. For bulk RNA-seq we used male mice to minimize potential batch effect due to gender differences. Infections were carried out by either subcutaneous inoculation, mosquito bite, or intravenous injection via tail vein injections. Briefly, 6–8-week-old male and female were infected with 1E5 PFU of CHIKV, CHIKV-ZsGreen, or mock-infected with carrier (PBS or DMEM) or similar inoculum of heat-inactivated virus (HI). For subcutaneous inoculation, mice were anesthetized by inhalation of isoflurane (Henry Schein Animal Health), and injected with 50 µl of virus or control in the left rear footpad. For intravenous inoculation mice were restrained and injected via tail vein injection with 100–200 µl of virus or control inoculum using a 29 G insulin syringe (Fisher Scientific). For natural transmission, mice were immobilized over a mesh covered pint cup containing 4–8 CHIKV-infected or uninfected (blood only) mosquitoes, and mosquitoes were allowed to feed for 40 min. Afterwards, mice were returned to their cages, mosquitoes were killed and homogenized, and viral titers of the infected mosquitoes were determined by plaque assay. Infected experimental mice were euthanized at different timepoints post infection by $CO_2$ inhalation. Blood was collected by cardiac puncture, and mice were immediately dissected for transcardiac perfusion. Mice were perfused with 30 ml of cold PBS with a 21–23 G needle. Quality of cardiac perfusion was assessed by confirming full blood removal in liver and kidneys. Then, hearts were extracted and placed in a 12-well plate with 1 ml PBS. Each heart was opened in four chambers and confirmed free of any blood clot. All the mice in this study were perfused with the exception of those used for histopathology and immunostaining. Heart, calf muscle and pancreas were collected in 500 µl of PBS containing one (heart and pancreas) or two (calf muscle) 5 mm stainless steel beads (QIAGEN), homogenized with TissueLyser II (QIAGEN) at 30 1/s for 2 min (heart and pancreas) or 4 min (calf muscle). Debris was removed by centrifugation at $6010 \times g$ for 8 min. Blood was collected in 1.5 ml tubes containing 20 µl of 0.5 M EDTA and centrifuged at $6010 \times g$ for 8 min. Viral titers were quantified by plaque assay in Vero cells or $TCID_{50}$ in BHK-21 cells.

## Median tissue culture infectious dose ($TCID_{50}$)

Briefly for $TCID_{50}$ determination, BHK-21 cells were seeded into 96-well plates (10,000 cells/well). For heart, muscle and pancreas samples the initial dilution corresponds to a dilution 1:5 from the original homogenate, for serum the initial dilution was 1:20. After the first dilution, ten-fold dilutions of tissue homogenates or serum were made in DMEM supplemented with 1% Antibiotic-Antimycotic (Anti-Anti, Invitrogen). Monolayers were infected with 50 µl of the dilutions. After 1 h, 100 µl of DMEM + 10% FBS + 1% Anti-Anti was added to each well. Cells were incubated at 37 °C with 5% $CO_2$ for 96 h. Plates were inspected under the microscope for evidences of cytopathic effect (CPE). Non-infected tissue was used as negative control, and mock-treated wells were included in each plate. Positive wells, corresponds to any well showing CHIKV specific CPE. Negative wells, corresponds to any well with no CPE. Following incubation, cells were fixed with 10% formalin, and visualized under the microscope. Each sample was measured in 6 or 7 technical replicates and the calculation was performed using the online $TCID_{50}$ calculator (https://www.klinikum.uni-heidelberg.de).

## RNA extractions and RT-qPCR

RNA extractions were performed using TRIzol™ reagent (Invitrogen, 200 µl of clarified tissue homogenate was added to 500 µl of TRIzol™) following the manufacturer's guidelines. Extracted RNA was quantified and diluted to 200 ng/µl. For Taqman assays, a standard curve spanning a range of 100 ng/µl to 1E-6 ng/µl of CHIKV viral RNA was generated for each dataset using in vitro transcribed CHIKV RNAs as described above. For the different tissues, we optimized the amount of total viral RNA to be within the linear range of the calibration curve. The number of genomic (nsp4) or subgenomic (E1) CHIKV RNA was quantified by RT-qPCR using 0.25-1 µg of total RNA as template and the Taqman RNA-to-CT One-Step kit (Applied Biosystems™) in a total reaction volume of 25 µl/well. The following primers and probes were used: CHIKV primers to amplify nsP4 fragment (CHIKV forward (IOLqPCR6856F): 5'-TCACTCCCTGCTGG ACTTGATAGA-3' reverse (IOLqPCR6981R): 5'-TTGACGAACAGAGT TAGGAACATACC-3') and a probe (5'-(6-carboxyfluorescein)-AGGT ACGCGCTTCAAGTTCGGCG-(black-holequencher)-3'). CHIKV primers to amplify E1 fragment (CHIKV forward (IOLqPCR10865F): 5'-TCGACGCGCCCTCTTTAA-3'); reverse (IOLqPCR10991R): 5'-ATCGA ATGCACCGCACACT-3' and a probe (5'-/56-FAM/ACCAGCCTG/ZEN/ CACCCATTCCTCAGAC/3IABkFQ/-3'). The final concentration of primer and probes were 1 µM and 0.6 µM, respectively. Reverse transcription was carried out on QuantStudio 3 qPCR instrument using the following protocol: 48 °C for 30 min, followed by 10 min at 95 °C. Amplification was accomplished over 40 cycles as follows: 95 °C for 15 sec, 60 °C for 1 min. For Sybr green quantifications, levels of *Isg15, Apod, Ccl5, and Ifitm3* in heart homogenates were measured by SYBR Green Master Mix (Thermo) and following manufacturer's instructions on a QuantStudio 3 qPCR instrument. Briefly, cDNA synthesis was performed on purified RNA samples as described above using the Maxima H minus-strand kit (Thermo Scientific) and random primers, using the following protocol: 25 °C for 10 min, 50 °C for 30 min, and 85 °C for 5 min. Amplification was accomplished over 40 cycles as follows: 95 °C for 15 s, 60 °C for 1 min. The melt curve was evaluated for each reaction. Primers: *Isg15* (fw:5'-GGTGTCCGTGACT AACTCCAT-3', and rv:5'-TGGAAAGGGTAAGA CCGTCCT-3'), *Apod* (fw:5'-CTGGGTGGAAACTTCAGTCATCTGATCT-3', and rv:5'-CGCCAG CGTGGCCAGGAAC-3'), *Ccl5* (fw:5'-TGCCCACGTCAAGGAGTATTT C-3', and rv:5'-TCCTAGCTCATCTCCAAATAGTTGATG-3'), *Ifitm3* (fw:5'-TTCAGTGCTGCCTTTGCTC-3', and rv:5'-CCTTGATTCTTTC GTAGTTTGGGG-3') and *18S* (fw:5'-GTAACCCGTTGAACCCCATT-3', and rv:5'-CCATCCAA TCGGTAGTAGCG-3'). RT-qPCR data was collected using QuantStudio software v1.4. The relative expression of the respective genes to 18S expression was calculated using the ΔΔCT method, and values were expressed as fold change normalized to mock-infected control. ΔΔCT for each sample (infected and mock-infected) was calculated against the average of all the ΔCT values calculated for the mock condition.

## RNA extractions and RNA-sequencing analysis

For bulk RNA-sequencing C57BL/6 male mice were infected intravenously with 1E5 PFU of CHIKV or HI control, and hearts were perfused with 30 ml of PBS and collected at 2 and 5 dpi. Hearts were collected in 500 µl of TRIzol™ (Invitrogen) with one 5 mm stainless steel bead and processed as described above. RNA was extracted using Zymo directzol RNA purification kit (Zymo research, #R2072) according to the manufacturer's instructions. A total of 500 ng per sample was sent to the NYU Genome Technology Center for quality control assessment, automated TruSeq stranded total RNA, with RiboZero Gold library preparation, and Illumina Novaseq6000 sequencing (single pair 100). FASTQ files were aligned to a concatenated file containing the mouse genome (version mm10) and the CHIKV genome (AM258994) through STAR (v2.5.2) in two pass basic mode using the gencode transcript annotation (vM25). Structural and non-structural CHIKV polyprotein

coordinates (CHIKVgp1 and CHIKVgp2) were added to the transcriptome annotation, and the transcript count matrix was generated using the summarize overlaps function form the Genomic Alignments (v1.22.1) package. Differential gene expression analysis based on negative binomial distribution was performed using DEseq2 (v1.26.0). For pathways enrichment analysis, genes were ranked based on $-Log_{10}(FDR)$ values, assigning a positive or negative value depending on the direction of change. The pathway lists were obtained from the msigdbr (v7.2.1) package (Hallmark, REACTOME or KEGG) and used as input for GSEA pathway enrichment analysis through the fgsea (v1.12.0) R package. Volcano plots were generated using ggplot2 (v3.3.3) and heatmaps were generated using pheatmap (v1.0.12).

## Tissue preparation for immunofluorescence and fluorescence microscopy

Hearts were removed from euthanized mice and fixed in PFA, lysine and periodate buffer (0.05 M phosphate buffer, 0.1 M L-lysine, pH 7.4, 2 mg/ml $NaIO_4$, and 10 mg/ml paraformaldehyde) at 4 °C for 24 h. Next, tissues were dehydrated in 30% sucrose overnight at 4 °C and subsequently embedded in OCT media. Frozen tissue sections were sectioned at 5 μm thickness at the NYU Experimental Pathology Research Laboratory. Fc receptors were blocked with 0.5% anti-CD16/32 Fc block antibody (Biolegend, clone 93) diluted in PBS containing 2% goat serum (Vector Laboratories, S-1000), 2% FBS (Atlanta) and 0.25% Triton-X 100 (Fisher, BP151-100) for 1 h at RT. Sections were washed 5 times with the same buffer, and treated with Trueblack Lipofuscin autofluorescence quencher (Biotium, CAT23007) in ethanol 70% (5% v/v) for 1 min. Sections were washed 5 times and stained with eF570-αSMA (Invitrogen, clone: 1A4) with either APC-CD45 (Biolegends, clone: 30-F11), AF647-TNT (BD Pharmingen, clone: 13-11), AF647-CD31 (BioLegends, clone: MEC13.3) or AF647-Vimentin (Abcam, clone: EPR3776) for 1 h, at RT. All antibodies were used at 1:100 dilution. For IFITM3 or ISG15 staining, slides were stained with 1:100 anti-Fragilis antibody (Abcam, ab15592, polyclonal) or 1:100 anti-ISG15 antibody (Invitrogen, PA5-79523, polyclonal) for 1 h at RT, respectively; washed and stained with a secondary AF555-anti-Rabbit antibody (Invitrogen) at 1:1000 dilution for 1 h. All the slides were then stained with DAPI (Thermo) at 1:1000 dilution. Sections were washed with the same buffer and mount for microscopy analysis. Images were acquired using a Zeiss LSM 880 confocal microscope (Carl Zeiss).

## Imaris surface colocalization analysis

The imaging data were processed and analyzed using Imaris software version 9.8 (Bitplane; Oxford Instruments). Briefly, surfaces were built for each channel using specific threshold, number of voxels and surface grain size. The selection of parameters was dependent on each experimental set of images. Once the parameters were defined batch analysis was performed to calculate the total amount of overlapping surfaces between two channels. Analysis was performed on 2 × 2 tiled images acquired at ×20 magnification from the Zeiss 880 confocal microscope, where 2–4 different sections per slide were imaged from two to three independent animals per condition.

## Histopathology

The heart and part of the ascending thoracic aorta were removed from euthanized mice, fixed in 10% buffered formalin (Fisher Scientific) for 72 h, and processed through graded ethanol, xylene and into paraffin in a Leica Peloris automated processor. Five-micron paraffin-embedded sections were cut parallel to the long axis of the heart from two to three distinct levels (400 μm, 800 μm, and 1200 μm; or 800 μm, and 1200 μm). Sections of the 4-chamber heart were deparaffinized and stained with hematoxylin and eosin (H&E) on a Leica ST5020 automated histochemical stainer or immunostained on a Leica BondRX® autostainer, according to the manufacturers' instructions. In brief, sections for immunostaining underwent

epitope retrieval for 20 min at 100 °C with Leica Biosystems ER2 solution (pH9, AR9640) followed by a 30 min incubation at 22 °C with either anti-CD11b (Novus, NB110-89474) diluted 1:10,000, with anti-CD3 (CST, 78588S; clone E4T1B) or with anti-cleaved CASP3 (CST, 9579S, clone D3E9) diluted 1:1000. The primary antibodies were detected using the BOND Polymer Refine Detection System (Leica, DS9800). As positive controls of the IHC, sections of mice spleen were stained with anti-CASP3 (Supplementary Fig. 2b), anti-CD3 and anti-CD11b. Sections were counterstained with hematoxylin scanned on either a Leica AT2 or Hamamatsu Nanozoomer HT whole slide scanners and imported into the NYU Omero image database for viewing and annotation. The severity of tissue pathology and determination of the different categories described in Table S1 and S5, were blindly analyzed by a pathologist (N.N.). Masson trichrome staining was performed as previously described[58]. In brief, deparaffinized slides were further fixed in Bouin solution for 1 h at 60 °C. The slides were then serially stained with Weigert Hematoxylin (Polysciences, 25088B1&B2) for 10 min, followed by Biebrich Scarlet-Acid Fuchsin (Polysciences, 25088C) for 1 min and Phosphotungstic acid (Polysciences, 25088D) for 10 min with washes between each step. Slides were then transferred directly into aniline blue (Polysciences, 25088E) for 5 min and differentiated in 1% acetic acid (Polysciences, 25088f) for 1 min before dehydration and mounting with permount (Fisher, SP15-100).

## CASP3 IHC quantification analysis

To determine the number of cleaved $CASP3^+$ cells whole slide scans were analyzed using Visiopharm$^R$ image analysis software version 2021.09. Images were imported to Visiopharm$^R$'s database. Manual outlining of regions of interest (ROI) was performed to divide the four-chamber heart section into right, left ventricles, septum and atria. A Visiopharm$^R$ app was generated to detect DAB-stained cells in the mouse heart. Briefly, image pre-processing steps were run to create features enhancing DAB signal and nuclear signal through color deconvolution and filtering. Segmentation was achieved using a trainable linear bayesian algorithm. Post-processing steps were applied to define nuclei as positive or negative. Positive and negative controls were included for each analysis. Results were provided as number of total nuclei, number of positive and number of negative nuclei per ROI analyzed.

## Cytokine and chemokine analysis

Mice were infected intravenously with 1E5 PFU of CHIKV or HI control, and heart were collected at 2 and 5 dpi. Heart tissue homogenates generated as described above were diluted 1:5 in 5X PBS:HALT protease inhibitor EDTA-free (Thermo). Total amount of protein was quantified using BCA protein assay (Thermo Fisher Scientific) and samples were diluted to 1.5 mg/ml in PBS:HALT protease inhibitor, 75 μl aliquots were made and stored at −80 °C until needed. For serum, blood was collected as described above, incubated at RT for 30 min, and centrifuged at 1000 × g for 15 min at 4 °C. The upper phased was then transferred to a clean tube and centrifuged at 10,000 × g for 10 min. Samples were moved to a clean tube and then 50 μl aliquots were made and stored at −80 °C until needed. Heart homogenates were analyzed for cytokine and chemokine levels using a Bio-Plex Pro Mouse Cytokine 23-plex Assay kit (Bio-Rad, #m60009rdpd). Heart homogenates and serum were analyzed for IFN-I levels using ProcartaPlex mouse IFN-α/IFN-β 2-plex (ThermoFisher, EPX02A-22187-901). Cytokine and chemokine were recorded on a MAGPIX machine (Luminex) and quantitated via comparison to a standard curve. xPONENT software (v 4.3.229.9) was used for the data collection and analysis. The values reported for all the analytes were within the detection limit of the method and represent the result values from the Luminex quantifications based on the individual calibration curves for each analyte.

## Mouse cardiac troponin-T ELISA

Mice were infected intravenously with 1E5 PFU of CHIKV or HI control, and blood collected as described above, incubated at RT for 30 min, and centrifuged at $1000 \times g$ for 15 min at 4 °C. The upper phased was then transferred to a clean tube and centrifuged at $10,000 \times g$ for 10 min. Samples were moved to a clean tube and then 50 µl aliquots were made and stored at −80 °C until needed. Serum was analyzed for mouse cardiac troponin-T using a precoated microtiter plate (amsbio, AMS.E03T0017) and using a recombinant cardiac troponin-T standard curve according to the manufacturer's instructions.

## Single-cell analysis of *Tabula muris* datasets

Data from *Tabula muris* (Tabula Muris et al., 2018) was downloaded from https://tabula-muris.ds.czbiohub.org. Data was analyzed using the Seurat (v3.2.1) R package. Briefly, data was normalized using the "Normalize Data" function with default parameters. After normalization, we applied the function "Find Variable Features" to identify genes of interest used for downstream dimensionality reduction via the function "RunPCA" and clustering through the "Find Neighbors" and "Find Clusters" functions, with default parameters. The original cluster annotations from *Tabula Muris* were maintained. Dot plot for normalized expression of potential cell markers was generated using the "DotPlot" function.

## Statistical analysis

Statistical significance was assigned when $P$ values were <0.05 using Prism Version 9.5.1 (528) (GraphPad) or R-studio (v3.6.0). Specific tests with exact $P$ values are indicated in the Figure legends. All experiments were performed at least in biological duplicates.

## Data availability

Bulk RNA-sequencing datasets generated in this study are available at GEO: GSE204689. Microscopy image data is stored in OMERO Plus v5.6 and is accessible through the NYU Data Catalog: https://datacatalog.med.nyu.edu/dataset/10621. Public datasets: mouse genome (version mm10, https://hgdownload.soe.ucsc.edu/goldenPath/mm10/bigZips/), CHIKV genome (AM258994, https://www.ncbi.nlm.nih.gov/nuccore/106880547). Public database: *Tabula muris* (https://tabula-muris.ds.czbiohub.org). Datasets generated and/or analyzed during the current study are appended as supplementary table. Source data are provided with this paper.

## Code availability

Code is available at GitHub: https://github.com/fizzo13/CHIKV.

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

## Acknowledgements

We would like to thank all members of the Stapleford lab for helpful comments on this manuscript. We thank NYUMC Experimental Pathology Research Laboratory, Mark Alu, and Branka Brukner Dabovic. We are also grateful to the NYU Genome Technology Center for assistance during sequencing and the NYU Langone Microscopy Laboratory for assistance during imaging. We thank Drs. Meike Dittmann and Ken Cadwell who kindly provided essential equipment and reagents. We thank Drs. Dominick Papandrea and Ludovic Desvignes for BSL3 assistance. We thank Drs. Ken Cadwell and Victor Torres for critical reading of the manuscript. This work was supported by a start-up package from the NYU Grossman School of Medicine (K.A.S), NIH/NIAID R01 AI162774-01A1 (K.A.S) and 1R01AI143861 (K.M.K), the NYU Cardiovascular Research Center Program pilot grant (K.A.S. and M.G.N), the American Heart Association Postdoctoral Fellowship 19-A0-00-1003686 (M.G.N.), Public Health Service Institutional Research Training Award T32 AI 7180-39 (B.A.R-R.), Training Program in Immunology and Inflammation Training Award 5-T32 AI100853-10 (E.B. and P.D.Y), the American Society of Hematology Fellow-to-Faculty Scholar Award ASH 2O4377-01 (F.I.), NIH 5F32 HL153598 (P.D.Y). The Experimental Pathology Research Laboratory (RRID:SCR_017928), the NYU Langone Microscopy Laboratory (RRID: SCR_017934) and the NYU Genome Technology Core (RRID: SCR_017929) are partially supported by NYU Cancer Center support grant P30CA016087 and by NYU Langone's Laura and Isaac Perlmutter Cancer Center. The Akoya® multispectral imaging system was purchased through a Shared Instrumentation Grant S10 OD021747.

## Author contributions

M.G.N. and K.A.S. conceived the project and designed experiments; M.G.N, S.N.S, E.B, S.T.Y, P.D.Y, B.A.R.R. performed experiments, F.I. analyzed RNA-seq data; M.G.N, E.B., F.I., and K.A.S. analyzed data, C.L. oversaw histology experiments, N.N. performed histopathology analysis; V.M. develop the Visiopharm app for histology analysis; M.Z.D. performed slide cryosections; K.K and K.A.S supervised the project. M.G.N. and K.A.S. wrote the original draft, and all authors were involved in manuscript review and editing.

## Competing interests

The authors declare no competing interests.
