## [Peer Review File · Nature Communications]

REVIEWERS' COMMENTS

Reviewer #1 (Remarks to the Author):

The authors have conducted additional experiments to support the article's findings, which has strengthened the paper. Appropriate explanations and descriptions have been added to the text and the figure legends. However, since the data on Mxra8 receptors have been removed from the article, questions remain about the receptors important in CHIKV infection of cardiac tissues. Here are some minor comments:

1. In analyzing CHIKV infection of cardiac tissues, the authors stress that perfusion of the heart was performed to prevent virus contamination from the bloodstream. Is there a significant difference in viral titers in heart tissues examined with or without perfusion?
2. The authors could consider including the demonstration of IFN production by CHIKV-infected human cardiac myocytes in the article.
3. It would be beneficial if the authors provided alternative explanations for the chronic infections observed in Mavs^{-/-} mice, given the similarity of IFN levels and IFN-induced genes found in WT mice.

Reviewer #2 (Remarks to the Author):

This revised version of the article entitled "MAVS signaling is required for clearance of chikungunya heart infection and prevention of chronic vascular tissue inflammation" brings many improvements. This article addresses an important specific issue concerning chikungunya virus infection (CHIKV) and its consequences on the heart diseases, which has not been explored much to date. Indeed, cardiac complications have been observed in patients with comorbidities, and cardiac manifestations have been reported in young patients without comorbidities. It is therefore important to understand and decipher the mechanisms leading to these cardiac attacks in the two situations presented. The exhaustive data presented in the article provide a solid basis for future understanding of the mechanistic of human cardiac disease associated with CHIKV infection and pave the way for exploring the individual characteristics of patients affected by these cardiac injuries. The authors have taken very careful note of the comments of all reviewers. They have provided argued responses, additions or rewording of the text, and new experiments on primary human heart cells. These in vitro CHIKV infection experiments of primary human cardiac fibroblasts, myocytes and endothelial cells of the microvasculature undoubtedly enrich the relevance of the article.

Reviewer #3 (Remarks to the Author):

The authors have rigorously and thoroughly addressed the comments from the prior review. The manuscript is substantially improved and should be of keen interest to the field.

Reviewer #1 (Remarks to the Author):

Occasional cases of heart complications have been reported with chikungunya virus infections in humans. The authors examined the damage to cardiac tissues in mice infected with CHIKV. Vimentin+/ α SMA- cardiac fibroblasts were the primary targets of CHIKV infection in wild-type (WT) mice. In WT mice, heart infection was cleared without inflammation or damage to cardiac tissue. Infection activates interferon-1 (IFN-1) signaling as shown by the expression of the proteins ISG15 and IFITM3 in cardiac tissues. Transcriptome analysis of CHIKV-infected cardiac tissues revealed activation of pathways associated with IFN-signaling, innate immunity, and adaptive immunity. In *ifnr*^{-/-} mice, mice succumb to CHIKV infections resulting in death. CHIKV infection persists longer in cardiac tissues in MAVS-deficient mice (*Mavs*^{-/-}) than in WT mice. The persistence of infection in *Mavs*^{-/-} mice was associated with infiltration of CD3⁺ and CD11b⁺ cells in cardiac tissues.

We thank the reviewer for the critical points addressed since we believe it has greatly improved our manuscript. In addition to addressing this reviewer comments (see point-by-point response), additional modifications and results have been included in this new version of the manuscript as summarized here:

1) Differential susceptibilities of cardiac tissue to CHIKV infection

- With the objective of underscoring the differences in distribution of infected cells across the cardiac tissue, we physically separated the heart into (1) top (vessels attached to the base of the heart as well as the atria) and (2) bottom (myocardium), and performed downstream analysis of the viral burden by qPCR. We identified a difference in viral RNA clearance between the top and the bottom part of the heart, demonstrating that regions containing the atria and vessels can harbor viral RNA for a longer period of time (See Fig. 1d and Fig. 4f). These results suggest that sections of the heart enriched in major vessels are more susceptible to sustained infection over time.

2) Cardiac fibroblasts are the main target of CHIKV infection in the hearts.

- In order to further strengthen our findings, we have now expanded our studies to human primary cardiac cells, including cardiac fibroblasts, microvasculature cardiac endothelial cells, and cardiomyocytes. Infection of the cardiac fibroblasts as well as the cardiomyocyte cells *in vitro* was observed when performing growth curves at a multiplicity of infection of 0.1 (See Fig. 1g).

3) Local type-I interferon response in cardiac tissue is essential to control CHIKV infection and prevent tissue damage.

- We measured systemic and local IFN-I levels at 12 and 24 hpi (Extended Data Fig. 3f).
- We added IF analysis showing local production of ISG-15 and colocalization with vimentin⁺ cells in heart tissue upon CHIKV infection (Fig. 3b).
- As suggested by the reviewer, we addressed the dependency of type I interferon to control CHIKV infection in human primary cardiac fibroblasts cells (see point-by-point response and Fig. 3c-d).

4) MAVS signaling in cardiac tissue mediates CHIKV clearance and tissue damage.

- We added data showing the requirement of a full MAVS KO phenotype in order to obtain persistent infection and tissue damage (Extended data Fig. 5a and Fig 5).
-We compared the kinetic of CHIKV clearance from infected hearts and calf muscle and show that while no significant differences in CHIKV genomes in calf muscle are observed at 10 dpi, cardiac tissue in MAVS KO mice harbored significant levels of viral RNA compared to WT (Fig 4d-e).
- We measured local and systemic IFN-I in both *Mavs* KO and WT mice at 24 hpi (see point-by-point response and Fig. 4h-i).

5) Persistence of CHIKV infection in the heart of MAVS KO led to focal myocarditis and chronic big vessels vasculitis.

- We have expanded our studies regarding the pathogenesis of the MAVS KO mice to 60 dpi and we included mice infected through subcutaneous inoculation. (See Fig 6, Extended Data Fig 6 and Extended Data Table 1).

Point-by-point response:

Major comments:

1. Identifying the factors and/or comorbidities contributing to cardiac damage in CHIKV-infected patients is crucial. According to the authors, future studies should investigate whether differences in IFN-I responses between individuals could contribute to the differential susceptibilities to cardiac complications resulting from CHIKV infection. Human data should be considered along with mouse data to support this study's conclusions. Including human data in the paper would enhance the relevance of the findings in mice. The authors should also investigate the levels of cardiac enzymes, cardiac troponin etc., in patients with acute and chronic CHIK disease or from CHIKV patients with low and high viral load. The human data would significantly strengthen the paper.

We thank the reviewer for this comment. The inclusion of human data is fundamental for the complete understanding of this atypical manifestations and why arboviral infections can lead to cardiovascular damage in certain individuals, and as stated in the discussion section it is part of our future projects. While of great interest, identifying the factors and comorbidities associated with cardiac damage in CHIKV-infected patients represent an epidemiological study requiring both a large patient cohort in order to accomplish sufficient statistical power, coupled with either targeted or whole genome sequencing studies to identify genetic variants potentially underlying the increase cardiac risk during CHIKV infection. While we recognize that this study is of the outmost importance, it lies beyond the scope of our current work.

The scope of the current study is to define the mechanisms through which CHIKV can lead to cardiac complications. To accomplish this, we leveraged different mouse models that provide valuable information regarding the dependence on IFNAR and MAVS for efficient clearance of infection from the cardiac tissue and its role in cardiac tissue damage. However, to partially address the reviewer's comment, we have now expanded our results to human primary cardiac cells (See below and **Fig. 1g and Fig.3c-d**).

In addition, we have now included in the discussion a new study published in January 2023 reporting significant differences in cardiovascular biomarkers between CHIKV fatal and non-fatal cases (**PMID:36183865**).

2. The paper would be substantially strengthened by including data from primary human muscle cells. Human cardiac fibroblasts cells are available commercially. It will be essential to examine the importance of IFN-1 signaling in the infection of cardiac fibroblasts with CHIKV.

We thank the reviewer for this suggestion, since it has improved our work. As stated above, we have now expanded our studies to human primary cardiac cells (cardiac fibroblasts, microvasculature cardiac endothelial cells, and cardiomyocytes). Using these primary cell types, we interrogate whether primary human cells *in vitro* recapitulate *in vivo* studies using mouse models. For that purpose, we performed growth curves at a multiplicity of infection of 0.1 and demonstrated that human primary cardiac fibroblasts, as well as cardiac myocytes can be infected *in vitro*. Interestingly, in contrast to what we observed in our *in vivo* experiments, human cardiac myocytes in culture can sustain active viral replication (**Figure 1g**, open circles). However, careful interpretation of these result must be taken. Cardiac myocytes in culture lose specific features that can potentially impact susceptibility to CHIKV infection. For example, while cardiac myocytes in tissue are characterized for being multinucleated and quiescent *in vivo*, the *in vitro* culture reverts these phenotypes, resulting in mono-nucleated and proliferating cells (**PMID:9798514**). Another factor that may confound these results is that *in vitro* infections bypass major barriers associated with *in vivo* infections, such as dissemination and the immune system. Therefore, while these results support the role of cardiac fibroblasts as the main target of CHIKV, for which we show supporting evidence *in vivo*, the observation regarding *in vitro* infection of cardiomyocytes must be taken carefully. These data have been added to figure 1 (**Fig 1g**).

In addition, the following modifications have been added to the text (L148-162):

“Finally, we sought to determine whether human cardiac cells show similar susceptibilities to CHIKV-infection than those observed in mice. For that purpose, we evaluated the capacity of CHIKV to infect human primary cardiac cells *in vitro*. We infected human primary cardiac fibroblasts (hCF), human cardiac

myocytes (hCM) and human cardiac microvasculature endothelial cells (hCE) at an MOI of 0.1 and measured the production of infectious particles. We found that CHIKV can efficiently infect hCFs (**Fig. 1g, black circles**), but no hCEs (**Fig. 1g, squares**), supporting our *in vivo* observations in mice. Interestingly, hCMs in culture produced CHIKV infectious particles at similar levels than the observed for hCFs (**Fig. 1g, open circles**). However, careful interpretation of this result must be taken, since hCMs in culture lose specific features that can potentially impact susceptibility to CHIKV infection. For example, while cardiac myocytes in tissue are characterized for being multinucleated and quiescent *in vivo*, the *in vitro* culture revert these phenotypes, resulting in mono-nucleated and proliferating cells²⁷. Altogether, our results demonstrate that CHIKV actively replicates in cardiac tissue with cardiac fibroblasts being the primary cellular target of CHIKV infection in immunocompetent mice.”

Next, we study the dependence on IFN-I signaling in human primary cardiac fibroblasts during CHIKV infection. We studied this by: (i) addressing the capacity of these cells to generate a robust IFN-I response after poly(I:C) stimulation and CHIKV infection; (ii) analyzing the susceptibility of these cells to CHIKV infection in the absence of IFN-I signaling using the JAK1/JAK2 inhibitor ruxolitinib; and (iii) we demonstrated that human cardiac fibroblasts respond to both IFN alpha and beta and generated a refractory state that inhibits CHIKV infection. These demonstrate that IFN-I signaling is a fundamental actor in controlling CHIKV infection in human cardiac fibroblasts, and further support our findings in mouse models. These results have now been added to **Fig 3c-d** and the text (L239-253):

“To determine if human cardiac cells respond to CHIKV infection similarly to murine cardiac tissue, we evaluated IFN-I signaling in hCFs infected with CHIKV. We evaluated this by: (i) measuring the capacity of hCFs to generate a robust IFN-I response after poly(I:C) stimulation and CHIKV infection; (ii) measuring the susceptibility of these cells to CHIKV infection in the absence of IFN-I signaling using the JAK1/JAK2 inhibitor ruxolitinib (Rux); and (iii) inducing a refractory state upon increasing doses of IFN-I pre-treatments. Indeed, we found that hCFs efficiently respond to poly(I:C) stimulation, resulting in ~ 80% of the monolayer expressing ISG15 (**Fig. 3c, left panel**). In agreement with our *in vivo* data, we found that infected hCFs responded to CHIKV infection by producing significant levels of ISG15 (**Fig. 3c, left panel**). In addition, Rux treatment significantly increased the percentage of CHIKV infected cells (**Fig. 3c, right panel**), and correlated with a reduction of ISG15 production (**Fig. 3c, left panel**). Furthermore, we found that exogenous addition of either IFN α or IFN β prevented CHIKV infection in a dose response manner, with IFN β showing increased blocking capacity (**Fig 3d**). These results support that, similarly to murine cardiac tissue, IFN-I signaling is critical in controlling CHIKV infection in human cardiac fibroblasts.”

3. Extended Fig 1A - There is a need for the investigators to explain why in intravenous CHIKV-zGreen infections, fold CHIKV RNA levels in Mxra8 Δ 8/ Δ 8 were much lower compared to WT CHIKV levels in intradermal infections (Fig 1c).

We thank the reviewer for pointing this out. For our study, the goal of using MXRA8 deficient mice was to further confirm our findings of active replication in cardiac tissue, and not necessarily to identify receptors implicated in CHIKV heart infection. We want to underscore that while further characterization of the receptors involved in CHIKV infection are valuable, the focus of the current study is centered in understanding how cardiac tissue responds to CHIKV infection, and identify mechanisms implicated in development of CHIKV-mediated heart disease. Therefore, considering that all the reviewers' comments and the conflicting data regarding CHIKV receptors, we decided to remove these data from our study.

4. Fig 1b shows the quantitation of genomic and sgRNA by intravenous route. Show the levels of the same in all three routes of inoculations.

We thank the reviewer for this comment. We have now measured active replication for both intravenous and subcutaneous inoculation routes. This was measured by: (i) qPCR with specific primers targeting nsp4 (as a surrogate of gRNA) and E1 (for sgRNA) as previously described (**PMID: 31465513 and PMID: 24131709**). We monitored for active replication in the heart by looking at timepoints 2, 5 and 10 dpi. Control animals were inoculated with the same virus preparation, but heat inactivated. Active replication was assessed by the ratio between E1 and nsP4 targets. These data have been added to **Fig. 1b** and to the text (L100-112):

“To evaluate if the CHIKV viral particles detected in the heart were actively replicating, we measured the ratio between CHIKV nsp4 and E1 transcripts by RT-qPCR as a surrogate of genomic and sub-genomic RNA, respectively. As a non-replicative control, we infected mice with heat-inactivated (HI) CHIKV. While no active replication was detected for the HI control, CHIKV active replication in the heart was detected for both subcutaneous and intravenous inoculation routes (**Fig. 1b**). For intravenously infected mice active replication was detected at 2 dpi (R [Ratio E1/nsp4] = 3.80, p = 0.04) and 5 dpi (R = 8.11, p = 0.008), while for subcutaneously infected mice active replication was detected at 5 dpi (R = 3.88, p = 0.008). The difference in kinetics between inoculation routes is consistent with the requirement of subcutaneous inoculation to reach viremia in order to disseminate and infect cardiac tissue, while the intravenous route bypasses this stage of the infection. These results demonstrate that cardiac tissue is a direct target of CHIKV infection in immunocompetent mice.”

To further support these data, we infected mice subcutaneously with a reporter virus CHIKV-ZsGreen. In particular, since the expression is regulated by CHIKV sub-genomic promoter, ZsGreen is detected exclusively after active replication of the CHIKV genome. These data have been added to **Extended Data Fig. 1b**.

Unfortunately, for the mosquito bite inoculation route, we were not able to perform these quantifications due to the low RNA quality from the mosquito bite experiments.

5. In analyses of pro-inflammatory cytokines, no information has been presented regarding levels of IFN-1 in cardiac tissues (Fig 2C and extended Fig 2). Include data on IFN-1 levels in *Mavs*^{-/-} mice to explain the induction in *Isg15* RNA levels similar to WT mice.

We thank the reviewer for pointing this out. Measurements of IFN α and IFN β levels in cardiac tissue homogenates and serum at 12 and 24 hpi have been added to **Extended Data Fig. 3f**. The following modification have been added to the text (L221-225):

“While the protein levels of serum IFN α and IFN β were elevated at 24 hpi, no changes in local levels of IFN α and IFN β were observed in infected hearts at 12 or 24 hpi (**Extended Data Fig. 3f**). This suggests that circulating IFN-I is a contributor to the local IFN-I response elicited by cardiac tissue upon CHIKV infection.”

In addition, in order to interrogate the nature of the *Isg15* induction observed in CHIKV-infected MAVS KO mice we measured the circulating levels (serum) and tissue levels (heart homogenates) of both IFN α and IFN β , for WT and MAVS KO mice at 24 hpi. We found that in the absence of MAVS signaling, mice can produce significant levels of systemic IFN alpha and beta, and IFN alpha in cardiac tissue. These data have now been added to **Figure 4 h-i**, and the following modifications have been added to the text (L308-316):

“Of note, we found that *Mavs*^{-/-} infected hearts induced similar levels of *Isg15* mRNA compared to WT (**Fig. 4g**). Prompted by this result, we measured the local and systemic production of IFN α or IFN β at 24 hpi, which corresponds to the peak of induction of *Isg15* in cardiac tissue (**Fig. 4g**). We found that while the local IFN-I production is low in the heart (**Fig. 4h**), both WT and *Mavs*^{-/-} mice produce significant systemic levels of both IFN α and IFN β (**Fig. 4i**). Surprisingly, the systemic levels of IFN α were higher in *Mavs*^{-/-} mice compared to WT (**Fig. 4i**), which correlates with the significantly higher viral loads observed in *Mavs*^{-/-} (**Fig. 4b and 4d-e**). These data suggest that the delay in CHIKV clearance might not be entirely explained by a reduced IFN-I induction associated with *Mavs*^{-/-}.”

And (L428-443):

“Intriguing, systemic levels of both IFN α and IFN β were significantly higher in *Mavs*^{-/-} mice compared to WT, and directly correlated with the viral loads observed in *Mavs*^{-/-}. Multiple PRRs are involved in the IFN-I production and protection associated with alphavirus infection^{31,49}, suggesting that other PRRs, such as TLR3 or TLR7 may be contributing to the rescue in the production of systemic IFN-I in this model⁵⁰. In line with this, we found that in *Mavs*^{-/-} mice, infected hearts display similar levels of local *Isg15* compared to WT, indicating that cardiac tissue is responding to IFN-I in the absence of MAVS signaling. These data suggest that either: (i) the delay in CHIKV clearance from cardiac tissue does not entirely depend on the reduced IFN-I induction associated with *Mavs*^{-/-} cells, or (ii) the bulk nature of our study does not provide the resolution to define local effects associated to *Mavs*^{-/-}”

^ specific responses. An example of the latter was observed for Rift Valley fever virus-infected brains, in which no differences in the global induction of the IFN-I signaling between *Mavs*^{-/-} and WT were observed by bulk RNA-seq⁵¹, but became apparent when analyzed at single-cell resolution⁵¹. Single-cell technologies have been recently applied to the study of cardiac tissue and viral myocarditis⁴². Thus, studies on CHIKV-infected cardiac tissue at single-cell level may help elucidate the role of the local cellular effects in CHIKV clearance.”

6. In Fig 2 - the findings indicate that cardiac tissue was infected in an MXRA8-dependent manner. However, many CHIKV infections occur in cardiac tissues of *Mxra8* Δ 8/ Δ 8 mice, indicating that the Fhl1 receptor and/or other receptors could also be important in promoting infection in cardiac tissues. Expression of *Mxra8* and *Fhl1* were inferred from the Tabula Muris dataset. CHIKV also utilizes other receptors for entry into susceptible cells, such as CD147 (De Caluwe et al., *FrontMicrobiol* 12 (Art 615165 (2021) and phosphatidyl serine receptor (TIM-1) (Kirui et al., *Cells* 10: 1828 (2021). Analyze the levels of expression of CD1147 and TIM-1 receptors in cardiac tissues. Furthermore, demonstrate the expression levels of *Fhl* in cardiac fibroblasts of *Mxra8* Δ 8/ Δ 8 mice. In the Discussion, the authors should discuss the importance of different receptors in CHIKV infection of cardiac tissues.

We thank the reviewer for pointing this out. However, considering all the reviewers' comments and the fact that our goal for this study was not to define the receptor/s driving CHIKV infection in cardiac tissue, we decided to remove all the data associated to receptors from the study.

7. The infection with CHIKV resulted in the up-and down-regulation of several genes in cardiac fibroblasts. Include RT-PCR data on a few crucial genes such as *ApoD*, *IFITM3*, *IsG15*, and *IFN-1* response genes.

In our bulk RNA-seq analysis, we found that CHIKV infection results in the up regulation and downregulation of several genes in mouse cardiac tissue. We incorporate to **Extended Data Fig 3d**, the qRT-PCR validation for the following genes: *Apod*, *Ifitm3*, *Isg15*, *Mx1* and *Ccl5*.

8. It would be helpful if the authors discussed the results of the reduced CHIKV replication in mice deficient in *Myd88*, *TrifLps2/Lps2* and *Casp1/11* compared to WT mice.

We thank the reviewer for bringing this point to our attention. The following changes have been added to the text (L424-427):

“However, we found reduced number of infectious particles isolated from infected hearts of *Casp1/11*^{-/-} and *Myd88*^{-/-} mice compared to WT, suggesting that the NLRP3 inflammasome as well as signaling through TLR7 may play a role in the early steps of CHIKV-heart infection.”

9. Include the disease scores and mortality of mice infected with CHIKV in WT, *ifnar1*^{-/-}, and *Mavs*^{-/-} mice. Literature reports on the severity of alphavirus infections in *Mavs*^{-/-} mice are conflicting. In RRV infections, Haist et al. (*J Virol* 95(6): e01538-20 (2021) reported that *Mavs*^{-/-} mice infected with RRV developed severe disease and succumbed to infection after five days post-infection. In agreement with the present study, Schulte et al. (*J Exp Med* 207(2):429-42 (2010) reported that *Mavs*^{-/-} mice were not susceptible to CHIKV infection. It is necessary for the authors to discuss their results in light of these reports.

We have now added disease scores at endpoint of mice infected with CHIKV in WT, *ifnar1*^{-/-}, and *Mavs*^{-/-} mice (see **Extended Data Table S5**).

In our studies we infected *Mavs*^{-/-} mice with 1E5 to 1E4 PFU of CHIKV IOL or reporter virus CHIKV IOL-ZsGreen by either subcutaneous (footpad) or intravenous (tail vein) inoculations. In our hands, all the animals infected via footpad infection survived to experimental timepoints, up to the longest time point measured at 60 dpi. However, we did detected mortality in our *Mavs*^{-/-} group infected intravenously. We observed that 3 out of 44 animals (6%) showed signs of lethargy and distress between 10 to 14 dpi and required to be euthanized. In addition, a single animal was found dead at 14 dpi. The remaining animals survived and with monitoring up to 60 dpi.

We have now included and discussed our findings in light of the findings for CHIKV and RRV.

The following changes have been added to the text (L444-449):

*"Mavs^{-/-} mice display different susceptibilities to related arthritogenic alphaviruses^{30,52}. While *Mavs^{-/-}* mice infected with Ross River Virus developed severe disease and succumbed to infection⁵², CHIKV infected mice survived when inoculated subcutaneously³⁰ (**Extended Data Table 5**). Comparative studies addressing the role of PRR in controlling the infection of related arthritogenic alphavirus would be fundamental for expanding our knowledge virus-specific mechanisms of pathogenesis."*

Minor Comments

1. Recently, Langsjoen et al. (Am J Trop Med Hyg 2021 Nov 29;106(1):99-104) have reported heart-specific changes in virus mutational profiles and host gene expression in mice infected with CHIKV. Heart-specific transcriptional changes included differential expression of genes critical for ion transport and muscle contraction. Include this information in the article.

We thank the reviewer for bringing this point to our attention. The study from Langsjoen et al performed bulk RNA-seq on hearts from IFNAR KO mice after CHIKV infection and highlight heart-specific transcriptional changes including differential expression of genes critical for ion transport and muscle contraction. While we agree that the statements by Langsjoen et al are directly related to our study and therefore should be discussed in the light of our findings, it is important to note that the authors themselves openly acknowledge the following limitations:

1. The authors stated in their manuscript that only two animals were used for mock controls in gene expression studies, and therefore their differential gene expression analysis remains preliminary and data should be interpreted with caution.

Statement from Langsjoen et al: "Although these results provide important insights into potential effects of alphavirus infection on the cardiovascular system with remarkable consistency between replicates, only two animals were used for mock controls in gene expression studies".

2. The authors stated that animals in this study were not perfused, therefore data may be contaminated with transcripts from cells in circulation.

Statement from Langsjoen et al: "Furthermore, animals in this study were not perfused prior to organ collection, thus both mutation data, as well as gene expression data, may be contaminated by circulating virus and contaminating host RNAs, respectively".

3. The study does not include a materials and methods section and no code for reproducibility of the analysis is available, and therefore neither we or others can recapitulate the computational analysis using their data set.

Therefore, while we acknowledge the findings of Langsjoen et al regarding the presence of viral RNA in the discussion of our original manuscript (Line: 315-318) and in our current manuscript (L54-55, L89-91 and L367-368), we are hesitant to highlight specific results regarding ion channels and muscle contraction in light of the limitations of the study. However, we leave it to the criteria of the reviewer and the editor to decide whether to expand on these observations in the discussion.

Reviewer #2 (Remarks to the Author):

Review

In this study entitled "Type I interferon and MAVS signaling restricts chikungunya virus heart infection and cardiac damage", Maria G. Noval et al. in their paper propose to explore Chikungunya virus (CHIKV) infection of the heart with the aim of understanding why and how CHIKV can lead to cardiac damage, sometimes fatal. To evaluate CHIKV infection in the heart, the authors used a C57BL/6 mouse model. Using this model, they show that cardiac fibroblasts are the main targets of CHIKV in cardiac tissue and that infection of this tissue induces a strong and robust type I interferon (IFN-I) response. In addition, they observe that loss of IFN-I signaling induces increased replication, dissemination, and apoptosis in cardiac tissue. They also point out that MAVS signaling is essential for CHIKV-clearance. Finally, by using MAVS-deficient mice, they observed the appearance of cardiac damage. The authors conclude that the study provides a new mouse model for CHIKV cardiac infection and mechanistic insight how CHIKV can lead to cardiac damage.

CHIKV is an arthritogenic arbovirus from Togaviridae family and Alphavirus genus, which is responsible for recurring epidemics over the years worldwide. CHIKV is considered an important public health problem because it is endemic in tropical and subtropical regions of the globe. Its transmission occurs by the bite of infected mosquitoes from genus *Aedes* spp., mainly *Aedes aegypti* and *Aedes albopictus*, which are highly extremely adaptable to environment changes, thereby resulting in an efficient spread across the countries and continents. Despite most of CHIKV-infected individuals are symptomatic, less than 15% of infected population do not develop any symptoms. In symptomatic forms CHIKV causes an acute febrile illness characterized by polyarthralgia, polyarthritis, maculopapular rash, and myalgia. Symptoms typically resolve within 1–2 weeks, but about 1–4% of cases result in chronic flaring joint pain that can persist for months or years after infection. In rare cases, neuro-invasive disease and other complications have been observed, leading to fatality.

Indeed, although CHIKV infection is known as a non-deadly disease, atypical and severe acute manifestations can evolve to multiple organ failure and death. Among neurological complications, the most prevalent symptoms seem to be abnormal mental status, headache, focal deficits, and seizures. Other symptoms such as meningoencephalitis, myeloradiculitis, myelitis, myeloneuropathy, external ophthalmoplegia, facial palsy, sensorineural deafness etc. and optic neuritis were described during the recent epidemics. Guillain-Barre syndrome as well as mild hemorrhage, myocarditis, and hepatitis were also reported and are usually observed in both the elderly population and individuals with comorbidities.

Cardiac manifestations (arrhythmias, abnormal echocardiograms and electrocardiograms, myocarditis and death) were also reported in young individuals. Myocarditis and cardiomyopathy manifestation have been in fact reported since 1972 following arbovirus infection by Dengue or Chikungunya viruses. In 2017, Maria Fernandez Alvarez et al reported that cardiovascular manifestation (mainly hypotension, shock and circulatory collapse, Raynaud phenomenon, arrhythmias, murmurs, myocarditis, dilated cardiomyopathy, congestive insufficiency, heart failure) have been noticed in France, India, Sri Lanka, Malaysia, Columbia, Venezuela and USA. Following the Reunion Island outbreak in 2004–2005, an increase in reporting of cardiac related incidents during CHIKV infection occurred, which may indicate newer strains have a higher likelihood of affecting the heart, or a lack of reporting in previous outbreaks. While the majority of reported cases occur around 2005 and later, which might also implicate the A226V mutation, the cases reported in the 1970s, the clustering evident in the phylogenetic trees and statistical analysis indicate that the A226V mutation is not associated with cardiac involvement. Following the Reunion Island outbreak it has been reported an overall outbreak mortality of 10%. Interestingly, heart failure was reported in 15% of cases, myocarditis and pericarditis in 5%, and acute myocardial infarction in 2%; among all deaths, as a result cardio-vascular mortality represented 22% of the total.

Limitations:

There are two questions in this work that are not clearly distinguished and which sometimes suggests that it will answer both questions.

1- How does the Chikungunya virus replicate in the heart: target cells in the cardiac tissue, IFN-I response etc. How is it in most cases eliminated without pathophysiological consequences?

2- Why, in some cases, can we observe damage to the heart? What are the predisposing factors for such damage (knowing that there are several cardiac diseases that do not necessarily have the same biological mechanism and/or variations depending on the target cells, for example, and also that cardiac damages are mostly observed in older adults or individuals with comorbidities)?

In this article, it seems that the question addressed is rather exclusively the first one even if the work carried out gives interesting clues to answer the second question: variation in the type I IFN response according to individuals, SNPs in the coding region of the human MAVS, link between age and decrease sensing through MAVS. Of course, this does not give the same impact to the paper. In addition, there are several weaknesses described in the following points.

Strengths:

This article addresses an important issue: cardiac involvement is not common in isolated episodes, but countries where these viruses have epidemic potential should be alarmed about this condition, especially during epidemics. The mechanism of involvement of cardiac damage is not clearly understood in CHIKV infection. This is because the viruses may directly damage the myocardium or cause a hypersensitivity or autoimmune response. CHIKV-related myocarditis is the most frequently observed cardiac injury and remains the leading cause of death in CHIKV-infected patients. Early diagnosis and appropriate management are necessary to improve patient outcomes. CHIKV-infected individuals should have their cardiac status monitored by echocardiogram and/or electrocardiogram from the time of hospital admission and beyond to detect possible changes in cardiac status and to allow physicians to intervene if complications arise.

First of all, we want to thank the reviewer for recognizing and highlighting both the mechanistic potential and the clinical relevance of this study, as well for the constructive criticism that led us to an improved version of the manuscript. As the reviewer pointed out, atypical cardiac manifestations account for a large proportion of CHIKV-related deaths, and it is therefore paramount to investigate the underlying causes and the potential predisposing factors increasing the probability of CHIKV-associated cardiomyopathies leading to mortality. As stated by the reviewer, understanding the mechanisms and consequences of CHIKV-induced cardiac pathophysiology can be leveraged for early diagnosis, prevention and therapeutic intervention.

We want to also thank the reviewer for recognizing our efforts in addressing the main question of this study: how does CHIKV replicate in the heart? What are the target cells and mechanisms involved? What are the pathophysiological consequences?

Here, we show that there is direct infection of cardiac tissue by CHIKV virus. However, one of the critical findings of our work is that such infection comprises actively replicating virus within the cardiac tissue. Since in this study perfusion was performed prior to downstream experimental handling and analysis, it allowed us to rule out the pervasive confounder present in other mouse model studies (PMID:31291581, 34844209), which the presence of viral particles or viral RNA in the blood stream. It is worth noting that the highly irrigated cardiac tissue can be particularly prone to this confounding factor. To further strengthen our observation of cardiac tissue infection, we orthogonally validated this observation by applying two complementary approaches: measurements of the ratio of sub-genomic to genomic viral RNA, and the use of a Zs-Green CHIKV reporter virus that allows to detect exclusively cells in which active viral replication is taking place. Therefore, we demonstrate not only that cardiac tissue is a bona fide target of CHIKV infection, but also that such infection is composed by actively replicating CHIKV.

Beyond demonstrating active replication, our study provides insight into the cellular target of CHIKV infection. While as the reviewer pointed out our initial assessment of the expression of putative receptors for CHIKV infection might induce bias, our immunostaining experiments were performed targeting all the major cell types in the heart: cardiac myocytes, cardiac fibroblasts, cardiac endothelial cells, cardiac smooth muscle cells and leukocytes. Therefore, while our initial observation might induce bias for the readers, our experimental results point out to cardiac fibroblasts

being the primary cellular target of CHIKV infection in comparison to all other cell types tested (**Figure 1e-f, Extended Data Fig. 1d-e, see below for point-by-point**).

Beyond defining the presence of active replication and the main cellular target of the infection, we provide robust evidence of the IFN-I pathway being a critical component required for CHIKV clearance from the heart, but furthermore we found a specific dependence on the MAVS signaling, which shows delayed clearance of CHIKV from the heart and increased pathological consequences of CHIKV infection. Thus, our work defines mechanistic dependencies for clearance of CHIKV infection from the cardiac tissue.

As the reviewer well pointed out, these findings have potential for translation into defining the predisposing factors of cardiac complications in CHIKV-infected patients. Regarding the second question stated by the reviewer: **Why, in some cases, can we observe damage to the heart? What are the predisposing factors for such damage?**

This is indeed a critical question that requires an epidemiological survey paired with either targeted or whole-genome sequencing efforts to identify genetic variants of risk in the CHIKV-infected population. As stated by the reviewer, the results from our work give interesting clues to answer this question. Indeed, we discuss future directions and potential studies that can be carried out either by our group or other groups that might be interested and made aware of our findings after publication of the present work. However, as the reviewer recognizes, the epidemiological nature of the second question goes beyond the efforts made here to address the more fundamental but equally critical question of what are the underlying mechanisms, target cells and pathophysiological consequences of CHIKV infection of cardiac tissue. We have now included further discussion of this point in the **Discussion** section and have clarified the main text to avoid confusion regarding the main point of the present work. Please **see below for a point-by-point** answer to the reviewer's concerns.

Point-by-point response:

Major comments

1 Cardiac tissue is infected by CHIKV in MXRA8-dependent manner

- Figure 1 a. To explore the presence of actively replicating virus in the heart the authors detected and quantified the CHIKV RNA by RT-qPCR and infectious particles by tissue culture infection dose from perfused hearts.

As the mice were infected either intradermally, via natural transmission or intravenously, it is difficult to conclude at this stage that it is indeed an active replication and not a simple dissemination, the infectious particles recovered as well as the genomes resulting then only from the adhesion to the external structures of the cells and not from an active replication. The mode of infection is an important point here. Indeed, the intradermal route can mimic the natural mode of transmission by mosquito bite, while the intravenous route not only bypasses the requirement for initial dissemination from a primary infection site or induces uneven replication of the virus in tissues near and distal to the inoculation site, this route also likely has significant effects on the immune response to this viral infection.

- Figure 1b. However, to assess replication the authors use an RNAseq approach to measure the ratio of CHIKV genome to sub-genomic RNAs after intravenous inoculation which indeed allows to highlight an active replication but which is perhaps not the reflection of the physiological reality. It would be useful to perform the same type of experiments using the intradermal route and the natural infection route to demonstrate that there is indeed active replication of CHIKV in cardiac tissue in an infected mouse model.

We thank this reviewer for bringing up two extremely important points, that we have now addressed as follows:

- Point (1) How can we distinguish that the viral genomes as well as infectious particles reported in our study are not a result of an artifact due to the adhesion of viral particles to the external structures of the cells in the tissue?

Indeed, the first point represents a limitation of previous studies of CHIKV infection of cardiac tissue using adult mice (**PMID: 31291581, 34844209**), including our own Cell Reports paper from 2019 (**PMID: 31291581**), in which CHIKV tropism to the heart was inferred by measuring infectious particles in non-perfused hearts. Indeed, the lack

of perfusion in these studies results in the inclusion of a pervasive confounder when attempting to measure both presence of viral particles and viral RNA.

In the present study, we went to great lengths to include an initial perfusion step in our experimental protocols, which allows to prevent this confounder in all downstream measurements and analysis. To address the reviewer's concern regarding the inoculation routes, we have now performed the following experiments to both orthogonally validate CHIKV active replication via complementary experiments (i.e. qPCR or Zs-Green construct expressed only in actively replicating virus) and performed these experiments via both intravenous as well as subcutaneous injection.

As shown below, we have now added quantifications of the gRNA:sgRNA ratio for both intravenous and subcutaneous inoculation routes. This was measured by qPCR with specific primers targeting nsp4 (as a surrogate of gRNA) and E1 (for sgRNA) as previously described (PMID: 31465513 and PMID: 24131709). We measured active replication at 2, 5, and 10 dpi. Control animals were inoculated with the same virus preparations but heat-inactivated (HI). Data were scored as active replication and represents the ratio between E1 and nsP4 targets. These data have been added to Fig. 1b and to the text (L100-112):

“To evaluate if the CHIKV viral particles detected in the heart were actively replicating, we measured the ratio between CHIKV nsp4 and E1 transcripts by RT-qPCR as a surrogate of genomic and sub-genomic RNA, respectively. As a non-replicative control, we infected mice with heat-inactivated (HI) CHIKV. While no active replication was detected for the HI control, CHIKV active replication in the heart was detected for both subcutaneous and intravenous inoculation routes (Fig. 1b). For intravenously infected mice active replication was detected at 2 dpi (R [Ratio E1/nsp4] = 3.80, p = 0.04) and 5 dpi (R = 8.11, p = 0.008), while for subcutaneously infected mice active replication was detected at 5 dpi (R = 3.88, p = 0.008). The difference in kinetics between inoculation routes is consistent with the requirement of subcutaneous inoculation to reach viremia in order to disseminate and infect cardiac tissue, while the intravenous route bypasses this stage of the infection. These results demonstrate that cardiac tissue is a direct target of CHIKV infection in immunocompetent mice.”

Unfortunately, for the mosquito bite inoculation route, we were not able to perform these quantifications due to the low RNA quality from the mosquito bite experiments.

To further support these data through the use of a complementary approach for measuring active viral replication, we infected mice subcutaneously with a reporter virus: CHIKV-ZsGreen. The CHIKV-ZsGreen virus is a reporter of CHIKV active replication, since the expression is regulated by CHIKV sub-genomic promoter, and therefore, ZsGreen will only be detected after active replication of the CHIKV genome. In addition to measuring only replication, the CHIKV-ZsGreen reporter allowed us to image the infected tissue, adding another layer of validation for direct replication of CHIKV in the heart, and further rule out the potential confounder of bloodstream viral load. Indeed, we found that CHIKV actively replicates in the cardiac tissue when performing subcutaneous injection. This data has been added to Extended Data Fig 1b and to the text (L113-118):

“To determine the localization of CHIKV infection within cardiac tissue, we used a reporter CHIKV that expresses the ZsGreen fluorescent protein (CHIKV-ZsGreen) exclusively under active viral replication¹⁷. Mice were infected with CHIKV-ZsGreen and cardiac tissue was harvested at 2 dpi. We observed multiple ZsGreen positive foci corresponding with CHIKV-infected cells for mice infected both intravenously (Fig. 1c) as well as subcutaneously (Extended Data Fig 1b).”

Finally, to further address the reviewer's point, we performed mosquito-bite infection using the CHIKV-ZsGreen reporter, that as stated above is only detected during active CHIKV viral replication. Indeed, we do see active replication in cardiac tissue after infection through a mosquito bite in IFNAR1 KO mice after three days post infection, as shown in reviewer Figure A.

Figure A. IFNAR1 KO mice were exposed to the bite of CHIKV infected mosquitoes and body weight loss was monitored daily for 3 days post infection. Animals were sacrificed and cardiac tissue was perfused with 30 mL of cold PBS. Active replication in cardiac tissue was assessed by looking at the microscope for the presence of CHIKV positive cells. Right panel: body weight loss for CHIKV mice (mice bitten by CHIKV infected mosquito) or control condition (mice bitten by non-infected mosquito).

Altogether, considering that (i) by perfusing cardiac tissue we remove all blood from the heart and therefore all potential confounding signal coming from viremia, (ii) we show that levels of sgRNA are significantly higher than levels of gRNA as quantified by qPCR for both subcutaneous and intravenous inoculation routes and (iii) finally, that we observe expression of ZsGreen virus in cardiac tissue independently of the inoculation route, we can confidently state that cardiac tissue is a target of CHIKV infection and that there is active CHIKV replication in the heart independently of the inoculation route.

- (Point 2): The potential effects of the different inoculation routes on the immune response mounted against CHIKV.

Once again, we want to thank the reviewer for making this important point regarding the significant effects of the infection route on the immune response to CHIKV infection. In light of this comment, we have now expanded all our studies regarding the pathogenesis of the MAVS KO mice to include subcutaneous inoculation. Indeed, we found that viral myocarditis and chronic vasculitis is observed up to 31 to 60 dpi in this model independently of the inoculation route. Therefore, these results support the observation that the development of pathogenesis in cardiovascular tissue is independent of the inoculation route. These results have now been added to **Fig 6** and **Extended Data Fig 6** and **Extended Data Table S1**.

The following modifications have been added to the text (L334-357):

“Chronic inflammation in myocardium and vasculature in CHIKV-infected *Mavs*^{-/-} mice persist for several months. Given the focal myocarditis and the major vessel vasculitis observed at 10 dpi in *Mavs*^{-/-} mice, we asked whether this inflammation can result in chronic inflammation potentially leading to long-term tissue damage. For that purpose, *Mavs*^{-/-} mice were infected intravenously or subcutaneously with CHIKV, and cardiac tissue was collected for histopathological analysis at 2 weeks, 1 month, or 2 months post-infection. Tissue was stained with H&E, and mononuclear cell infiltrates were evaluated for CD3 and CD11b markers by IHC. We found that a fraction of *Mavs*^{-/-} mice developed focal and multifocal myocarditis accentuated around the vessels in mice infected by either intravenous or subcutaneous inoculation routes (**Fig. 6a-b** and **Extended Data Table 1**). These inflammatory foci were characterized by CD11b⁺, and CD3⁺ cell infiltrates and persisted for up to 60 dpi or up to 31 dpi in mice infected intravenously or subcutaneously, respectively (**Fig.6b**, and **Extended Data Fig. 6a-b** and **6e**). Interestingly, we did not detect changes in the serum levels of cTnT (**Fig. 6c**), suggesting that despite myocarditis there is no significant cardiac myocyte damage in this model. Of note, cardiac tissue of an infected *Mavs*^{-/-} mice that succumbed to CHIKV infection at 15 dpi (**Extended Data Table 5**), showed substantial tissue damage featured by patchy myocyte dystrophic calcification affecting single myocytes, and focal myocarditis composed by inflammatory infiltrates of CD11b⁺ and CD3⁺ cells, with no signs of fibrosis (**Extended Data Fig. 7**). Strikingly, major vasculitis characterized by transmural CD3⁺ and CD11b⁺ infiltrates was detected in the majority of the *Mavs*^{-/-} infected mice up to 60 dpi independently of the inoculation route (**Fig. 6d**; **Extended Fig. 6c-d**, and **Extended Data Table 1**). Altogether, these results demonstrate that persistence of CHIKV infection in cardiac tissue leads to chronic inflammation characterized by focal or multifocal myocarditis and major vessel vasculitis.”

- Figure 1C. The authors then discuss the use of the MXRA8 receptor by CHIKV to infect cardiac tissue by hypothesizing that MXRA8 is the only one receptor of CHIKV (residual CHIKV infection is also detectable in vitro and in vivo in the absence of *Mxra8*).

CHIKV has a wide cellular and tissue tropism which may be attributed to use of ubiquitously expressed molecules or several cell specific factors for entry. Multiple attachment factors and putative receptors for CHIKV have been documented. For instance, prohibitins, glycosaminoglycans, phosphatidylserine (PtdSer)-mediated virus entry-enhancing receptors (PVEERs), and MXRA8 are host factors in mammalian cells. Interaction with the cell surface molecules is mediated by the viral E2 glycoprotein, whose domain B contains receptor binding sites. For MXRA8, a recently identified receptor for several alphaviruses E1 is additionally important as MXRA8 engages amino acid residues at the E1 and E2 glycoprotein heterodimer interface. TIM-1 and Axl receptor tyrosine kinase (Axl) are PVEERs associated with enhanced cell entry by enveloped viruses. Using CHIKV glycoprotein based pseudoviruses, the TIM family of proteins and Axl were shown to enhance infection.

Using *Mxra8* deficient mouse model the authors observed that the absence of functional MXRA8 results in a

significant reduction of RNA in cardiac tissue as well as the induction of Isg15 in *Mxra8*^{-/-} infected mouse following intradermal inoculation (using RT-qPCR approach as described #1 Fig. 1a). The authors stated that the RNA levels in serum remained unaltered excluding the possibility that the observed decrease in CHIKV infection in *Mxra8*^{-/-} infected is driven by reduced viremia. Finally, using *Mxra8*^{-/-} and WT mice and an intravenous inoculation route to introduce a fluorescent CHIKV virus, the authors found reduced CHIKV infection in *Mxra8*^{-/-} cardiac tissue but also a significant reduced amount of viral RNA in serum of *Mxra8*^{-/-} mice.

First, it would have been useful to quantify the expression of MXRA8 in cardiac tissue and even better to distinguish the expression of this receptor in the different tissues of the heart (and not use published data from transcriptomes of 20 organs of non-infected C57BL/6JN), moreover, it would have been better to quantify the active replication as well in the first series of experiments.

Finally, the difference observed between the two sets of experiments in the amount of viral genome present in the serum raises questions and does not allow conclusions about CHIKV active replication in the heart. The only difference here is related to the mode of inoculation: intradermal versus intravenous, and does not allow to conclude on the receptors (other than MXRA8) used by CHIKV to infect the cardiac tissue. The more efficient infection of a particular heart tissue could explain the differences in cardiac pathology observed in some CHIKV infection events. In this context, it would be relevant to test other potential receptors.

We thank the reviewer for pointing this out. For our study, the goal of using MXRA8 deficient mice was to further confirm our findings of active replication in cardiac tissue, and not necessarily to identify receptors implicated in CHIKV heart infection. We want to underscore that while further characterization of the receptors involved in CHIKV infection are valuable, the focus of the current study is centered in understanding how cardiac tissue responds to CHIKV infection, and identify mechanisms implicated in development of CHIKV-mediated heart disease. Therefore, considering that all the reviewers' comments and the conflicting data regarding CHIKV receptors, we decided to remove these data from our study.

2 Cardiac fibroblast are the primary cellular target of CHIKV infection in infected hearts.

Since the authors conclude that CHIKV infect cardiac tissue and that this infection is in part dependent of MXRA8, they interrogated cell type specific expression levels of the *Mxra8* receptor in cardiac tissue by leveraging the single-cell dataset from Tabula Muris. They observed that two main factors critical for CHIKV infection are co-expressed in 75% of cardiac fibroblasts "suggesting that cardiac fibroblasts may be the primary cardiac cell type infected. This argument, which excludes any other receptor, induces a bias. Of course, these fibroblasts may represent a target, but they are not necessarily the main and/or the only target.

We thank the reviewer for this observation. As stated above, our initial analysis of the co-expression of MXRA8 and FHL1 suggested the cardiac fibroblasts as the main cellular target. Regardless of this initial observation, we performed immunostaining and measurements of CHIKV-infected cells using cell markers for all major cell types in the heart (cardiac myocytes, cardiac fibroblasts, cardiac endothelial cells, cardiac smooth muscle cells and leukocytes) and therefore our downstream testing of this initial observation is unbiased. In our study, we developed panels to characterize all the major cell types of the heart by immunofluorescence. We performed the selection of the markers for each cardiac cell type based on previous studies on cardiac cells (PMID: 15621032, PMID: 6190091), and we leveraged the *Tabula Muris* dataset to verify that the markers selected allowed us to differentiate between cell types (Extended Data Fig 1c-d).

- cardiac fibroblasts (Vimentin+ aSMA- cells CD31- CD45-),
 - cardiomyocytes (cTnT + cells),
 - cardiac endothelial cells (CD31 + cells),
 - cardiac smooth muscle cells (aSMA + cells),
 - and cardiac leukocytes (CD45+ cells).

To confidently and in an unbiased manner define the main cell type targeted by CHIKV infection, we performed colocalization analysis between virus and marker/s, followed by subsequent quantifications. We performed this analysis by looking at three to four independent fields (2x2 tiles at 20X magnification) for three independent infected

animals. Our quantifications demonstrate that 78.52 % of the infected cells in cardiac tissue colocalized with cardiac fibroblasts markers (Vimentin+ α SMA- cells CD31- CD45-), and only less than 4.39% of infected cells colocalize with cardiomyocytes (cTnT), cardiac endothelial cells (CD31+, CD45-), or cardiac leukocyte markers (CD45+). Indeed, cardiac fibroblasts co-localization with CHIKV infection was significantly higher than all other measured cell types (CF vs CM: $p = 0.0008$; CF vs CEC: $p = 0.0067$; CF vs CL: $p = 0.0336$ and CF vs SMCs: $p < 0.0001$).

However, to prevent any bias in the readers, we have clarified Figure 1 and now removed the panel associated with the receptor's co-expression from the Tabula Muris (**original Figure 1**) from the current manuscript, to comply with the reviewer's suggestion.

Finally, in this new version of the paper, we expanded our study to human cardiac primary cells (**Figure 1g**), and demonstrated that CHIKV can infect efficiently human cardiac fibroblast. Interestingly, in contrast to what we observed in our *in vivo* experiments, human cardiac myocytes in culture can sustain active viral replication (**Figure 1g**, open circles). However, careful interpretation of these result must be taken. Cardiac myocytes in culture lose specific features that can potentially impact susceptibility to CHIKV infection. For example, while cardiac myocytes in tissue are characterized for being multinucleated and quiescent *in vivo*, the *in vitro* culture reverts these phenotypes, resulting in mono-nucleated and proliferating cells (**PMID:9798514**). An additional confounder of these results is that *in vitro* infections bypass all the major barriers during *in vivo* infections (i.e. dissemination, immune system, etc). Therefore, while these results support the role of cardiac fibroblasts as the main target of CHIKV, for which we show supporting evidence *in vivo*, the observation regarding *in vitro* infection of cardiomyocytes must be taken carefully. For this reason, in future studies, we are planning to work with human heart tissue explants which may recapitulate a more physiologically relevant scenario. However, the latter is out of the scope of this study. The following modifications have been added to the text (L133-162):

“To identify the specific cell types infected by CHIKV in cardiac tissue, hearts were harvested at 2 dpi, fixed, and processed to obtain serial cryosections followed by staining with markers for specific cardiac cell types: CD31 (endothelial cells), CD45 (leukocytes), α SMA (smooth muscle cells, pericytes, and myofibroblasts), cTnT (cardiomyocytes), and vimentin. Of note, vimentin is expressed at high levels in cardiac fibroblasts²² but can also be expressed in smooth muscle cells, endothelial cells, and macrophages²³⁻²⁵. Therefore, we identified cardiac fibroblasts cells as vimentin⁺/ α SMA⁻/CD31⁻/CD45⁻ populations, a combination that we orthogonally validated via single-cell RNA expression from the *Tabula Muris*²⁶ (**Extended Data Fig. 1c**). By using fluorescence microscopy and co-localization analysis with the different cell type markers, we found that the 78.5% of CHIKV-ZsGreen signal colocalized with vimentin⁺/ α SMA⁻ cells (**Fig. 1e-f, and Extended Data Fig. 1d-e**). This result, together with the reduced colocalization between CHIKV-ZsGreen signal and markers for endothelial cells (CD31⁺; 7.3%), leukocytes (CD45⁺; 7.2%), smooth muscle/pericytes/myofibroblasts (α SMA⁺; 1%), and cardiomyocytes (cTnT⁺; 4.4%; **Fig. 1f; Extended Data Fig. 1e**) demonstrate that the cardiac fibroblasts are the primary target of CHIKV infection of cardiac tissue *in vivo*.

Finally, we sought to determine whether human cardiac cells show similar susceptibilities to CHIKV-infection than those observed in mice. For that purpose, we evaluated the capacity of CHIKV to infect human primary cardiac cells *in vitro*. We infected human primary cardiac fibroblasts (hCF), human cardiac myocytes (hCM) and human cardiac microvasculature endothelial cells (hCE) at an MOI of 0.1 and measured the production of infectious particles. We found that CHIKV can efficiently infect hCFs (**Fig. 1g, black circles**), but no hCEs (**Fig. 1g, squares**), supporting our *in vivo* observations in mice. Interestingly, hCMs in culture produced CHIKV infectious particles at similar levels than the observed for hCFs (**Fig. 1g, open circles**). However, careful interpretation of this result must be taken, since hCMs in culture loose specific features that can potentially impact susceptibility to CHIKV infection. For example, while cardiac myocytes in tissue are characterized for being multinucleated and quiescent *in vivo*, the *in vitro* culture revert these phenotypes, resulting in mono-nucleated and proliferating cells²⁷. Altogether, our results demonstrate that CHIKV actively replicates in cardiac tissue with cardiac fibroblasts being the primary cellular target of CHIKV infection in immunocompetent mice.”

And to the discussion (L456-459):

“Another alternative is that human and murine cardiac myocytes have different susceptibilities to CHIKV infection. Further studies using human heart tissue explants or human cardiomyocyte organoids are required to evaluate the relative susceptibilities of human cardiac myocytes to CHIKV infection.”

One symptom of CHIKV infection that is found in all age groups has been myocarditis. Myocarditis refers to

inflammation of heart tissue, and it can be caused by viral infection, thus becoming viral myocarditis. For viral myocarditis to occur, cardiomyocytes must be directly infected, resulting in cell death. There is evidence for cardiac infection of cardiomyocytes by CHIKV, including via MXRA8 (low expression) or prohibitin (medium expression).

As the reviewer points out, viral myocarditis is one of the recurrent pathologies associated with CHIKV cardiac manifestations. However, while most cardiotropic viruses are implicated in direct infection of cardiomyocytes (i.e. coxsackie virus, influenza virus, SARS-CoV-2), viral myocarditis has been described for non-cardiotropic viruses, in which cardiac injury is associated to a cytokine storm or cellular immune response due to molecular mimicry (PMID: 33046850).

We will appreciate if the reviewer provides the references for this statement “There is evidence for cardiac infection of cardiomyocytes by CHIKV, including via MXRA8 (low expression) or prohibitin (medium expression)”, since we are only able to find two studies reporting the presence of CHIKV antigen in human post-mortem cardiac tissue samples, and only one provides information on the cell type/s harboring CHIKV antigen (PMID: 18679124, PMID:32615591). In the first paper, viral inclusion was detected in myocardial cells (without distinction of myocytes or fibroblast nature). In the latter, viral antigen was detected in fibroblast and interstitial connective tissue, vessels or vascular endothelium, epicardial adipose tissue, surrounding nerves, and within vascular lumens of individuals that succumb to CHIKV infection (PMID:32615591). This raises the possibility that CHIKV-induced myocarditis may be occurring not necessarily through direct infection of cardiomyocytes.

Indeed, our results are in agreement with the human data, supporting the relevance of mice models for mechanistic studies of CHIKV cardiac pathogenesis. We demonstrated that CHIKV infects mainly cardiac fibroblasts and cardiomyocytes represent ~ 5% of the infected cells, even in the absence of type-I interferon signaling (See Fig. 3f and Extended Data Fig.4c). In addition, while our MAVS KO develops focal or multifocal myocarditis (Fig 5 & 6, and Extended Data Fig 6a and 6b, and Extended Data Table S1) between 10 and 30 dpi, we do not detect changes in the serum levels of cardiac Troponin T (Fig. 6c), supporting that there is no cardiomyocyte damage in this model.

The following modifications have been added to the text (L450-459):

“We demonstrated that CHIKV infection led to focal and multifocal myocarditis in *Mavs*^{-/-} mice. Yet, serum levels of cTnT remained unaltered, supporting the absence of cardiac myocyte damage. We found that CHIKV does not infect cardiac myocytes in mice, consistent with the lack of evidence of cardiac myocyte infection in human post-mortem cardiac tissue samples¹². However, elevated cardiac troponins have been observed in individuals with CHIKV-associated cardiac manifestations^{6,7,13,16}, raising the possibility that under certain conditions CHIKV can induce cardiac myocyte damage. Another alternative is that human and murine cardiac myocytes have different susceptibilities to CHIKV infection. Further studies using human heart tissue explants or human cardiomyocyte organoids are required to evaluate the relative susceptibilities of human cardiac myocytes to CHIKV infection.”

3 IFN-I signaling pathway restrains CHIKV cardiac infection, preventing viral spreading and cardiac tissue damage.

Rose M. Langsjoen, Yiyang Zhou, Richard J. Holcomb, and Andrew L. Routh recently published a paper on the infection of IFN- α R^{-/-} mice to investigate heart-specific changes on host gene expression. They found that CHIKV replicates in the hearts of immunodeficient mice with implications for cardiac pathologies suggesting that an IFN-I response is important to control CHIKV infection and prevent cardiac tissue damage. In consequence, the results presented here lose their originality.

We thank the reviewer for bringing this point into our consideration. However, we believe the results described in this section are original since none of our findings have been previously described by others. In our manuscript, we demonstrated that a local type I interferon response is fundamental to controlling CHIKV infection in cardiac tissue. Particularly, using a IFNAR KO mice we demonstrated that in the absence of a local type I interferon response:

- 1- CHIKV infection is significantly increased compared to WT animals by looking at both infectious particles ($p=0.001$) as well as viral RNA ($p=0.0025$) after perfusion (**Extended Data Fig. 4b**).
- 2- Using our CHIKV reporter virus we demonstrate that in the absence of a local IFN-I response CHIKV infection spread in cardiac tissue (**Fig. 3e**).
- 3- We showed that cardiac fibroblast (~40%) and cardiac endothelial cells (~18%) represent the main targets of the infection in the absence of IFNAR receptor (**Fig. 3f and Extended Data Fig 4c**).
- 4- And finally, we demonstrate for the first time in this model cardiac tissue damage by performing immunohistochemistry against cleaved-caspase 3. Immunohistochemical analysis has been done in 5 independent animals for both WT and IFNAR KO mice (**Fig. 3g**).

While the title of the paper from Langsjoen et al suggests that the study demonstrate that CHIKV replicates in the heart of immunodeficient mice and speculate about the potential implications in cardiac tissue damage, the authors do not directly demonstrate these findings. Indeed, the authors in the discussion acknowledged the limitations of their study as follows:

“Although these results provide important insights into potential effects of alphavirus infection on the cardiovascular system with remarkable consistency between replicates, only two animals were used for mock controls in gene expression studies. **Therefore, differential gene expression analysis remains preliminary and should be interpreted with caution**. Furthermore, **animals in this study were not perfused prior to organ collection, thus both mutation data, as well as gene expression data, may be contaminated by circulating virus and contaminating host RNAs, respectively**”.

As stated before, the first caveat of the study is that the authors do not perfuse the hearts, and as this reviewer pointed out there is no way to distinguish “that the tissue itself is being infected and the infectious particles are not a result of an artifact due to the adhesion of viral particles to the external structures of the cells in the tissue without perfusion”. Moreover, in this study, the authors conclude that CHIKV infects the heart by detecting viral RNA in their RNA seq experiments, and by identifying minority variants that seem to be heart-specific. While this last point can support that there is some tissue-specific viral replication, the authors are not showing (i) the production of viral particles, (ii) which cells are being infected, (iii) how this infection compares to a WT infection, or (iv) how the loss of IFN-I promotes viral spreading. Moreover, two years before the authors published this paper, our own group demonstrated that heart from IFNAR KO mice can harbor CHIKV infectious particles (Noval MG, et al, Cell Reports 2019, **PMID: 31291581**). However, because in that study perfusion was not performed, we were not able to conclude that CHIKV can actively replicate in the heart.

In addition, based solely on RNA-seq data the authors conclude tissue damage. As the reviewer stated regarding active viral replication, tissue damage should be also demonstrated through histopathological analysis and not inferred from sequencing data. One example of the importance of additional orthogonal validation of RNA-seq findings can be exemplified by our own RNA-seq data coupled with histopathology analysis for immunocompetent mice. As we mentioned in the discussion of the original manuscript (L343-348):

“Intriguingly, using the KEGG datasets we observed upregulation of the viral myocarditis pathway and downregulation of cardiac tissue contractility pathway at 5 dpi (**Extended Data Fig. 4d-g; Extended Data Table 4**). However, we did not observe any sign of cardiac tissue damage by histopathology (**Extended Data Table S1**), suggesting that CHIKV infection is triggering transcriptional programs involved in cardiac dysfunction that are not sufficient to result in overt phenotypes of cardiac damage.”

By all the stated above, we believe our results are novel and fundamental to support the conclusions on how IFN-I signaling pathway restrains CHIKV cardiac infection, preventing viral spreading and cardiac tissue damage.

However, to further support the findings of the present study, particularly in demonstrating the importance of IFN-I signaling in controlling CHIKV infection in cardiac fibroblasts, we performed additional experiments. We addressed the requirement of type I interferon signaling in human primary cardiac fibroblasts by studying the dependency of CHIKV infection in the presence of an inhibitor of the JAK1/JAK2 pathway, as well as by evaluating the inhibitory potential of IFN α and β . We found that in human primary cardiac fibroblasts, inhibition of JAK1/JAK2, a critical component of the IFN-I pathway, results in increased infection. In addition, to strengthen our findings related to IFN-I effects on CHIKV infection of cardiac fibroblasts, we performed a dose-response experiment incubating

primary human cardiac fibroblasts with increasing concentrations of either IFN alpha or IFN beta. Indeed, increasing doses of either IFNalpha or IFNbeta resulted in reduced infection of primary human cardiac fibroblasts. These results demonstrate that IFN-I is fundamental to controlling CHIKV in human cardiac fibroblasts, and further support our findings in mice. These results have now been added to **Fig 3c-d** and the text (L239-253):

“To determine if human cardiac cells respond to CHIKV infection similarly to murine cardiac tissue, we evaluated IFN-I signaling in hCFs infected with CHIKV. We evaluated this by: (i) measuring the capacity of hCFs to generate a robust IFN-I response after poly(I:C) stimulation and CHIKV infection; (ii) measuring the susceptibility of these cells to CHIKV infection in the absence of IFN-I signaling using the JAK1/JAK2 inhibitor ruxolitinib (Rux); and (iii) inducing a refractory state upon increasing doses of IFN-I pre-treatments. Indeed, we found that hCFs efficiently respond to poly(I:C) stimulation, resulting in ~ 80% of the monolayer expressing ISG15 (**Fig. 3c, left panel**). In agreement with our *in vivo* data, we found that infected hCFs responded to CHIKV infection by producing significant levels of ISG15 (**Fig. 3c, left panel**). In addition, Rux treatment significantly increased the percentage of CHIKV infected cells (**Fig. 3c, right panel**), and correlated with a reduction of ISG15 production (**Fig. 3c, left panel**). Furthermore, we found that exogenous addition of either IFN α or IFN β prevented CHIKV infection in a dose response manner, with IFN β showing increased blocking capacity (**Fig 3d**). These results support that, similarly to murine cardiac tissue, IFN-I signaling is critical in controlling CHIKV infection in human cardiac fibroblasts.”

Reviewer #3 (Remarks to the Author):

Noval et al. use mouse models to evaluate the capacity of Chikungunya virus (CHIKV) to infect heart tissue. The rationale is that CHIKV infection is associated with cardiomyopathies in humans. The experimental approach taken is straightforward. C57BL6 mice are infected by various routes and, independent of route, CHIKV is found to infect the heart. Cardiac fibroblasts are demonstrated to be the main target of infection in the heart. The use of CHIKV receptor (MXRA8) deficient mice demonstrates that heart infection requires MXRA8. The wildtype mice do not develop heart damage and clear the virus after a few days. This clearance is accompanied by an interferon (IFN) response. Elimination of the IFN response by knockout of the IFN alpha receptor (IFNAR) or IFN inducing signaling molecule MAVS prolongs infection of the heart and results in damage to the heart.

Overall, the experiments appear to be well performed and the manuscript is clearly written except for several places where the wrong figure panel is referred to in the text. The topic of study is of interest, given its potential relevance to human disease.

Limitations of the study include its novelty; a previous study reported CHIKV infection of IFNAR knockout mouse heart (Langsjoen et al. AJTMH 2021, 10.4269/ajtmh.21-0719). Infection of wildtype mouse heart has also been reported (Clavarino Plos Pathogens 2012, <https://doi.org/10.1371/journal.ppat.1002708>). The latter study also demonstrated that knockout in neonatal mice of GADD4, which plays roles in the unfolded protein and innate immune responses, results in significant susceptibility to CHIKV, including increased replication in the heart. Further, while the MAVS knockout model may prove useful for further characterization of CHIKV induced heart disease, it is not clear to what extent defective IFN responses contribute to human CHIKV-induced heart dysfunction. Overall, therefore, the impact of this study is likely to be moderate.

We thank the reviewer for recognize the importance of this study and for the constructive criticism regarding novelty which have helped us to better structure this new version of our study, and several new experiments and text modifications have been added to this new version of the manuscript.

While recent work started to appreciate the importance of atypical manifestations associated with arboviral infections, a mechanistic understanding of how the virus interact with cardiac tissue, how heart respond to CHIKV infection and how CHIKV infection can lead to heart disease was lacking. In the present work, we leveraged different mouse models and have now added human primary cells to define mechanisms implicated in CHIKV cardiac tissue infection, and to identify host factors involved in clearance of CHIKV infection and tissue damage. To stress the novelty of our work, these are the specific findings that were not defined prior this study:

- 1- CHIKV actively replicates in cardiac tissue and targets cardiac fibroblasts. In this study, we demonstrated using multiple complementary approaches that CHIKV can actively replicate, and generate infectious particles in the myocardium, atrium, vessels and valves of immunocompetent WT mice, and we demonstrate that the primary target of CHIKV infection in the heart corresponds to cardiac fibroblasts.

As per the reviewer comment, the study from Clavarino *et al* (PMID: 22615568) used for their experiments 12 days-old mice neonate mice, which is not a fully immunocompetent model. While this study made a fundamental contribution in demonstrating the role of GADD34 in CHIKV biology, it does not directly demonstrate active viral replication in cardiac tissue of a fully immunocompetent model.

However, we acknowledge the reviewer's comment, and recognize that in the original version of our manuscript the introduction lacked a section explaining the state-of-the-art knowledge about animal models and CHIKV cardiac infection, and most importantly which questions remained unanswered. For that reason, we have now included a modified section to the introduction as follows (L54-58):

“While previous studies reported a link between CHIKV and infection of cardiac tissue in animal models¹⁷⁻²¹, several questions remain unanswered: (i) is cardiac tissue a direct target of CHIKV infection and a site of active replication in immunocompetent hosts?; (ii) what are the mechanisms of CHIKV-cardiac tissue interaction?; (iii) how does the heart respond to CHIKV infection?; and (iv) how does CHIKV infection lead to heart disease?.”

Finally, in this new version the following modifications have been added to **Fig 1**:

- We added active replication measurements looking at genomic and sub-genomic ratio for subcutaneous inoculation route (**Fig 1b**)
- We identified a difference in viral RNA clearance between the top and the bottom part of the heart, demonstrating that regions containing the atria and vessels can harbor viral RNA for a longer period of time (**See Fig. 1d**). This result strongly suggests that cardiac tissue is not equally susceptible to CHIKV infection.
- We show that the myocardium as well as vessels, valves, and atria can harbor active viral replication (**See Fig. 1c**).
- In order to further strengthen our findings, we have now expanded our studies to human primary cardiac cells, including cardiac fibroblasts, microvasculature cardiac endothelial cells, and cardiomyocytes. Infection of the cardiac fibroblasts as well as the cardiomyocyte cells *in vitro* was observed when performing growth curves at a multiplicity of infection of 0.1. (**See Fig. 1f**)
- We have removed all data related receptor usage on CHIKV cardiac infections from this study. Two reviewers brought very important points about the requirement of other receptors for infection. While this is a very important point, it is not within the scope of this study to define all the receptors involved in CHIKV infection of the heart. Therefore, we decided to remove all data related to Mxra8 KO animals.

2- Infected hearts clear CHIKV infection without inducing cardiac damage in WT mice. We evaluated the kinetics of CHIKV-clearance from cardiac tissue in adult immunocompetent animals and demonstrate that cardiac tissue control CHIKV infection by triggering a transient inflammatory response (characterize by the increase of pro-inflammatory cytokines such as CCL2, CCL5, IL6, and TNF α at 2 dpi), and a local type-I interferon response – in both infected and non-infected cells, that results in viral clearance from the myocardium without tissue damage. These results are novel and provide a detailed insight into how cardiac tissue can control CHIKV infection in immunocompetent conditions.

3- Local type-I interferon response in cardiac tissue is essential to control CHIKV infection and prevent tissue damage. In our study, we demonstrated that a local type I interferon response is fundamental to controlling CHIKV infection in cardiac tissue. Particularly, using IFNAR KO mice we demonstrated that in the absence of a local type I interferon response:

- CHIKV infection is significantly increased compared to WT animals by looking at both infectious particles ($p=0.001$) as well as viral RNA ($p=0.0025$) after perfusion (**Extended Data Fig. 4b**).
- Using our CHIKV reporter virus we demonstrate that in the absence of a local IFN-I response CHIKV infection spread in cardiac tissue (**Fig. 3e**).
- We showed that cardiac fibroblast (~40%) and cardiac endothelial cells (~16.5%) represent the main targets of the infection in the absence of IFNAR receptor (**Fig. 3f and Extended Data Fig 4c**).
- And finally, we demonstrate for the first time in this model cardiac tissue damage by performing immunohistochemistry against cleaved-caspase 3. Immunohistochemical analysis has been done in 5 independent animals for both WT and IFNAR KO mice (**Fig. 3g**).

While the title of the paper from *Langsjoen et al* suggests that the study demonstrate that CHIKV replicates in the heart of immunodeficient mice and speculate about the potential implications in cardiac tissue damage, the authors do not directly demonstrate these findings. Indeed, the authors themselves acknowledged the limitations of their study in their discussion section, as follows:

“Although these results provide important insights into potential effects of alphavirus infection on the cardiovascular system with remarkable consistency between replicates, only two animals were used for mock controls in gene expression studies. Therefore, differential gene expression analysis remains preliminary and should be interpreted with caution. Furthermore, animals in this study were not perfused prior to organ collection, thus both mutation data, as well as gene expression data, may be contaminated by circulating virus and contaminating host RNAs, respectively”.

The first caveat of the study is that the authors do not perfuse the hearts. In this study, the authors conclude that CHIKV infects the heart by detecting viral RNA in their RNA seq experiments, and by identifying minority

variants that seem to be heart-specific. While this last point can support that there is some tissue-specific viral replication, the authors are not showing (i) the production of viral particles, (ii) which cells are being infected, (iii) how this infection compares to a WT infection, or (iv) how the loss of IFN-I promotes viral spreading. Moreover, two years before the authors published this paper, our own group demonstrated that heart from IFNAR KO mice can harbor CHIKV infectious particles (Noval MG, et al, Cell Reports 2019, PMID: 31291581). However, because in that study perfusion was not performed, we were not able to conclude that CHIKV can actively replicate in the heart.

In addition, based solely on RNA-seq data the authors conclude tissue damage. Tissue damage should be also demonstrated through histopathological analysis and not inferred from sequencing data. The importance of additional orthogonal validation of RNA-seq findings can be exemplified by our own RNA-seq data coupled with histopathology analysis. As we mentioned in the discussion of the original manuscript (L343-348):

*“Intriguingly, using the KEGG datasets we observed upregulation of the viral myocarditis pathway and downregulation of cardiac tissue contractility pathway at 5 dpi (**Extended Data Fig. 4d-g; Extended Data Table 4**). However, we did not observe any sign of cardiac tissue damage by histopathology (**Extended Data Table S1**), suggesting that CHIKV infection is triggering transcriptional programs involved in cardiac dysfunction that are not sufficient to result in overt phenotypes of cardiac damage.”*

By all the stated above, we believe our work provides novel and fundamental results to support the conclusions on how IFN-I signaling pathway restrains CHIKV cardiac infection, preventing viral spreading and cardiac tissue damage.

However, given the reviewer’s concern and to further support the findings of the present study particularly in demonstrating the importance of IFN-I signaling in controlling CHIKV infection, we performed additional experiments. We addressed the requirement of type I interferon signaling in human primary cardiac fibroblasts by studying the dependency of CHIKV infection in the presence of an inhibitor of the JAK1/JAK2 pathway, as well as by evaluating the inhibitory potential of IFN α and β . We found that in human primary cardiac fibroblasts, inhibition of JAK1/JAK2, a critical component of the IFN-I pathway, results in increased infection. In addition, to strengthen our findings related to IFN-I effects on CHIKV infection of cardiac fibroblasts, we performed a dose-response experiment incubating primary human cardiac fibroblasts with increasing concentrations of either IFN α or IFN β . Indeed, increasing doses of either IFN α or IFN β resulted in reduced infection of primary human cardiac fibroblasts. These results demonstrate that IFN-I is fundamental to controlling CHIKV in human cardiac fibroblasts, and further support our findings in mice. These results have now been added to **Fig 3c-d** and the text (L239-253):

*“To determine if human cardiac cells respond to CHIKV infection similarly to murine cardiac tissue, we evaluated IFN-I signaling in hCFs infected with CHIKV. We evaluated this by: (i) measuring the capacity of hCFs to generate a robust IFN-I response after poly(I:C) stimulation and CHIKV infection; (ii) measuring the susceptibility of these cells to CHIKV infection in the absence of IFN-I signaling using the JAK1/JAK2 inhibitor ruxolitinib (Rux); and (iii) inducing a refractory state upon increasing doses of IFN-I pre-treatments. Indeed, we found that hCFs efficiently respond to poly(I:C) stimulation, resulting in ~ 80% of the monolayer expressing ISG15 (**Fig. 3c, left panel**). In agreement with our *in vivo* data, we found that infected hCFs responded to CHIKV infection by producing significant levels of ISG15 (**Fig. 3c, left panel**). In addition, Rux treatment significantly increased the percentage of CHIKV infected cells (**Fig. 3c, right panel**), and correlated with a reduction of ISG15 production (**Fig. 3c, left panel**). Furthermore, we found that exogenous addition of either IFN α or IFN β prevented CHIKV infection in a dose response manner, with IFN β showing increased blocking capacity (**Fig 3d**). These results support that, similarly to murine cardiac tissue, IFN-I signaling is critical in controlling CHIKV infection in human cardiac fibroblasts.”*

- 4- MAVS signaling is required for viral clearance in CHIKV-infected hearts. We demonstrate that MAVS signaling is essential to controlling CHIKV infection from cardiac tissue (**Fig4a-c**). Interestingly, we found that CHIKV RNA clearance followed different kinetics between calf muscle (primary target of infection) and cardiac tissue, with viral RNA in cardiac tissue persisting for longer times (**Fig4d-e**). These results points

towards not only the role of MAVS in controlling CHIKV infection, but also to a tissue-specific dependency on MAVS for successful clearance of CHIKV viral infection.

- 5- Persistence of CHIKV infection in cardiac tissue correlates with viral myocarditis and chronic big vessels vasculitis. We expanded all our studies regarding the pathogenesis of the MAVS KO mice to include subcutaneous inoculation. Indeed, we found that viral myocarditis and chronic vasculitis is observed up to 31 to 60 dpi in this model independently of the inoculation route. These results have now been added to **Fig 6** and **Extended Data Fig 6 and Extended Data Table S1**. The consistency of this result points towards MAVS as an underlying factor of long term CHIKV-induced cardiovascular damage.

The reviewer comment about how IFN-I or MAVS defective responses contribute to human CHIKV-heart/cardiovascular dysfunction is a critical question that it is fundamental to be addressed. Indeed, we specifically discuss the importance of these points in L409-421.

“Therefore, the risk of cardiac complications in CHIKV-infected patients could be associated with genetic and non-genetic factors (such as co-infections, diet, among others) leading to a reduced IFN-I response. Single nucleotide polymorphisms (SNPs) in TLR3 found in patients with enteroviral myocarditis are associated with a decreased TLR3 mediated signaling and shown an impaired IFN-I response after Coxsackievirus B3 infection⁴⁵. Thus, SNPs in IFN-I and MAVS signaling genes could underlie an increased susceptibility to cardiac complications in CHIKV-infected patients. SNPs in the coding region of the human MAVS can result in impaired function^{46,47}. In addition, recent studies suggested a link between age and decreased sensing through MAVS⁴⁸, which could be a contributing factor to the observed correlation between age and CHIKV severity^{5,15}. Further work should focus on determining whether variability in IFN-I and MAVS signaling among individuals can explain the differential susceptibilities to CHIKV-induced cardiac complications.”

We would like to highlight recent work started to appreciate the importance of atypical manifestations associated with arboviral infections and we have a long way to go in order to established surveillance structures in endemic areas to start addressing these aspects of arboviral disease. We see our work as the first study using animal models supporting the causality of cardiac manifestations due to CHIKV infection in the heart. However, to complement findings in laboratory settings, it is indeed necessary to collect accurate epidemiologic and clinical data. This would require an epidemiological survey paired with either targeted or whole-genome sequencing efforts to identify genetic variants of risk in the CHIKV-infected population. While we agree on the utmost importance of this type of studies, it is beyond the scope of this particular work. However, we now include in the discussion section future directions and potential studies that can be carried out either by our group or others. Indeed, other research groups interested in arboviral-associated cardiac complications might be made aware of CHIKV as a potential causal agent thanks to our findings, after publication of the present work (see L468-475):

“Overall, this study demonstrates the direct causality between CHIKV active replication in the heart and the development of cardiovascular manifestations. Future studies focalized on atypical manifestations of arboviral diseases and their pathogenesis are fundamental for understanding the full spectrum of the consequences of these infections. This knowledge is paramount for monitoring of atypical manifestations in endemic areas as well as in future epidemics. In turn, this could be leveraged for early diagnostics, prevention, and therapeutic interventions especially in endemic areas where the circulation of these viruses represents a public health burden.”

Specific Comments:

1. The Fig 1a and b legend should indicate that heart tissue is being examined. Text has been modified.
2. Line 173 and Figure 2. Figure 2 lacks panel d. We apologize for the mistake; this has been solved.
3. Lines 187 and 191. Figures referred to as Fig 3f and 3g in the text are actually Fig 2f and 2g. These figures look for CHIKV induced cTnT and cleaved caspase 3. Both panels lack positive controls to prove that cTnT and cleaved CASP3 can be detected. These controls should be included.

We thank the reviewer for pointing this out. Below the explanation about technical controls for staining and ELISA.

Cleaved CASP3 analysis. Staining for IHCs using cleaved caspase 3 antibody (CST,9579S, clone D3E9) was performed by our NYU Experimental Pathology Core. As a positive control of staining, we used spleen tissue from

mice which has a high level of apoptosis. This information has been added to the material and methods section and to **Extended Data Fig. 2b**. Text (L764-765):

“As positive controls of the IHC, sections of mice spleen were stained with anti-CASP3, anti-CD3 and anti-CD11b.”

cTnT ELISA. To measure the serum levels of cTnT in CHIKV-infected animals we used the cTnT ELISA kit from amsbio-AMS.E03T0017. This kit includes recombinant cTnT to build a calibration curve to estimate the concentration of cTnT through the interpolation of the absorbance at 450 nm (OD_{450nm}) of your unknown sample into a calibration curve ranging from 1 to 25 ng/ml of recombinant cTnT (provided by the kit). We diluted our samples to be within the limits of detection of the calibration curve and not have to extrapolate data. All samples were diluted 1 in 2 in PBS and their OD_{450nm} ranged from 0.66 to 2.01. Therefore, we are confident that the antibody and the detection method work for the detection of cTnT. Below is the calibration curve plot of our ELISA and we highlight the range of the signal where the unknown samples fell.

calib curve TnT ELISA

Figure 1. Calibration curve from cTnT ELISA experiment. In orange is indicated the OD_{455nm} values from the samples tested.

However, acknowledging the reviewer comment we added the following clarification in the materials and methods section (L804-806):

“Serum was analyzed for mouse cardiac troponin-T using a precoated microtiter plate (amsbio, AMS.E03T0017) and using a recombinant cardiac troponin-T standard curve according to the manufacturer’s instructions.

4. Line 216 Fig 4c should be Fig 3c. We apologize for the mistake; this has been solved.
5. Line 222 Fig 3c should be Fig 3d. We apologize for the mistake; this has been solved.
6. Line 253. The reference in the text to Fig 5c is incorrect. It is unclear what data this refers to. We apologize for the mistake; this has been solved.
7. Line 282 change “wean” to “wane.” Text has been modified.

Reviewer #4 (Remarks to the Author):

Prompted by observations that a fraction of human infections with chikungunya virus (CHIKV) result in cardiac complications, Noval et al used a mouse model to investigate infection of the heart during CHIKV infection. The authors find that in immunocompetent WT mice, CHIKV infection of heart tissue can be detected. This infection is rapidly cleared and appears to be benign as no evidence of cardiac tissue injury or dysfunction was detected. Using RNAseq and immunofluorescence microscopy with a recombinant CHIKV strain that expresses ZsGreen in infected cells, the authors present evidence that cells in the heart are directly infected. Further analysis revealed that the majority of infected cells in the heart are cardiac fibroblasts. In addition, the authors find that overall heart infection levels are reduced in mice carrying a genetic lesion in the *Mxra8* gene – the protein product of this gene has previously been reported to function as a cell entry receptor for CHIKV. Finally, the authors show that cardiac infection and pathology is more severe in mice with genetic defects in the type I IFN pathway, a well-established pathway for the control of CHIKV and other viral infections.

A weakness of the study is the limited conceptual advances. That is, a review of the literature indicates that prior studies have reported that (1) Alphaviruses can infect heart tissue (e.g., PMID: 34844209, PMID: 31291581, PMID: 33361425, PMID: 32615591), (2) Fibroblasts are a major cell type targeted by CHIKV and other alphaviruses (e.g., PMID: 29769725, PMID: 31465513, PMID: 31554973, PMID: 17604450), (3) CHIKV infection of tissues, particularly those distal to the site of inoculation, is reduced in mice with genetic lesions in *Mxra8* (e.g., PMID: 31484075, PMID: 32075743), (4) CHIKV infection elicits a type I IFN response (e.g., PMID: 28207896), and (5) mice with genetic defects in the type I IFN pathway, including *lfnar1* and *Mavs*, develop higher viral tissue burdens, expanded viral tropism, and more severe tissue pathology and disease (e.g., PMID: 31619554, PMID: 20123960, PMID: 18282093, PMID: 22761364).

Nevertheless, there are a number of notable strengths of this study. This is a well-written manuscript, and the authors conclusions are generally careful and supported by the data presented, with only a few concerns (see specific comments below). Technical features are excellent and appropriate controls are included. The statistical analyses seem to be appropriate and defined in the text and/or figure legends (clarification required in some cases). Finally, this appears to be the most detailed report of CHIKV infection of cardiac tissue to date and may help explain some observations in CHIKV-infected patients.

First of all, we want to thank the reviewer for highlighting the potential and the relevance of this study, as well for the constructive criticism that led us to an improved version of the manuscript.

While recent work started to appreciate the public health burden of atypical manifestations associated with arboviral infections in endemic areas, it is currently not understood how CHIKV can lead to these manifestations. Particularly for cardiac tissue, CHIKV has been reported to be associated with cardiac complications in individuals with and without comorbidities, and it is responsible of 20% of CHIKV related mortality cases. While previous studies have reported the presence of virus (antigen, viral RNA, or infectious particles) in cardiac tissue, no clear evidence of active viral replication in the tissue of immunocompetent hosts has been provided. In addition, a mechanistic understanding of how the virus interacts with cardiac tissue, how heart responds to CHIKV infection and how CHIKV infection can lead to heart disease is lacking.

In the present work, we leveraged different mouse models and have now added human primary cardiac cells to define mechanisms implicated in CHIKV cardiac tissue infection, and to identify host factors involved in clearance of CHIKV infection and tissue damage. Our results demonstrated that in the absence of MAVS signaling CHIKV infections can lead to chronic inflammation in major vessels (such as aorta, pulmonary artery), suggesting that in immunodeficient states (genetic polymorphisms, comorbidities, chemotherapy treatments, among others) infection of cardiac tissue by CHIKV can lead to vascular inflammation and potentially long-lasting cardiovascular complications.

We believe that studies focalized in atypical manifestations of arboviral diseases and their pathogenesis are fundamental to understand the full spectrum of the consequences of these infections. In turn, this could be leveraged for early diagnostics, prevention, and therapeutic interventions specially in endemic areas where the circulation of these viruses represents a public health burden.

While this reviewer makes a fair point regarding the conceptual advances of our results in the context of what is known about general CHIKV infection, we believe our findings are novel and move the field forward regarding what we know about CHIKV and atypical cardiac complications. These are the specific findings that were not defined prior to this study:

1- CHIKV actively replicates in cardiac tissue and targets cardiac fibroblasts in immunocompetent mice. We believe this is the first study that undeniably demonstrated that CHIKV can actively replicate and generate infectious progeny in cardiac tissue. By the use of multiple complementary approaches (looking at viral RNA and infectious particles, visualizing infected cells with a CHIKV reporter of active replication, and measuring at relative abundances of genomic and subgenomic RNA) we demonstrated that CHIKV can actively replicate and generate infectious particles in the myocardium, atrium, vessels and valves of immunocompetent WT mice, and we demonstrate that the primary cellular target of CHIKV infection in the heart corresponds to cardiac fibroblasts.

While previous studies have suggested cardiac tissue as a target of CHIKV infection, the lack of perfusion in such studies confounds any conclusions regarding active CHIKV replication in the cardiac tissue itself (e.g., PMID: 34844209, PMID: 31291581). In addition, while CHIKV antigen has been detected in cardiac tissue of post mortem samples of individual who died of CHIKV (PM:32615591), we believed that functional experiments demonstrating that cardiac tissue and cardiac cells are susceptible and permissive to CHIKV infection are required to direct link presence of antigen with direct infection of the tissue. However, as we mentioned in the discussion of our original manuscript (L323-325) the presence of CHIKV antigen in cardiac fibroblast of post mortem samples support similar tropisms between human and the mouse model, which validate the relevance of the use of this mouse model to study this particular aspect of CHIKV infection.

As this reviewer pointed out, several studies have shown that: (i) CHIKV can infect cell lines of fibroblast origin as well as primary human and murine dermal and muscle fibroblasts (PMID: 29769725, PMID: 17604450, PMID: 31554973) *in vitro*, and that (ii) CHIKV can infect dermal and muscle fibroblasts *in vivo* (PMID: 31465513). While our results are in agreement with the expected susceptibilities of cardiac cell types to CHIKV, our unbiased analysis identified cardiac fibroblast to be the main CHIKV infection target cell type in the heart, while cardiomyocytes represent ~ 5% of the infected cells, even in the absence of type-I interferon signaling (See Fig. 3f, and Extended Data Fig. 4c). This finding has a conceptual advance directly linked to the pathogenesis of CHIKV in the heart, since while cardiomyocytes do not represent a main cellular target, patients do develop CHIKV-induced myocarditis. This raises the possibility that CHIKV-induced myocarditis may be occurring without direct infection of cardiomyocytes. In line with this observation post mortem cardiac tissue of individual who succumb to CHIKV infection stained positive for CHIKV antigen in non-cardiomyocyte cells (cardiac fibroblast, interstitial connective tissue, vessels, vascular endothelium, epicardial adipose tissue, among others) (PMID:32615591).

2- Infected hearts clear CHIKV infection without inducing cardiac damage in WT mice and elicit a local type-I interferon response both in infected and non-infected vimentin⁺ cells. Only a small proportion of individuals develop atypical manifestations due to CHIKV infection, which suggests that in the majority of infected individuals the course of infection is resolved without major complications. However, the underlying cause that makes certain individuals more susceptible to these atypical manifestations remain largely unknown.

While previous studies have reported that CHIKV elicits a robust type-I interferons upon infection, most of the studies focused on understanding these responses in primary target tissues (PMID: 28207896, PMID: 3161955). Here we demonstrate that, in a model of immunocompetent host, cardiac tissue is infected but it is capable of controlling CHIKV infection by triggering a transient inflammatory response (characterized by the increase of pro-inflammatory cytokines such as CCL2, CCL5, IL6, and TNF α at 2 dpi), and a local type-I interferon response – in both infected and non-infected cells, that results in viral clearance from the myocardium without tissue damage.

We believe these results provide a conceptual advance in the sense that we demonstrate that cardiac tissue is a *bona fide* target of CHIKV infection, rather than a secondary target that can be infected only in the right immunosuppression context. Moreover, our results provide mechanistic evidence to demonstrate that in an immunocompetent host, a local type-I interferon response elicited from both infected and non-infected cells

in the infected cardiac tissue correlates with lack of viral infection progression and prevents overt cardiac tissue damage.

3- MAVS signaling plays a fundamental role in CHIKV clearance in cardiac tissue. We showed that specifically in the absence of the adaptor molecule MAVS, CHIKV infection persists in cardiac tissue. While previous studies have shown that mice with genetic defects in MAVS pathway develop higher viral tissue burden and show more severe disease, expanded tropism and tissue pathology, none of these studies were focalized in the study of pathogenesis and disease of cardiac tissue.

4- In the absence of MAVS signaling CHIKV-infected mice develop big vessel vasculitis independent of the inoculation route that persist up to 60 days post infection. We found that failure of clearance can lead to a consistent severe inflammation (CD3+/CD11b+ infiltrates) in the major vessels attached to the base of the heart that persist up to 60 days post infection.

The results from sections 3 and 4 above demonstrate that in the absence of a robust innate immune response, CHIKV infection can persist for several days compared to WT animals. And it is the persistence of CHIKV infection what leads to cardiac tissue damage. We show that even in the absence of cardiomyocyte infection, MAVS KO animals can develop myocarditis and big vessel vasculitis upon CHIKV infection both via subcutaneous or intravenous inoculation (**See Fig6, Extended Data Fig6 and Extended Table 1**).

A few additional Specific Comments to improve the manuscript are outlined below.

1. In a number of instances, the display of the data requires clarification. Some examples are listed below: **(a) In Fig 1c, 2a (and extended data 1a), left panel (i.e., the CHIKV RT-PCR assay), does the gray box represent the limit of detection for this assay?** Typically, these types of assays have a limit of detection. If so, how can the authors discriminate values below the limit of detection. Also, in Fig 1c and extended data 1a, the authors show Isg15 RNA expression as a fold-change. Given the mean value presented for the mock-infected group in each graph, it is unclear how these values were normalized.

We thank the reviewer for pointing this out, and clarifications have been added in figure legends. Regarding the data analysis, we want to clarify that all the reported values are above the limits of detection of the assays performed. Below a point-by-point clarification on the data analysis:

1-RT-qPCR analysis for CHIKV genome determination, the gray boxes indicate the mean of the CHIKV genomes quantification obtained for the mock-infected conditions. Since these group of samples represent non-infected conditions, we do not expect to have amplification of viral RNA. Thus, any amplification within this group of samples represents unspecific reaction and determines the background levels of the assay.

We leverage the total RNA amount added to each reaction to have the CT values for the background level of the assay above the CT values of the limit of detection. We defined the limit of detection in our assay based the calibration curve done with *in vitro* transcribed viral RNA (**See Reviewer Fig 1, right panel**).

Reviewer Fig 1. Right panel: raw data showing the CT values distribution for CHIKV-infected and mock-infected controls. Left panel: calibration curve showing the ranges spanning by the data.

In our RT-qPCR assay we run our samples for 40 cycles alongside a calibration curve that spans from 100 ng/ul to 1E-6 ng/ul of CHIKV viral RNA. These curves typically cover a range of CTs from CT=6 to CT=35 (see example below). Importantly, for different tissues, we optimized the concentration of total viral RNA to have all the determinations for the CT values within the linear range of the calibration curve. Below an example of the data distribution specifically for the analysis of Extended Fig1a (See Reviewer Fig 1):

The following clarifications have been added to the Figure legends:

“The gray box represents the background RNA levels determined for mock-infected controls”.

The following clarifications have been added to Materials and Methods (L664-681):

“For Taqman assays, a standard curve spanning a range of 100 ng/μl to 1E-6 ng/μl of CHIKV viral RNA was generated for each dataset using *in vitro* transcribed CHIKV RNAs as described above. For the different tissues, we optimized the amount of total viral RNA to be within the linear range of the calibration curve. The number of genomic (nsp4) or subgenomic (E1) CHIKV RNA was quantified by RT-qPCR using 0.25-1 μg of total RNA as template and the Taqman RNA-to-CT One-Step kit (Applied Biosystems™, Beverly, MA, USA) in a total reaction volume of 25 μl/well. The following primers and probes were used: CHIKV primers to amplify nsp4 fragment (CHIKV forward (IOLqPCR6856F): 5'-TCACTCCCTGCTGGACTTGATAGA-3'; reverse (IOLqPCR6981R): 5'-TTGACGAACAGAGTTAGGAACATACC-3') and a probe (5'-(6-carboxyfluorescein)-AGGTACGCGCTTCAAGTTCGGCG-(black-holequencher)-3'). CHIKV primers to amplify E1 fragment (CHIKV forward (IOLqPCR10865F): 5'-TCGACGCGCCCT CTTTAA-3'); reverse (IOLqPCR10991R): 5'-ATCGAATGCACCGCACACT-3' and a probe 5'- /56-FAM/ACCAGCCTG/ZEN/CACCCATTCTCAGAC/ 3IABkFQ/-3'). The final concentration of primer and probes were 1 μM and 0.6 μM, respectively. Reverse transcription was carried out on QuantStudio 3 qPCR instrument using the following protocol: 48 °C for 30 min, followed by 10 min at 95 °C. Amplification was accomplished over 40 cycles as follows: 95 °C for 15 sec, 60 °C for 1 min.”

2-Fold change data normalization. Our RT-qPCR from our Sybgreen experiments was analyzed using the $\Delta\Delta CT$ method, where we calculated fold changes based on the following equation:

$$(i) \quad \text{Fold change} = 2^{-\Delta\Delta CT}$$

$$(ii) \quad \Delta\Delta CT = \frac{(\text{CT}_{\text{target}} - \text{CT}_{\text{housekeeping}})_{\text{Sample}} - (\text{CT}_{\text{target}} - \text{CT}_{\text{housekeeping}})_{\text{Mock control}}}{\Delta\Delta CT}$$

As target we used the primer set of interest (e.g., *Ccl5*, *Isg15*, *ApoD*, *Mx1*, etc) and as housekeeping target we used 18S. The selection of the housekeeping gene was done by specifically looking for a reference gene that won't be affected in the tissue of interest during the course of the infection (See Reviewer Fig2a).

b

Raw data						Fold change analysis					
		CT1	CT2	Average	ST DEV	Analysis		DCT	DDCT (ACT- ΔCT ^{HI average})	Fold change	
CCL5						Sample 1	CHIKV 2 dpi	13.67	-4.099	17.14	
						Sample 2	CHIKV 2 dpi	14.68	-3.087	8.50	
						Sample 3	CHIKV 2 dpi	15.15	-2.623	6.16	
						Sample 4	CHIKV 2 dpi	14.90	-2.871	7.31	
						Sample 5	HI 2 dpi	17.81	0.042	0.97	
						Sample 6	HI 2 dpi	17.73	-0.039	1.03	
						Sample 7	CHIKV 5 dpi	14.69	-3.083	8.47	
						Sample 8	CHIKV 5 dpi	14.58	-3.192	9.14	
						Sample 9	CHIKV 5 dpi	14.12	-3.654	12.59	
						Sample 10	CHIKV 5 dpi	14.65	-3.118	8.68	
						Sample 11	HI 5 dpi	17.47	-0.299	1.23	
						Sample 12	HI 5 dpi	18.07	0.296	0.81	
18S						Sample 1	CHIKV 2 dpi	10.24	10.45	10.34	0.15
						Sample 2	CHIKV 2 dpi	10.10	10.28	10.19	0.13
						Sample 3	CHIKV 2 dpi	10.00	10.11	10.06	0.08
						Sample 4	CHIKV 2 dpi	10.08	10.05	10.07	0.02
						Sample 5	HI 2 dpi	10.13	10.22	10.18	0.07
						Sample 6	HI 2 dpi	9.83	9.96	9.89	0.09
						Sample 7	CHIKV 5 dpi	10.08	10.05	10.07	0.02
						Sample 8	CHIKV 5 dpi	9.92	9.91	9.92	0.01
						Sample 9	CHIKV 5 dpi	10.69	10.70	10.69	0.00
						Sample 10	CHIKV 5 dpi	9.93	10.01	9.97	0.06
						Sample 11	HI 5 dpi	9.86	10.01	9.93	0.10
						Sample 12	HI 5 dpi	9.70	na	9.70	na

ΔCT^{HI average} 17.77

Average of ΔCT determinations for HI control condition used to calculate ΔΔCT for the infected and mock infected samples.

Reviewer Fig2.a. Selection of the housekeeping gene for RT-qPCR. Example of a reference gene that is not affected in the tissue of interest during the course of the infection (18S) versus a housekeeping gene that changes with the infection.

b. Example of the RT-qPCR normalization analysis.

Because we wanted to represent the variability of the data between both the mock infected control and the infection datasets, we calculate the $\Delta\Delta CT$ for each sample (infected and mock-infected) against the average of all the ΔCT -Mock control. An example of the data normalization can be observed in **Reviewer Fig 2b**. As it can be observed in the example, the fold change value for the mock condition (HI samples) ranges between 0.83 to 1.23. This variation around 1 is important to determine statistical significance, especially in the context of poorly induced or reduced targets.

We added the following clarifications to materials and methods (L696-699):

“The relative expression of the respective genes to 18S expression was calculated using the $\Delta\Delta CT$ method, and values were expressed as fold change normalized to mock-infected control. $\Delta\Delta CT$ for each sample (infected and mock-infected) was calculated against the average of all the ΔCT values calculated for the mock condition.”

(b) Fig 2c, and Extended data 2. For some analytes, many or all the values reported are below the detection limit. Despite this, the authors seem to be able to differentiate the amount detected in individual samples and seem use these different values for statistical analysis. This requires clarification.

The values reported for all the analytes were within the detection limit of the method and represent the result values from the Luminex quantifications based on the individual calibration curves for each analyte. The gray shades represent the last value of the calibration curve, and therefore, the value below which the concentrations were calculated by extrapolation. The reason we believe it is informative to show this information is because, for some analytes (i.e. G-CSF and IL10), we see significant production of cytokine in cardiac tissue after CHIKV infection ($p=0.0117$ and $p=0.0282$, respectively) even if for the infected conditions some or all the results were obtained by extrapolation. By extrapolating, the software is assuming the curve follows a similar pattern outside the fitted range, which might not be necessary true.

The following clarifications have been added to the figure legends:

“Values within the gray shading represent concentrations of the analyte that were beyond the calibration curve and therefore determined by extrapolation of the curve by the Luminex Software

(c) Fig 5: Some of the data points shown do not seem to match the figure legend. For example, there appear to be 3 data points in 5a for MAVS KO at 2 dpi. Data points in 5c WT 15 dpi, 5f 5dpi, and 5f WT 7 dpi also do not seem to match the legend.

We thank the reviewer for pointing this out. The figure legend has been modified.

2. Based on the data shown in Fig 1c, the authors argue that the differences in heart infection between WT and *Mxra8delta8* mice following ID inoculation is not due to effects on dissemination based on the similarities in viremia. These findings seem at odds with prior studies in *Mxra8delta8* mice that have shown defects in viral dissemination to sites distal to the inoculation site. In the next section, the authors then rationalize the use of the iv system for much of the study due to variability in dissemination, suggesting that the authors do observe variability following ID inoculation. Perhaps I am just misinterpreting some of the statements. Some clarification would be useful.

We thank the reviewer for pointing this out. In the original version of the manuscript, we stated “*Importantly, viral RNA levels in serum remained unaltered (Fig 1c), excluding the possibility that the observed decrease in CHIKV in Mxra8delta8 infected hearts is driven by reduced viremia*”. This statement was based on our data where we do not see statistically significant differences between CHIKV serum levels by qPCR ($p=0.0766$) for WT or *Mxra8* deficient animals. However, the reviewer is correct regarding that the level of infectious particles is reduced in this model during acute infection and subcutaneous inoculation as reported by Zhang et al (**PMID: 31484075**). Therefore, since viremia levels are affected between the models, we cannot differentiate whether the observed reduction in CHIKV RNA in heart tissue of *Mxra8* deficient mice is due to (i) a reduce viremia, (ii) a direct dependency of the *Mxra8* receptor for infection, or (iii) a combination of both. We want to underscore that while further characterization of the receptors involved in CHIKV infection would be valuable, the focus of the current study is centered in understanding how the cardiac tissue responds to CHIKV infection, and identify mechanisms implicated in development of CHIKV-

mediated heart disease. Therefore, considering the comments from this and other reviewers, and the conflicting data regarding CHIKV receptors, we decided to remove these data from our study.

3. Fig 5. The authors conclude that the kinetics of viral infection in cardiac tissue of *Mavs*^{-/-} mice is unique based on comparison to the serum. To further support this idea, the authors should evaluate viral burden in a broader range of tissues of WT and *Mavs*^{-/-} mice.

We thank the reviewer for this suggestion. In this new version, we have now included viral burden analysis in calf muscle – a primary site of CHIKV infection. We evaluated for both WT and MAVS KO the levels of viral RNA in calf muscle at 2, 10, and 15 dpi. We found that CHIKV clearance in the calf muscle is different from cardiac tissue. In calf muscle, both MAVS KO and WT mice showed a progressive reduction of viral RNA over the days, and at 10 dpi no significant differences in CHIKV RNA were observed. However, for cardiac tissue, while we observed a progressive clearance in the RNA levels of WT mice, for MAVS KO mice we observed that the levels of viral RNA are maintained up to 10 dpi, and evidence of RNA decay is observed by 15 dpi. Notably, we also found that the top part of the heart (vessels and atrium) is the one that harbors higher levels of viral RNA than the bottom part of the heart (myocardium). These results further support the observation that the kinetics of viral clearance in MAVS KO is tissue-specific, with the infection in cardiac tissue persisting for longer time points, particularly in the region containing vessels. These results have been added to **Fig. 4e-f**, and the following modifications have been added to the text (L294-307):

“To evaluate whether this deficiency in viral clearance from cardiac tissue is observed in other tissues we evaluated the levels of CHIKV RNA in calf muscle and the heart at 2, 10, and 15 dpi for both WT and *Mavs*^{-/-} mice. We found that the kinetics of CHIKV clearance from the calf muscle is different from cardiac tissue. For cardiac tissue, we observed a progressive clearance in the RNA levels of WT mice, for *Mavs*^{-/-} mice we observed that the levels of viral RNA are maintained up to 10 dpi, and evidence of RNA decay is observed by 15 dpi (**Fig. 4d**). However, in calf muscle, both *Mavs*^{-/-} and WT mice showed a progressive reduction of viral RNA over time, and at 10 dpi no significant differences in CHIKV RNA were observed (**Fig. 4e**). Interestingly, when we evaluated the CHIKV RNA levels in the top section or bottom section of the heart at 10 dpi, we observed higher viral RNA levels in top sections relative to the bottom sections (**Fig. 4f**). These results are in agreement with our observation in WT animals (**Fig. 1d**), and supports that heart sections enriched in atria and/or vessels are more prone to sustained infection over time. Collectively, these results demonstrated that sensing through MAVS is required for clearance of CHIKV infection from cardiac tissue.”

Minor specific comments

1. In the legend for Fig 1a:

- (a) It would be helpful to indicate that heart tissue was evaluated.
- (b) In addition, the authors extracted heart tissue from both uninfected and infected animals.
- (c) Please define the grey boxes shown in Fig 1a, left panel.

Modifications have been added to legend of Fig 1.

2. Line 148: The authors state that “ZsGreen signal colocalized with 78.5% of vimentin+/SMA- cells.” Is this accurate? Or do the authors mean to state that 78.5% of the ZsGreen positive cells were vimentin+/SMA-?

3. Fig 2b. The authors should clarify the extent to which ZsGreen positive cells were detected at 5 dpi. This figure has been removed from the new version of the manuscript.

4. Fig 3b. Please check the < symbols for accuracy. We thank the reviewer for pointing this out.

5. Line 187: Please check the figure call out We thank the reviewer for pointing this out.

6. Line 191: Please check the figure call out We thank the reviewer for pointing this out.